# The impact of lightning and radar reflectivity factor data assimilation on the very short term rainfall forecasts of RAMS@ISAC: application to two case studies in Italy

Stefano Federico[1], Rosa Claudia Torcasio[1], Elenio Avolio[2], Olivier Caumont[3], Mario Montopoli[1], Luca Baldini[1], Gianfranco Vulpiani[4], Stefano Dietrich[1]

1. ISAC-CNR, via del Fosso del Cavaliere 100, Rome, Italy
2. ISAC-CNR, zona Industriale comparto 15, 88046 Lamezia Terme, Italy
3. CNRM UMR 3589, University of Toulouse, Météo-France, CNRS, 42 avenue G. Coriolis, 31057 Toulouse, France
4. Dipartimento Protezione Civile Nazionale Ufficio III - Attività Tecnico Scientifiche per la Previsione e Prevenzione dei Rischi, 00189 Rome

## Abstract

In this paper, we study the impact of lightning and radar reflectivity factor data assimilation on the precipitation VSF (Very Short-term Forecast, 3 hours in this study) for two severe weather events occurred in Italy. The first case refers to a moderate and localised rainfall over central Italy occurred on 16 September 2017. The second case, occurred on 9 and 10 September 2017, was very intense and caused damages in several geographical areas, especially in Livorno (Tuscany) where nine people died.

The first case study was missed by several operational forecasts, including that performed by the model used in this paper, while the Livorno case was partially predicted by operational models.

We use the RAMS@ISAC model (Regional Atmospheric Modelling System at Institute for Atmospheric Sciences and Climate of the Italian National Research Council), whose 3D-Var extension to the assimilation of RADAR reflectivity factor is shown in this paper for the first time.

Results for the two cases show that the assimilation of lightning and radar reflectivity factor, especially when used together, have a significant and positive impact on the precipitation forecast. For specific time intervals, the data assimilation is of practical importance for civil protection purposes because changes a missed forecast of intense precipitation ($\geq 40$ mm/3h) in a correct one. While there is an improvement of the rainfall VSF thanks to the lightning and radar reflectivity factor data assimilation, its usefulness is partially reduced by the increase of the false alarms, especially when both data area assimilated.

**Keywords:** data assimilation, lightning, radar reflectivity factor, RAMS@ISAC.

## 1. Introduction

Initial conditions of numerical weather prediction (NWP) models are a key point for a good forecast (Stensrud and Fritsch, 1994; Alexander et al., 1999). Nowadays limited area models are operational

at the kilometric scale  (< 5 km) and data assimilation of observations with high spatio-temporal
resolution as lightning or radar reflectivity factor[1], is crucial to correctly represent the state of the
atmosphere at local scale (Weisman et al., 1997; Weygandt et al., 2008). This is especially important
over the sea, where the absence of local observations can misrepresent convection.
The assimilation of radar reflectivity factor is useful to improve the weather forecast considering
the high spatio-temporal resolution of  radar data.
First attempts to assimilate radar reflectivity factor are reported in Sun and Crook (1997, 1998), who
expanded VDRAS (Variational Doppler Radar Analysis System) to include microphysical retrieval.
Following these studies, several systems to assimilate radar observations, both Doppler velocity and
reflectivity factor, were developed (Xue et al., 2003, Zhao et al., 2006; Xu et al., 2010). All these
studies showed the stability and robustness of assimilating radar observations as well as the
improvement of weather forecast.
In addition to direct methods, which assimilate the radar reflectivity factor adjusting the
hydrometeor contents, there are indirect methods adjusting other variables. In particular, the
method of Caumont et al. (2010) assimilates the relative humidity field. It consists of two different
steps: a 1D retrieval of relative humidity (pseudo-profile), which depends on the radar reflectivity
factor observations, followed by 3D-Var assimilation of the pseudo-profile. This method has the
advantage to reduce the computational cost at the kilometric scale.
The choice of updating the moisture field directly is motivated by its greater impact on analyses and
forecasts in comparison to that of hydrometeor-related quantities (e.g., Fabry and Sun, 2010).
Caumont et al. (2010) showed that the method improved the weather prediction of a heavy
precipitation event in southern France and of an eight-day long assimilation cycle experiment.
The method was applied in other studies (Wattrelot et al., 2014, using AEROME model; Ridal and
Dalbom, 2017; using HARMONIE model), or modified using 4D-Var in place of 3D-Var (Ikuta and
Honda, 2011; using JNoVa model) showing its capability to improve the weather forecast. The
methodology is also used in the operational context (Wattrelot et al., 2014).
Lightning is another important source of asynoptic data due to its ability to locate precisely the
convection with few temporal gaps (Mansell et al., 2007). In the last two decades, there have been
attempts to assimilate lightning into meteorological models both at low horizontal resolution, which

---

[1] Throughout the paper we use the expression radar reflectivity factor, which is the quantity provided by the radar (and expressed in $mm^6 m^{-3}$ or dBz) after conversion from the received power. The radar reflectivity factor is different from reflectivity and is obtained in the special case of Rayleigh scattering. Reflectivity is not the quantity that radars usually provide and display on their screens although most of people refer to it.

need a cumulus parameterization scheme to simulate convection, and at convection permitting scales.

First attempts to assimilate lightning in NWP models were based on relationships between lightning and rainfall rate estimated by microwave sensors on board polar satellites (Alexander et al., 1999; Chang et al., 2001; Jones and Macpherson, 1997; Pessi and Businger, 2009). In this approach, the rainfall rate was computed as a function of the density of lightning observations and then transformed into latent heat, which was assimilated. The results of these studies showed a positive impact of the lightning data assimilation on the forecast up to 24h also for fields at the large scale, as sea-level pressure.

The study of Papadopulos et al. (2005) used lightning to locate convection and the simulated water vapour profile was nudged towards vertical profiles recorded during convective events.

Mansell et al. (2007) modified the Kain-Fritsch (Kain and Fritsch, 1993) cumulus convective scheme to force convection when/where flashes are observed while the convective scheme was not activated in the model simulation, demonstrating the potential of lightning to improve the convection forecast. A similar approach was introduced by Giannaros et al. (2016) into WRF showing the positive impact of lightning data assimilation on the precipitation forecast up to 24h for eight convective events occurred over Greece.

Fierro et al. (2012) introduced a methodology to assimilate lightning at convection-resolving scales by modifying the water vapour mixing ratio simulated by the WRF according to a function depending on the flash-rate and on the simulated graupel mixing ratio. The water vapour could be assimilated by nudging (Fierro et al., 2012) or 3D-Var (Fierro e al., 2016).

Qie et al. (2014), using WRF, adopted the methodology of Fierro et al. (2012) to assimilate ice crystals, graupel and snow, showing promising results for deep convective events in China.

Fierro et al. (2015) studied the performance of the Fierro et al. (2012) method for 67 days spanning the 2013 warm season over the CONUS giving a statistically robust estimation of the performance of the method. The computationally inexpensive lightning data assimilation method improved considerably the short-term (≤ 6h) precipitation forecast of high impact weather.

Lynn et al. (2015) and Lynn (2017) also applied the method of Fierro et al. (2012) to boost the local thermal buoyancy where/when lightning is observed. Results show that lightning data assimilation improved lightning forecast. Importantly, Lynn et al. (2015) offer an approach to address spurious convection (i.e., convection removal), which is a more challenging problem to tackle.

Federico et al. (2017a) implemented the methodology of Fierro et al. (2012) in RAMS@ISAC model,
showing the systematic and significant improvement of the precipitation forecast at the very short
range (3h) for twenty case studies occurred over Italy; the impact of lightning data assimilation for
longer time ranges (6h-24h; Federico et al., 2017b) showed considerable impact on the 6h
precipitation forecast, with smaller (negligible) effects at 12 h (24 h).
In this paper, we study the impact of radar reflectivity factor and lightning data assimilation on the
very short term (3h) rainfall prediction for two case studies in Italy. We use the method of Fierro et
al. (2012) to assimilate lightning and the method of Caumont et al. (2010) to assimilate the radar
reflectivity factor. The case studies occurred in September 2017. The first case, hereafter also
referred to as Serano, occurred on 16 September, was characterized by moderate-intense and
localized rainfall. The second case, hereafter also referred to as Livorno, occurred on 09-10
September, and was characterized by deep convection and very intense precipitation in several
parts of Italy. Even if the Livorno case occurred before the Serano case, we reverse the chronological
order in the discussion, ordering the event from the less intense to the most intense.
The forecast of severe events at the local scale still remains a challenge because of the multitude of
physical processes involved over a wide range of scales (Stensrud et al., 2009). The Serano case
study, being localized in space, poses challenges in forecasting the exact position and timing of
convection initiation; the Livorno event involves the interaction between a high impact storm and
the complex orography of Italy, which is difficult to simulate at the local scale. For the above reasons
the forecast of both events was challenging, as confirmed by the poor forecast of RAMS@ISAC. The
difficulty to forecast timely and accurately the precipitation field is the reason for choosing them as
test cases.
This paper presents for the first time the assimilation of the total lightning (intra cloud + cloud to
ground) and radar reflectivity factor in RAMS@ISAC and shows how the assimilation of the radar
reflectivity factor works together with total lighting data assimilation. Also, this paper shows that
the precipitation forecast using cloud scale observations over complex terrain can be accurate,
contributing to a number of works on the same subject.
The paper is organized as follows: Section 2 gives details on the synoptic environment of the case
studies showing daily precipitation, lightning and radar observations; Section 3 gives details on the
meteorological model, lightning and radar data assimilation; Section 4 shows the results for three
very short-term forecast (VSF), one for Serano and two for Livorno; Discussion and conclusions are
given in Section 5. This paper has additional material where we discuss: a) how the lightning and

radar reflectivity factors data assimilation impact the total water field evolution; b) the sensitivity

of the results to the choice of key parameters of lightning data assimilation; c) the sensitivity of the

results to two aspects of the radar formulation; d) the sensitivity of the results to two aspects of

RAMS@ISAC setting; e) the impact of lightning data assimilation for a well predicted case study.

Supplemental material gives also the form of the forward radar operator.

## 2. The case studies

*2.1 The 16 September 2017 (Serano) case study*

During the 16 September 2017 Italy was under the influence of a cyclone that developed to the lee

of the Alps. The storm crossed Italy from NW to SE leaving light precipitation over most of the

peninsula with moderate rainfall over Central Italy. Figure 1 shows the precipitation recorded by

the Italian raingauge network on 16 September 2017. Light precipitation (< 5 mm/day) is reported

by 1018 raingauges out of the 1666 stations measuring precipitation ($\geq$ 0.2 mm/day) on this day.

Fourteen stations over Central Italy recorded more than 50 mm/day. The maximum precipitation

was 90 mm/day in Città di Castello (Umbria Region, Figure 1). Because the meteorological radar

closest to the maximum precipitation is over mount Serano (Figure 1), hereafter this event will be

referred to as Serano.

The synoptic condition during the event is shown in Figure 2. At 500 hPa (Figure 2a) a trough,

elongated in the SW-NE direction, extends over Western Europe and air masses are advected from

SW towards western Alps. The interaction between the airflow and the Alps generates a low

pressure to the lee of the Alps over Northern Italy.

The analysis at the surface (Figure 2b) shows the meteorological front represented by the equivalent

potential temperature gradient between air masses advected over the Mediterranean Sea from NW

and air masses advected from the South over the Tyrrhenian Sea. Notable is the feeding of warm

unstable air masses towards Central Italy.

Infrared satellite images (Figure 3), from 00 UTC on 16 September to 00 UTC on 17 September, show

the cold front structure moving slowly from NW to SE. Interestingly, at 00 UTC on 16 September, it

is apparent the well-defined cloud system over Central Italy (red circle of Figure 3a), which caused

most of the daily precipitation observed between 43.50 and 45.0 N.

The well-defined cloud system over Central Italy is also shown in the radar Constant Altitude Plan

Position Indicator (CAPPI) at 3 km above sea level at 02 UTC on 16 September (Figure 4). This CAPPI

is formed by interpolating all the available data from the federated Italian radar network

coordinated by the Department of Civil Protection (twenty-two radars, see Section 3.3 for their positions) and it is also referred to as the national radar composite (hereafter also mosaic). Several convective cells exceeding 35 dBz can be noted over central-northern Italy. Importantly, the cloud system over Central Italy shown by the satellite infrared channel at 00 UTC (Figure 3a) and that of the radar at 02 UTC have similar positions, showing that the cloud system was active for several hours over Central Italy.

Figure 5 shows the lightning recorded by the LINET network (Betz et al., 2009) on 16 September 2017. More than 105.000 flashes were recorded; most of them occurred in the afternoon and evening, but a secondary maximum occurred in the night, from 00 UTC to 06 UTC. In this phase, more than 3000 flashes were observed over Central Italy.

*2.2 The 09-10 September 2017 (Livorno) case study*

During the days 09 and 10 September 2017, Italy was hit by a severe storm characterised by intense and widespread rainfall over the country. Figure 6a shows the precipitation on 09 September recorded by the Italian raingauge network. Rainfall was intense over the Alps, where the maximum daily precipitation was observed (193 mm/day), and over Liguria, with precipitation of the order of 30-50 mm/day. One station over Tuscany reported 90 mm/day, showing that intense precipitation already started over the Region. The storm on 09 September was intense : 20 raingauges reported more than 100 mm/day and 70 raingauges more than 60 mm/day. In most cases, this precipitation occurred in few hours.

The following day (see Figure 6b) had higher rainfall. Precipitation occurred mainly over Central Italy, especially over Lazio, and over Northern Italy, in particular the North-East. In Tuscany, the two stations close to the sea, in the Livorno area, recorded about 150 mm/day mostly fallen in the hours between 00 and 06 UTC.

Synoptic conditions leading to this storm are shown in Figure 7. At 500 hPa (Figure 7a) a trough extends from Northern Europe towards the Mediterranean. The interaction between the air-masses and Western Alps generated a pressure low to the lee of the Alps, which crossed the whole peninsula from NW to SE. It is noted the divergent flow over Central and Northern Italy favouring upward motions.

At the surface, Figure 7b, the equivalent temperature gradient over the western Mediterranean is caused by the contrast between air masses pre-existing over the sea and air masses advected from France towards the Mediterranean. The pressure field at the surface advects air masses from the

South over the Tyrrhenian Sea. These warm and humid air masses feed the cyclone during its
development.
From a synoptic point of view, Livorno and Serano cases are similar and represent two cyclones
developing to the lee of the Alps (Buzzi and Tibaldi, 1978). However, the Livorno case is more intense
than Serano.
The notable intensity of the Livorno case is confirmed by the lightning observations (Figure 8).
During the evening of 9 September (after 18 UTC) about 38.000 flashes were recorded by LINET. On
10 September about 290.000 flashes were recorded over Italy, following the movement of the storm
propagating from NW to SE. So, more than 300.000 flashes were recorded from 18 UTC on 09
September to 00 UTC on 11 September, which are more than three times those recorded for Serano.
Thermal infrared satellite images (channel, 10.8 micron; Figure 9) show the extension of the cloud
coverage every 12 hours. It is well evident the cloud system associated with the cold front over
Europe. More specifically, the satellite image at 00 UTC shows the cloud system over Livorno area
(red circle in Figure 9b), before the most intense precipitation period over Tuscany (00-06 UTC),
while Figure 9c shows the cloud system over Central Italy (orange circle), at the end of the period
of intense precipitation over Lazio (06-12 UTC).
We conclude the synoptic analysis of the case study with two CAPPI at 3 km observed by the radar
network of the Department of Civil Protection. The CAPPI in Figure 10a, at 00 UTC on 10 September,
shows the cloud system over Tuscany with reflectivity factor up to 40 dBz. Other clouds cause
rainfall over northern Italy. The CAPPI of Figure 10a is the last assimilated by the 00-03 UTC VSF on
10 September shown in Section 4.2.1.
Figure 10b shows the CAPPI of the national radar mosaic at 3 km above the sea level and at 06 UTC.
The cloud system is moving towards Central Italy with reflectivity up to 45 dBz. Other cloud systems
are apparent over northern Italy. Figures 10a-10b well represent the movement of the storm
towards SE and Figure 10b shows the last CAPPI assimilated by the 06-09 UTC VSF shown in Section

219     4.2.2.


**3.Data and Methods**
*3.1 RAMS@ISAC and simulations set-up*
The RAMS@ISAC is used as NWP driver in this work. The model is based on the RAMS 6.0 model
(Cotton et al., 2003) with the addition of four main features, as well as a number of minor
improvements. First, it implements additional single moment microphysical schemes, whose
performance is shown in Federico (2016): among them, the WSM6 (Hong and Lim, 2006) is used in
this paper. Second, it predicts the occurrence of lightning following the diagnostic method of Dahl
et al. (2011), the implementation being discussed in Federico et al. (2014). Third, the model
assimilates lightning through nudging (Fierro et al., 2012, 2015; Federico et al., 2017a). Fourth, the
model implements a 3D-Var data assimilation system (Federico, 2013, hereafter also RAMS-3DVar),
whose extension to the radar reflectivity factor is presented in this paper (Section 3.3).
The list of the physical parameterisation schemes used in the simulations of RAMS@ISAC is shown
in Table 1.
Considering the domains and the configuration of the grids (Figure 11 and Table 2), two different
set-ups are used for Serano and Livorno. For the first case, we use the domains D1 and D2, while for
Livorno we use also the domain D3. The first domain covers a large part of Europe and extends over
the North Africa. Grid horizontal resolution is 10 km (R10). The second domain covers the whole
Italy and part of Europe and the grid has 4 km horizontal resolution (R4). The third domain covers
the Tuscany Region, has 4/3 km horizontal resolution (R1), and it is used for Livorno to represent
with higher spatial detail the precipitation field over Tuscany. The fine structures of the precipitation
field are smeared out over Tuscany using only domains D1 and D2. The operational implementation
of the RAMS@ISAC model uses the domains D1 and D2 and no refinements for specific areas of Italy
are used because Italy is a complex orography country and grid refinements for a specific event can
be done only a-posteriori, i.e. after the occurrence of the event.
All domains share the same vertical grid. It covers the troposphere and the lower stratosphere.
Vertical levels are more packed close to the ground. Among the 36 levels used in this paper 10 are
below 1 km, 14 below 2 km and 17 below 3 km. The first vertical level is at 50 m above the surface
in the terrain following coordinates used by RAMS@ISAC, the level 21 is at 5122 m. Above 6 km the
model levels are about 1000 m apart, while the maximum allowed distance between two levels is
1200 m. The complete list of the vertical levels is shown in the supplemental material of this paper
(Table S2).
The vertical grid is the same as the operational setting of RAMS@ISAC and is a compromise between
vertical resolution and computing time. The number of vertical levels will be increased to 42, starting
from September 2019, to better resolve the phenomena in this direction (Planetary Boundary Layer
processes, vertical motions, interaction between air masses and orography etc.), nevertheless the
current setting was successfully applied to the forecast of several heavy precipitation events over
Italy. A sensitivity test, using 42 vertical levels for the Livorno case, shows similar results to those
reported in the next section. Details on this simulation can be found in the supplemental material
of this paper.
The nesting between the first and second domains is one-way, while the nesting between the
second and the third domains is two-way.
VSF is implemented as shown in Figure 12. First a run with R10 configuration is performed using the
0.25° horizontal resolution GFS analysis/forecast cycle issued at 12 UTC as initial and boundary
conditions. R10 run, which starts at 12 UTC on 16 September for Serano and at 12 UTC on 09
September for Livorno, lasts 36 h and doesn't assimilate neither radar reflectivity factor nor
lightning. The R10 run is not updated after the acquisition of new data by the analysis system and
this is a limitation of the results shown in this paper. However, a sensitivity test for Livorno case
study shows that this limitation doesn't have a significant impact on the results presented in the
next Section. Details on this experiment can be found in the supplemental material of this paper.
Starting from 12 UTC, ten VSF are performed using R4 for Serano and both R4 and R1 for Livorno.
The VSF lasts 9h and uses R10 simulation as initial and boundary conditions (one-way nesting). The
9h forecast is divided into two parts: the first six hours are the assimilation stage when RAMS@ISAC
simulation is adjusted by data assimilation, whereas the last three hours are the forecast stage,
without data assimilation. During the assimilation stage, flashes are assimilated by nudging (Section
3.2), while radar reflectivity factor is assimilated every one-hour by RAMS-3DVar (Section 3.3).
It is noted that data assimilation is performed over the domain D2 (R4) only, and the innovations
are transferred to the domain D3 (R1), for the Livorno case, by the two way-nesting. The domain D3
is used for the Livorno case to refine the resolution of the precipitation field over Tuscany and to
show the spatial and temporal precision of the precipitation forecast over Tuscany using data
assimilation. However, its usage is exceptional because, as stated above, Italy is a complex
orography country and grid refinements for specific areas are used only after the occurrence of the
event. For this reason, the domain D3 is usually not used in RAMS@ISAC and no statistics about the
background error are available for this grid.
Because lightning and radar reflectivity factor are cloud scale observations, their assimilation at
higher horizontal resolution by 3D-Var is foreseeable in future works.
The verification of the VSF for precipitation is done by visual comparison of the model output with
the raingauge network of the Department of Civil Protection, which has more than 3000 raingauges
all over Italy.
In addition we consider the FBIAS (Frequency Bias; range [0, + ∞) ), where 1 is the perfect score, i.e.
when no misses and false alarms occur), POD (Probability of Detection; range [0, 1], where 1 is the
perfect score and 0 the worst value), ETS (Equitable Threat Score; range [-1/3,1], where 1 is the
perfect score and 0 is a useless forecast), TS (Threat Score; range [0,1] where 1 is the perfect score
and 0 the worst value). Scores are computed from 2x2 dichotomous contingency tables (Wilks,
2006) for different rainfall thresholds and for different neighbourhood radii. Moreover,
performance diagrams (Roebber, 2009) are used to summarise the scores.

*3.2 Lightning data assimilation*
Lightning data are provided by LINET (LIghtning detection NETwork; Betz et al., 2009;
www.nowcast.de) which has more than 500 sensors worldwide with the greatest density over
Europe (more than 200 sensors). The network has a good coverage over Central Europe and
Western Mediterranean (from 10 W to 35 E and from 30 N to 60 N). The area of good coverage
includes the region considered in this paper.
LINET exploits the VLF/LF electromagnetic bands and provides measurements of both intra-cloud
(IC) and cloud to ground (CG) discharges. IC strokes are detected as long as lightning occurs within
120 km from the nearest sensor thanks to an optimised hardware and advanced techniques of data
processing (TOA-3D, Betz et al., 2004). According to Betz et al. (2009), LINET has a location accuracy
of 125 m for an average distance of 200 km among the sensors verified by strikes into towers of
known positions.
The good performance of the LINET network and its ability to detect IC strokes is shown in
Lagouvardos et al. (2009) for a storm in southern Germany, while the good performance over Italy,
including both CG and IC strokes, is discussed in Petracca et al. (2014).
The lightning data assimilation scheme is that of Fierro et al. (2012; 2014; 2015) and uses the total
lightning, i.e. intra-cloud plus cloud to ground flashes.
The method starts by computing the water vapour mixing ratio $q_v$:

$$q_v = Aq_s + Bq_s \tanh(CX)(1 - \tanh(Dq_g^\alpha))$$

315                                                                                                              (1)

Where coefficients are set to A=0.86, B=0.15, C=0.30, D=0.25, $\alpha$=2.2, $q_s$ is the saturation mixing ratio
at the model atmospheric temperature, and $q_g$ is the graupel mixing ratio (g kg$^{-1}$). *X* is the number
of total flashes (IC+CG) falling in a grid box of domain D2 (R4) in the past five minutes. The mixing
ratio $q_v$ of Eq. (1) is computed only for grid points where flashes are recorded. More specifically, for
each grid point we consider the number of flashes falling in a grid box centred at the grid point in
the last five minutes. The mixing ratio of Eqn. (1) is compared with that predicted by the model. If
the mixing ratio of Eqn. (1) is larger than the simulated one, the latter is nudged towards the value
of Eqn. (1), otherwise the modelled mixing ratio is left unchanged. This method can only add water
vapour to the forecast.
The check and eventual substitution of the water vapour is performed every five minutes and it is
made within the mixed phase layer zone (0 °C, -25°C), wherein electrification processes caused by
the collision of ice and graupel are the most active (Takahashi 1978, Emersic and Sounders, 2010;
Fierro et al., 2015).
The scheme of Fierro et al. (2012; 2015) was adapted to RAMS@ISAC in Federico et al. (2017a). In
particular, the coefficient C of Eqn. (1) was rescaled from that of Fierro et al. (2012) considering the
different spatio-temporal resolution of gridded lightning data; then the coefficient C was tuned
(increased) by trials and errors considering two case studies of HyMeX-SOP1 (15 and 27 October
2012). The C constant was adapted subjectively as a compromise of increasing the hits and
minimising false alarms. POD and ETS scores were considered as metrics for this purpose. Then, Eqn.
(1) was applied to twenty case studies of HyMeX-SOP1 giving a statistically significant (90, or 95%
depending on the rainfall threshold) improvement of the RAMS@ISAC precipitation VSF (3h).
Nevertheless, a definitive statistic on the performance of rainfall VSF to nudging formulation in
RAMS@ISAC is missing and further studies are needed in this direction. Also, the optimal choice of
the coefficients A, B, C, D and $\alpha$ is case dependent.
Fierro et al (2012) applied the method using the ENTLN network, which has a detection efficiency
(DE) greater than 50% for IC over Oklahoma, where the ENTLN data were used. The emphasis on IC
flashes in the set-up of Fierro et al. (2012) is given because observational and model studies have
provided evidence that IC flashes correlate better than CG flashes with various measures of
intensifying convection (updraft strength, volume, graupel mass flux etc.; MacGorman et al. 1989;
Carey and Rutledge 1998; MacGorman et al. 2005; Wiens et al. 2005; Kuhlman et al. 2006; Fierro et
al. 2006; Deierling and Petersen 2008; MacGorman et al. 2011). For these reasons methods using
both IC and CG flashes perform better than those using CG only, being CG flashes correlated with
the descent of reflectivity cores and the onset of the demise of the storm's updraft core
(MacGorman and Nielsen, 1991).
The analysis of the case studies shows that IC strokes are about 30% of the total number of strokes
reported by LINET. Also, the fraction of IC strokes to the total strokes depends on the position. For
example, for the Serano case, the fraction of IC strokes detected by LINET over the area hit by the
largest precipitation is more than 50% while over the Adriatic Sea it decreases to 10%.
It is also noted that DE for IC strokes cannot be reliably compared between LINET and ENTLN,
because the area is different and the technical details about IC detection remain unclear (type of
signals, VLF/LF or VHF, discrimination IC-CG).
For all the above reasons the application of the Fierro method to RAMS@ISAC is not straightforward
and it is appropriate to study the dependence of the rainfall VSF to the nudging formulation. This
subject is studied in the supplemental material of this paper (Section S.3) and the results show that
the choice of the coefficient of Eqn. (1) used in this paper is reasonable.
It is finally noted that despite the limitations noted above, the lightning data assimilation, with the
setting of this paper, had a significant and positive impact on RAMS@ISAC rainfall VSF (Federico et
al., 2017a; 2017b).


*3.3 Radar data assimilation*
The method assimilates CAPPI of radar reflectivity factor operationally provided by the Italian
Department of Civil Protection (DPC). Radar data are provided over a regular Cartesian grid with 1
km horizontal resolution and for three vertical levels (2, 3, 5 km above the sea level). The CAPPIs at
2, 3, and 5km can be considered as under-sampled vertical profiles. CAPPIs are composed starting
from the 22 radars of the Italian Radar Network (Figure 13) 19 operating at the C-band (i.e., 5.6 GHz)
and 3 at X-band (i.e., 9.37 GHz). Data quality control and CAPPI composition is performed by DPC.
Data quality processing chain aims at identifying most of the uncertainty sources as clutter, partial
beam blocking and beam broadening. The radar observations are processed according to nine steps
detailed in Vulpiani et al. (2014), Petracca et al. (2018) and references therein.
Radial velocity is not assimilated into RAMS@ISAC because it is not operationally processed, the
scan strategy being optimized for QPE purposes. Furthermore, the implementation of a radial
velocity data assimilation scheme is under development in RAMS-3DVar and it is not currently
available for testing. For these reasons, we didn't consider the assimilation of this parameter.
Before entering data assimilation, the Cartesian grid is downscaled to 5 km by 5 km in order to
reduce the numerical cost of the data assimilation and the effect of correlated observation errors
(Rohn et al., 2001). Thus, the radar grid (Figure 4, for example) is a Cartesian grid with 5 km grid-
spacing and three vertical levels.
It is important to note that pure sampling of the data could result in implementation of errors (for
example reflectivity given by insects or birds) or extremes. Creating superobservations would
reduce this problem, the main drawback being the missing of very localised phenomena. While the
aim of this paper is to present the update of the data assimilation system of RAMS@ISAC and its
application to two challenging cases, the problem of using superobservations will be considered in
future studies because it impacts the results.
The methodology to assimilate radar reflectivity factor is that of Caumont et al. (2010), named
1D+3DVar, which is a two-step process: first, using a Bayesian approach inspired to GPROF (Goddard
Profiling Algorithm; Olson et al., 1996; Kummerow et al., 2001), 1D pseudo-profiles of model
variables are computed, then those pseudo-profiles are assimilated by 3DVar. Both steps are
discussed below.
The first step computes a pseudo-profile of relative humidity weighting the model profiles of relative
humidity around the radar profile (Bayesian approach). The pseudo-profile is computed by:
$$\mathbf{z_o^p} = \frac{\sum_i \mathbf{RH_i} W_i}{\sum_j W_j} \tag{2}$$

Where $RH_i$ is the RAMS@ISAC vertical profile of relative humidity at a grid point inside a square of
$50*50 \text{ km}^2$ centred at the radar vertical profile, $W_i$ is the weight of each profile and $\mathbf{z_o^p}$ is the relative
humidity pseudo-profile. The weights are determined by the agreement between the simulated and
observed reflectivity factor:
$$W_i = \exp\left\{ -\frac{1}{2} \left[ \mathbf{z_o} - h_z(x_i) \right]^T \mathbf{R_z^{-1}} \left[ \mathbf{z_o} - h_z(x_i) \right] \right\} \tag{3}$$

Where $h_z$ is the forward observation operator, transforming the background column $\mathbf{x_i}$ into the
observed reflectivity factor. The forward radar observation operator is taken from the RIP
(Read/Interpolate/Plot) software (https://dtcenter.org/wrf-
nmm/users/OnLineTutorial/NMM/RIP/index.php, last access 03 March 2019) and is given in the
supplemental material of this paper (Section S8). It assumes a Marshall-Palmer hydrometeors size-
distribution, Rayleigh scattering, and depends on the mixing ratios of rain, graupel and snow.
The matrix $\mathbf{R_z}$ in Eqn. (3) is diagonal and its value is $n\sigma^2$, where $\sigma$ is 1 dBz and $n$ is the number of
available observations in the vertical profile (from 1 to 3). In this way, we give more weight to
vertical profiles containing more data.
The error of radar data is assumed small (1dBz) for two reasons: a) reflectivity data are carefully
checked by the Civil Protection Department; b) the performance of control simulation, not
assimilating any data, is rather poor for the case studies. This setting, however, could not be optimal
for cases when the control forecast performs better. A sensitivity test using $\sigma$ =5 dBz for the Livorno
case showed small differences compared to $\sigma$ =1 dBz. The results of this sensitivity test are detailed
in the supplemental material of this paper (Section S4).
It is important to point out that the 50 km length-scale of the above step doesn't represent the
horizontal correlation length-scale of the background error, which determines the horizontal spread
of the innovations in the 3D-Var data assimilation (the latter length-scale is between 14 and 25 km
depending on the level). The 50 km length-scale is used to set a square for computing the pseudo-
profile of relative humidity (Eqn. (2)). This profile is given by a weighted average whose weights are
determined by the agreement between the simulated and observed reflectivity factor. The larger
the agreement the larger the weight. This distance is appropriate because the spatial error of
meteorological models in simulating meteorological features, for example fronts, can be of this
order. The control simulation of the two events considered in this paper confirms this choice.
The method is not able to force convection when the model has no rain, snow or graupel in a square
around (50*50 km$^2$) a radar profile with reflectivity factor greater than zero. In this case, the pseudo-
profile of relative humidity is assumed saturated above the lifting condensation level and with no
data below (Caumont et al., 2010).
It is also noted that the method is able to reduce spurious convection when the reflectivity factor is
simulated but not observed, because the pseudo-profile of relative humidity gives more weight to
the drier relative humidity profiles simulated by RAMS@ISAC inside the 50*50 km$^2$ square centred
at the radar profile. Of course, the ability to reduce spurious convection depends on the availability
of dry model profiles around the specific radar profile (see the example below). Finally, if the
observed profile is dry and the profile simulated by RAMS@ISAC is dry too, the pseudo-profile is not
computed.
In summary, pseudo-profiles are computed for each profile of the radar grid whenever reflectivity
is observed or simulated.
The pseudo-profiles computed with the procedure introduced above, are then used as observations
in the RAMS-3DVar data assimilation (Federico, 2013), minimising the cost-function:

$$J(\mathbf{x}) = \frac{1}{2}(\mathbf{x} - \mathbf{x}_b)^T \mathbf{B}^{-1}(\mathbf{x} - \mathbf{x}_b) + \frac{1}{2}(\mathbf{z}_o^p - h(\mathbf{x}))^T \mathbf{R}^{-1}(\mathbf{z}_o^p - h(\mathbf{x}))$$

442                                                                                          (4)

Where **x** is the state vector giving the analysis when $J$ is minimized, $\mathbf{x}_b$ is the background, **B** and **R** are the background and observations error matrices, $\mathbf{z}_o^p$ is the pseudo vertical profile computed by Eqn. (2) and $h$ is the forward observation operator transforming the state vector (RAMS@ISAC water vapour mixing ratio) into observations. The cost function in RAMS-3DVar is implemented in incremental form (Courtier et al., 1994) and its minimization is performed by the conjugate-gradient method (Press et al., 1992). No multi-scale approach is used.

The background error matrix is divided into three components along the three spatial directions ($x$, $y$, $z$). The $\mathbf{B_x}$ and $\mathbf{B_y}$ matrices account for the spatial correlation of the background error. The correlations are Gaussian with length-scales between 14 and 25 km, depending on the vertical level. These distances are computed using the NMC method (Barker et al., 2012) applied to the HyMeX-SOP1 (Hydrological cycle in the Mediterranean Experiment – First Special Observing Period occurred in the period 6 September-6 November 2012; Ducroq et al., 2014) period. It is again stressed that the spread of the innovations along the horizontal spatial directions in the 3D-Var analysis is determined by the length scales of $\mathbf{B_x}$ and $\mathbf{B_y}$ matrices and varies between 14 and 25 km, depending on the level.

The $\mathbf{B_z}$ matrix contains the error for the water vapour mixing ratio, which is the control variable used in RAMS-3DVar. This error is about 2 g/kg at the surface and decreases with height. In particular, it is larger than 0.5 g/kg below 4 km, and less than 0.2 g/kg above 5 km. The vertical decorrelation of the background error depends on the level and can be roughly estimated in 500-2000 m. The observation error matrix **R** in Eqn. (4) is diagonal and observations' errors are uncorrelated. This choice is partially justified by under sampling the radar reflectivity factor observation by choosing one point every five grid points in both horizontal directions of the radar Cartesian grid. However, correlation observations errors have significant impact on the final analysis, as shown for example in Stewart et al. (2013), and different choices of the matrix **R** will be considered in future studies.

The value of the elements on the diagonal of **R** depends on the vertical level and are 1/4 of the diagonal element of the $\mathbf{B_z}$ matrix at the corresponding height. With these settings, larger weights are given to the observations than to the background and analyses strongly adjust the background towards observations. The background error matrix is computed using the NMC method (Parrish and Derber, 1992; Barker et al. 2004) applied to the HyMeX-SOP1 (Hydrological cycle in the Mediterranean Experiment – First Special Observing Period occurred from 6 September to6 November 2012; Ducroq et al., 2014). This choice is motivated by the fact that HyMeX-SOP1 contains several heavy precipitation events over Italy and the background error matrix is

representative of the convective environment of the cases considered in this paper. In particular, 10 out of 20 declared IOP (Intense Observing Period) of HyMeX-SOP1 occurred in Italy (Ferretti et al., 2014). In contrast, the period of September 2017, especially before the events selected in this study was characterised by fair and stable weather conditions over Italy and the background error matrix for September 2017 is less representative of the convective environment that characterise the events of this paper.

Because it is the first time that we show the assimilation of radar reflectivity factor in RAMS@ISAC, it is useful to discuss an example of analysis. We select the analysis of Livorno case study at 06 UTC. The observed CAPPI at 3km above sea level is shown in Figure 10b. The corresponding CAPPI simulated by the background is shown in Figure 14a. In general, the comparison between simulated and observed reflectivity factor highlights the difficulty of the model to represent convection properly. In particular, the model is able to represent the convection over Northern Italy but it has poor performance over Sardinia, south of Sicily and over Central Italy. The difference between the analysis and background relative humidity after and before the analysis is shown in Figure 14b (absolute values less than 1% are suppressed in the figure for clarity). Both positive (convection enhancing) and negative (convection suppressing) adjustments are found. Over Central Italy, Sardinia and South of Sicily relative humidity is increased because the model doesn't simulate the observed reflectivity (Figure 10b).The occurrence of this condition added most of the water vapour to the RAMS@ISAC simulation for the case studies of this paper.  Over northern Italy the model is partially dried for two different reasons: over northwest of Italy because RAMS@ISAC simulates unobserved reflectivity, over north and northeast of Italy because the model simulates larger values of reflectivity factor compared to the observations. The RAMS-3DVar reduces the relative humidity field north of Corsica island, where the RAMS@ISAC predicts unobserved reflectivity, while RAMS-3DVar didn't suppress the unobserved convection west of Sardinia because the pseudo profiles computed over this area weren't appreciably drier than the background.  Cross correlations among different variables of the data assimilation system are neglected in this study and the application of the RAMS-3DVar affects the water vapour mixing ratio only. Cross correlations among different variables can improve the performance of data assimilation system, and an example of their impact in the RAMS-3DVar is shown in Federico (2013). Nevertheless, the impact of cross correlations among different variables in the precipitation VSF will be explored in future works.

Because also lightning data assimilation adjusts the water vapour mixing ratio, it follows that the data assimilation presented in this study adjusts only this parameter.

Despite the fact that both radar reflectivity factor and lightning adjust the water vapour mixing ratio,
different impacts on the VSF can be expected *a-priori* because radar reflectivity factor and lightning
are different types of observations and because they are used in different ways in the data
assimilation system.
In particular, lightning is recorded when deep convection develops, while radar reflectivity factor is
observed also for light stratiform rain. Flashes of ground based network, as LINET, are available over
the open sea, even if with a reduced detection efficiency, while radar reflectivity factor is confined
to the range of coastal radars in the network. Lightning has a seasonal dependence over Italy, with
the maximum in summer and fall, while radar reflectivity factor is available in all seasons.
Also, differences in data assimilation of lightning and radar reflectivity factor play a role. In addition
to the methods used to assimilate observations, lightning saturates the layer 0°C/-25°C where/when
it is detected, while radar reflectivity factor can be assimilated by pseudo-profiles or by saturation
above the lifting condensation level where observed reflectivity is greater than zero.
So, despite both observations adjust the same model prognostic variable, which is a drawback of
the methodology presented in this paper, the impacts of lighting and radar reflectivity factor is
expected to be different as will be evident from the results of this paper.
There are, however, advantages using the methodology presented in this paper. In addition to being
simple, it doesn't rely on approximate relationship between radar reflectivity factor with
hydrometeors mixing ratio, leaving to the model the task of evolving the water vapour
added/subtracted. Also, the impact of the data assimilation on model results are substantial (Fabry
and Sun, 2010; Caumont et al., 2010), as also shown by the results of this paper.
Lightning and radar data assimilation may produce sharp gradients in vertical direction caused by
the addition of water vapour to specific layers. In the case of lightning, the water vapour is added
by nudging to reduce sharp gradients. However, radar data assimilation, which accounts for the
largest mass of water added to RAMS@ISAC (see Section S.2 of the supplemental material), directly
adjusts the water vapour into the model. Our experience with RAMS@ISAC, however, shows that
results are reliable and the sudden addition of water vapour doesn't cause shocks to the model
simulation, despite the notable gradients of specific humidity.
It is finally noted that the data assimilation increase/decrease the water vapour into the model
depending on the cases. The eventual increase/decrease of the forecasted rainfall depends on the
physical and dynamical processes occurring into the meteorological model, without any specific
tuning.

## 4. Results

In this section, we discuss the most intense phase of the Serano case, 03-06 UTC on 16 September, and two VSF forecasts, 00-03 UTC and 06-09 UTC on 10 September, for the Livorno case. The two VSF for Livorno correspond to the most intense phase of the storm in Livorno and to a very intense phase over Lazio region, Central Italy. The aim of the section is to show the notable improvement given by lightning and radar reflectivity factor data assimilation to the VSF.

We consider four types of VSF (Table 3): a) CTRL, without radar reflectivity factor and lightning data assimilation; b) LIGHT, assimilating lightning but not radar reflectivity factor; c) RAD, assimilating radar reflectivity factor but not lightning; d) RADLI, assimilating both lightning and radar reflectivity factor.

Several aspects of lightning and radar reflectivity factor data assimilation are considered in the supplemental material of this paper: a) the relative contribution to the total water mass given by lightning and radar reflectivity factor data assimilation (Section S.2); b) the sensitivity of the precipitation VSF to the nudging formulation (Section S.3); c) the sensitivity of rainfall VSF to two specific aspects of radar reflectivity factor data assimilation (Section S4); d) the sensitivity of rainfall VSF to RAMS@ISAC setting (Section S5); e) the impact of lightning data assimilation for a case study well predicted by the control forecast (Section S6); f) different plots of Figures 15-17 (Section S7) and g) the forward radar operator used in RAMS-3DVar (Section S8).

*4.1 Serano: 03-06 UTC on 16 September 2017*

In this period, an intense and localised storm hit central Italy, while light precipitation occurred over northern Italy (Figure 15a). Considering the storm over central Italy, 10 raingauges observed more than 30 mm/3h, 6 more than 40 mm/3h, 3 more than 50 mm/3h and 1 more than 60 mm/3h, the maximum observed value being 63 mm/3h.

The CTRL forecast, Figure 15b, misses the storm over central Italy and considerably underestimates the precipitation area over Northern Italy, giving unsatisfactory results.

The assimilation of the radar reflectivity factor improves the forecast, as shown in Figure 15c. In particular, RAD forecast shows localized precipitation (30-35 mm/3h) close to the area were the most abundant precipitation was observed. Maximum precipitation is underestimated. Also, the RAD forecast better represents the precipitation over Northern Italy compared to CTRL.

The rainfall forecast of LIGHT, Figure 15d, shows some improvements compared to CTRL because the precipitation over central Italy has a maximum of 25-30 mm/3h, close to the area where the maximum precipitation was observed. LIGHT, however, has a worse performance compared to RAD because it underestimated the precipitation area over northern Italy. LIGHT underestimates the maximum precipitation in central Italy.

RADLI forecast, Figure 15e, has the best performance. The precipitation over central Italy is well represented because the maximum rainfall (40-45 mm/3h) is in reasonable agreement with observations, and also because the area of intense precipitation (> 25 mm/3h) is elongated in the SW-NE direction in agreement with raingauge observations. The precipitation over northern Italy is well represented by RADLI.

Performance diagram for 1 mm/3h and 30 mm/3h and for 4 km and 25 km neighbourhood radii is shown in Figure 15f. Different radii are considered to account for the well-known double penalty error (Mass et al., 2002; Mittermaier et al., 2013) caused by displacement errors of the detailed precipitation forecast in convection allowing grids. RADLI has the best performance thanks to the synergistic contribution of lightning and radar reflectivity factor data assimilation.

*4.2 Livorno*

The Livorno case study lasted for several hours starting at 18 UTC on 9 September 2017 and ending more than a day later. The most intense phase in Livorno and its surroundings was observed during the night between 9 and 10 September. In the following, we will show two representative VSF (3h), including the most intense phase in Livorno.

*4.2.1 Livorno: 00-03 UTC on 10 September 2017*

This period represents the most intense phase of the storm in Livorno. In particular, the raingauge close to the label A (Figure 16a) reported 151 mm/3h (Collesalvetti), while the one close to the label B measured 82 mm/3h. Among the 518 raingauges reporting valid data, 75 observed more than 10 mm/3h, 31 more than 20 mm/3h, 17 more than 30 mm/3h, 9 more than 40 mm/3h, and 6 more than 50 mm/3h.

The CTRL precipitation forecast is shown in Figure 16b. The forecast is poor because it misses the precipitation swath from the coast towards NE. A precipitation swath is forecasted about 50 km to the North of the real occurrence, but it is less wide compared to the observations.

The RAD forecast, Figure 16c, shows that the assimilation of radar reflectivity factor gives a clear improvement to the forecast. The largest precipitation in the coastal part of the swath (we searched for the maximum in the area with longitudes between 10.20E and 10.70E and latitudes between 43.10N and 43.60N) is 94 mm/3h. Another local maximum is in the southern part of the domain (label B of Figure 16a). The maximum location is well represented, but the forecast value (55 mm/3h) underestimates the observed maximum (82 mm/3h).

An improvement, compared to both CTRL and RAD, is given by the assimilation of lightning (Figure 16d). The maximum value close to Livorno, i.e. in the coastal part of the swath, is 158 mm/3h. LIGHT simulation shows the local maximum in the southern part of the domain (about 50 mm/3h), but the amount is underestimated.

Figure 16e shows the RADLI rainfall forecast. The precipitation swath from coastal Tuscany towards NE is more intense compared to LIGHT and RAD. The maximum rainfall accumulated close to Livorno is 186 mm/3h. Also, the second precipitation maximum in the southern part of the domain reaches 70 mm/3h in good agreement with observations (82 mm/3h). RADLI is the only run giving a satisfactory precipitation VSF over the south-eastern Emilia Romagna (north-eastern part of the domain), to the lee of the Apennines. It is also noted that the main precipitation swath forecasted by RADLI is too broad in the direction crossing the swath compared to the observations. This is confirmed by the FBIAS of RADLI (not shown), which is more than 3 for thresholds larger than 42 mm/3h.

The performance diagram (Figure 16f) shows that LIGHT has better scores than RAD for this VSF.

*4.2.2 Livorno: 06-09 UTC on 10 September 2017*

In this period, the most intense precipitation occurred over the coastal part of Lazio (Figure 17a). More in detail, among the 2695 raingauges reporting valid data over the domain of Figure 17a, 307 reported more than 10 mm/3h, 132 more than 20 mm/3h, 86 more than 30 mm/3h, 66 more than 40 mm/3h, 49 more than 50 mm/3h and 35 more than 60 mm/3h. Among the 35 raingauges measuring more than 60 mm/3h, 33 were over Lazio, showing the heavy rainfall occurred over the Region.

Some precipitation persisted over Tuscany but the rainfall is much lower compared to previous 6h (the rainfall over Tuscany between 03 and 06 UTC was very intense, not shown).

Figure 17b shows the rainfall simulated by CTRL. The forecast is unsatisfactory, mainly for the following two reasons: a) heavy precipitation is simulated over Tuscany (> 75 mm/3h), also close to

the Livorno area; b) precipitation is missed over central Italy. The rainfall over NE of Italy is well
represented in space, but overestimated.
Considering the evolution of CTRL forecast for the two VSF of Livorno, we conclude that it was able
to predict abundant rain over Livorno, but the rainfall forecast was delayed compared to the real
occurrence. A similar behaviour was found in Ricciardelli et al. (2018) using the WRF model, showing
that the results of this paper for Livorno are likely not tied to the specific model used.
The rainfall simulated by RAD (Figure 17c) clearly improves the forecast compared to CTRL. First,
the precipitation over Lazio is well predicted. Second, the precipitation over Tuscany is less than for
CTRL, showing the ability of radar reflectivity factor data assimilation to dry the model when it
predicts reflectivity that is not observed. This is confirmed by the inspection of the analysis of Figure
14b, the last analysis used before this VSF, which gives a decrease of the relative humidity over most
of Tuscany and over the sea in front of Livorno. It is noted, however, that the area of intense rainfall
(>60 mm/3h) is overestimated by RAD, showing a wet forecast. The wet bias of the RAD forecast is
apparent in the representation of the rainfall VSF shown in the supplemental material of this paper
(Figure S12).
LIGHT forecast, Figure 17d, shows a worse performance compared to RAD for this time period. The
precipitation forecast is mainly over Tuscany, where it is overestimated, with a small precipitation
spot over Lazio.
The precipitation forecast of RADLI, Figure 17e, represents very well the precipitation over Lazio,
and the rainfall amount is better predicted compared to RAD. The precipitation over Sardinia is well
represented by RADLI as well as the precipitation over Central Alps, giving the best results among
all VSF.
Figure 17f shows the better performance of RAD compared to LIGHT for this precipitation VSF. RADLI
has the best performance being closer to the upper right corner of the diagram.
To better understand the changes of the precipitation VSF to different data assimilation set-up,
Figure 18 shows maps of water vapour mixing ratio averaged between 3 and 10 km at the end of
the assimilation phase (06 UTC on 10 September 2017). It is important to note that those maps
contain the effects of both data assimilation and model evolution.
The comparison between CTRL (Figure 18a) and RAD (Figure 18b) shows that RAD has a line of high
water vapour values over Central Italy, extending over the Tyrrhenian Sea and Sardinia, which is not
simulated by CTRL. This line results from both radar data assimilation and convection, which
transports water vapour from lower to upper levels. The comparison between CTRL and RAD shows
the substantial impact of radar reflectivity factor data assimilation on the model evolution despite
we are not using relationship between hydrometeors mixing ratios and radar reflectivity factor in
data assimilation.
LIGHT averaged water vapour (Figure 18c) over the Tyrrhenian Sea and west of Sicily is higher
compared to CTRL because of lightning data assimilation and model processes. Convection develops
over Tuscany, northern Lazio and NE of Italy, causing the increase of averaged water vapour in those
areas.
Because RAD and LIGHT both assimilate water vapour it is important to highlight the differences
between the two fields. First, LIGHT it is not able to represent a compact line of high water vapour
over Central Italy that, in the following hours, caused high precipitation over Lazio. Second,
averaged water vapour simulated by RAD is larger than for LIGHT over Central Italy, which is caused
by a deeper convection developing in RAD than in LIGHT, as well as by the different contributions
of data assimilation.Finally, RADLI (Figure 18d) is similar to RAD but it shares also features with
LIGHT as the increase of water vapour over the Tyrrhenian Sea.
It is also interesting to compare vertical cross sections of relative humidity for different data
assimilation set-up. Figure 19 show the longitude-height cross sections of relative humidity from
different data assimilation configurations.
Comparing RAD with CTRL it is evident the difference of the relative humidity field over the
Tyrrhenian Sea and western part of Italy (more specifically at longitudes between 10.5 and 12.5).
LIGHT shows two areas with high relative humidity: west of Corsica and over the Tyrrhenian Sea.
The wet area west of Corsica is caused by the assimilation of lighting (Figure 8b) and it is not
simulated by RAD because Corsica is not well sampled by the radar network and because of different
model evolutions. Lightning data assimilation also increases the humidity over the Tyrrhenian Sea
and on the western part of Italy, as shown by the comparison with CTRL, nevertheless their effect is
lower compared to radar reflectivity data assimilation.
RADLI has features of both lightning and radar reflectivity factor data assimilation.
So, considering the results of Figure 18 and 19 as well as the rainfall VSF, the impact of lightning and
radar reflectivity factor on the VSF can be very different despite they both adjust the water vapour
mixing ratio.

**5. Discussion and Conclusions**
In this paper, we showed the impact of lightning and radar reflectivity factor data assimilation on
the very short term precipitation forecast (3h) for two case studies occurred in Italy. We used
RAMS@ISAC model, whose 3DVar extension to the assimilation of radar reflectivity factor is shown
in this paper for the first time.
The first case study occurred on 16 September 2017 and it is a moderate case with localised rainfall
over central Italy. It was chosen because the control forecast, i.e. without radar reflectivity factor
or lightning data assimilation, missed the event. The second event, occurred on 9-10 September
2017, was characterised by exceptional rainfall over several parts of Italy. This event was partially
represented by the control forecast. In particular, the forecast of the event was incorrect because:
a) the control forecast was delayed compared to the observations; b) the control forecast missed
the rainfall over central Italy (Lazio Region).
It is important to recall that the impact of the lightning data assimilation on the precipitation
forecast of RAMS@ISAC was already studied for the HyMeX-SOP1 period (Federico et al., 2017a,
2017b), and a robust statistic is already available. The results of this study confirm the important
role of the lightning data assimilation on the rainfall forecast for other two case studies. However,
considering the assimilation of radar reflectivity factor, and its combination with lightning data
assimilation in RAMS@ISAC, the results of this paper are new.
Because we analysed only two case studies, no definitive conclusions can be derived on the
performance of RAMS@ISAC for radar reflectivity factor data assimilation. There are, however, few
points worth of mention.
The VSF performance of RAMS@ISAC is systematically improved by the assimilation of radar
reflectivity factor. This improvement is of paramount importance for some specific VSF (for example
for the 00-03 UTC of Livorno), when the control forecast missed the event while it was correctly
predicted by radar reflectivity factor data assimilation. Sometimes the improvement of reflectivity
factor data assimilation has less impact on the precipitation forecast, as for the period 18-21 UTC
on 9 September 2017 (Livorno, not shown, see the discussion paper Federico et al. (2018) for a
description of this VSF). This suggests that there is room for improvement for all components of the
VSF: observations, data assimilation, meteorological model.
Lightning and radar observations are different and both add value to the VSF. Some examples have
been shown: the light precipitation over Northern Italy for Serano is well forecasted assimilating
radar reflectivity factor, while it is not simulated assimilating flashes because they are too few in
this area to force convection; lightning data assimilation is able to better represent the deep

convection occurring during the intense phase of the Livorno case (00-03 UTC), especially because it is able to force convection where it occurs, reducing false alarms. The ability of lightning data assimilation to reduce false alarms compared to RAD and RADLI it is shown by the fact that the ETS score for LIGHT is sometimes the best among all simulations (see also the Section S2 of the supplemental material of this paper). These results show also that the influence of different observations depends on the meteorological situation.

The model configuration assimilating both radar reflectivity factor and lightning (RADLI) is able to retain important features of both data assimilation. For example, the simulation of the Livorno case in the phase 06-09 UTC was able to simulate the heavy precipitation over Lazio thanks to the radar reflectivity factor data assimilation and the precipitation over Sardinia, as well as the moderate precipitation over central Alps, thanks to lightning data assimilation.

The property of RADLI to retain the precipitation features of both RAD and LIGHT it is shown by the POD score, which is the best, for most cases and thresholds, for RADLI.

Another interesting feature is the considerable improvement of the POD of RADLI compared to CTRL for the lowest thresholds.

It is also underlined that the data assimilated, both lightning and radar reflectivity factor, are available in real time and could be used for an operational implementation of the VSF.

It is worth noting that several sensitivity tests were conducted for the case studies, whose results are shown in the supplemental material. In particular, we studied the sensitivity of the rainfall VSF to: a) nudging formulation used for lightning data assimilation; b) increasing the observation error of radar reflectivity factor; c) changing the shape of the searching area to compute the relative humidity pseudo-profile; d) updating IC/BC as new observations are available; e) increasing the vertical resolution of RAMS@ISAC by using 42 vertical levels. All these sensitivity tests confirm the findings of this paper and generalise in some measure the finding of this paper.

The above results are promising and deserve future studies to better understand the role of radar reflectivity factor data assimilation and its interaction with lightning data assimilation to improve the precipitation forecast, especially at the very short range (0-3 h).

There are, however, less satisfactory aspects of assimilating both radar reflectivity factor and lightning data. In particular, the wet bias of RAD and RADLI forecast is the main drawback of the results of this paper. To reduce the moisture added by radar and lightning data assimilation further research is needed and different approaches are possible (Fierro et al., 2016). In particular: a) assimilating for a shorter time (0-6h in this paper); b) reducing the length-scales of the 3D-Var in the

horizontal directions to limit the spreading of the innovations, or assuming an innovation equal to zero for grid points without lightning and with zero reflectivity factor; c) reducing the amount of water vapour added to the model (for example reducing the values of A and B constants for lightning data assimilation or relaxing the request of saturation when radar reflectivity is observed in areas where the model has zero reflectivity); d) adding moisture to a shallower vertical layer.

It is also noted that a combination of heating and moistening could provide the same buoyancy with less water vapour addition (Marchand and Fulberg, 2014) and this approach could be used in future studies.

In addition to the acquisition of more case studies, there are two directions of future development of this work. The lightning data assimilation can be formulated by 3DVar, using a strategy similar to the radar reflectivity factor in which pseudo-profiles of relative humidity are first generated where flashes are recorded, and then those profiles are assimilated by 3DVar. This methodology was already reported in Fierro et al. (2016). The assimilation of both radar reflectivity factor and lightning using RAMS-3DVar will be explored in future studies.

Another important point to study is how long the innovations introduced by data assimilation lasts in the forecast. While in this study we consider the VSF at 3h, future studies must explore longer time ranges. This kind of study was performed for lightning data assimilation (Fierro et al., (2015); Federico et al., 2017b; Lynn et al. (2015) among others) and for radar data assimilation (Hu et al. (2006); Jones et al. (2014), among others), using a rationale similar to that used in this paper.

In general, the performance of the forecast and the impact of lightning and radar data assimilation decrease with forecasting time because boundary conditions propagate inside the domain and because model errors grow and eventually become dominant. Improving the data assimilation system also contributes to a longer resilience of model performance. The studies cited above showed that lightning and radar data assimilation can have an impact up to 24h depending on several factors (meteorological model, data assimilation, quality of the data, meteorological conditions, initial and boundary conditions).

A study considering both radar reflectivity factor and lightning should be performed to understand the resilience of the innovations introduced by data assimilation.

**ACKNOWLEDGMENTS**

This work is a contribution to the HyMeX program. Part of the computational time used for this paper was granted by the ECMWF (European Centre for Medium range Weather Forecast)

throughout the special project SPITFEDE. LINET data were provided by Nowcast GmbH
(https://www.nowcast.de/) within a scientific agreement between H.D. Betz and the Satellite
Meteorological Group of CNR-ISAC in Rome.
This work was partially funded by the agreement between CNR-ISAC and the Italian Department of
Civil Protection.

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

**TABLES**
Table 1: RAMS@ISAC physical parameterisations used in this paper.

| Physical parameterization | Selected scheme |
|---|---|
| Parametrized cumulus convection | Modified Kuo scheme to account for updraft and downdraft (Molinari and Corsetti, 1985). The scheme is applied to R10 only. |
| Explicit precipitation parameterization | Bulk microphysics with six hydrometeors (cloud, rain, graupel, snow, ice, water vapour). Described in Hong and Lim (2006). |
| Exchange between the surface, the biosphere and atmosphere. | LEAF3 (Walko et al., 2000). LEAF includes prognostic equations for soil temperature and moisture for multiple layers, vegetation temperature and surface water, and temperature and water vapour mixing ratio of canopy air. |
| Sub-grid mixing | The turbulent mixing in the horizontal directions is parameterised following Smagorinsky (1963), vertical diffusion is parameterised according to the Mellor and Yamada (1982) scheme, which employs a prognostic turbulent kinetic energy. |
| Radiation scheme | Chen-Cotton (Chen and Cotton, 1983). The scheme accounts for condensate in the atmosphere. |


Table 2: Basic parameters of the RAMS@ISAC grids (R10, R4 and R1, corresponding, respectively, to the domains D1, D2
and D3). NNXP is the number of grid points in the WE direction, NNYP is the number of grid-points in the NS direction,
NNZP is the number of vertical levels, DX is the size of the grid spacing in the WE direction, DY is the grid spacing in the
SN direction. Lx, Ly, and Lz are the domain extensions in the NS, WE, and vertical directions. CENTLON and CENTLAT are
the coordinates of the grid centres.

| | R10, D1 | R4, D2 | R1, D3 |
|---|---|---|---|
| NNXP | 301 | 401 | 203 |
| NNYP | 301 | 401 | 203 |

| NNZP | 36 | 36 | 36 |
|---|---|---|---|
| Lx | 3000 km | 1600 km | ~270 km |
| Ly | 3000 km | 1600 km | ~270 km |
| Lz | ~22400 m | ~22400 m | ~22400 m |
| DX | 10 km | 4 km | 4/3 km |
| DY | 10 km | 4 km | 4/3 km |
| CENTLAT (°) | 43.0 N | 43.0 N | 43.7 N |
| CENTLON (°) | 12.5 E | 12.5 E | 11.0 E |

Table 3: Types of simulations performed.

| Experiment | Description | Data assimilated | Model variable impacted |
|---|---|---|---|
| CTRL | Control run | None | None |
| RAD | RADAR data assimilation | Reflectivity factor CAPPI (RAMS-3DVar) | Water vapour mixing ratio |
| LIGHT | Lightning data assimilation (A=0.85; B=0.16 in Eqn. (1)) | Lightning density (nudging) | Water vapour mixing ratio |
| RADLI | RADAR + lightning data assimilation (A=0.86; B=0.15 in Eqn (1)) | Reflectivity factor CAPPI (RAMS-3DVar) + Lightning density (nudging) | Water vapour mixing ratio |




**FIGURES**


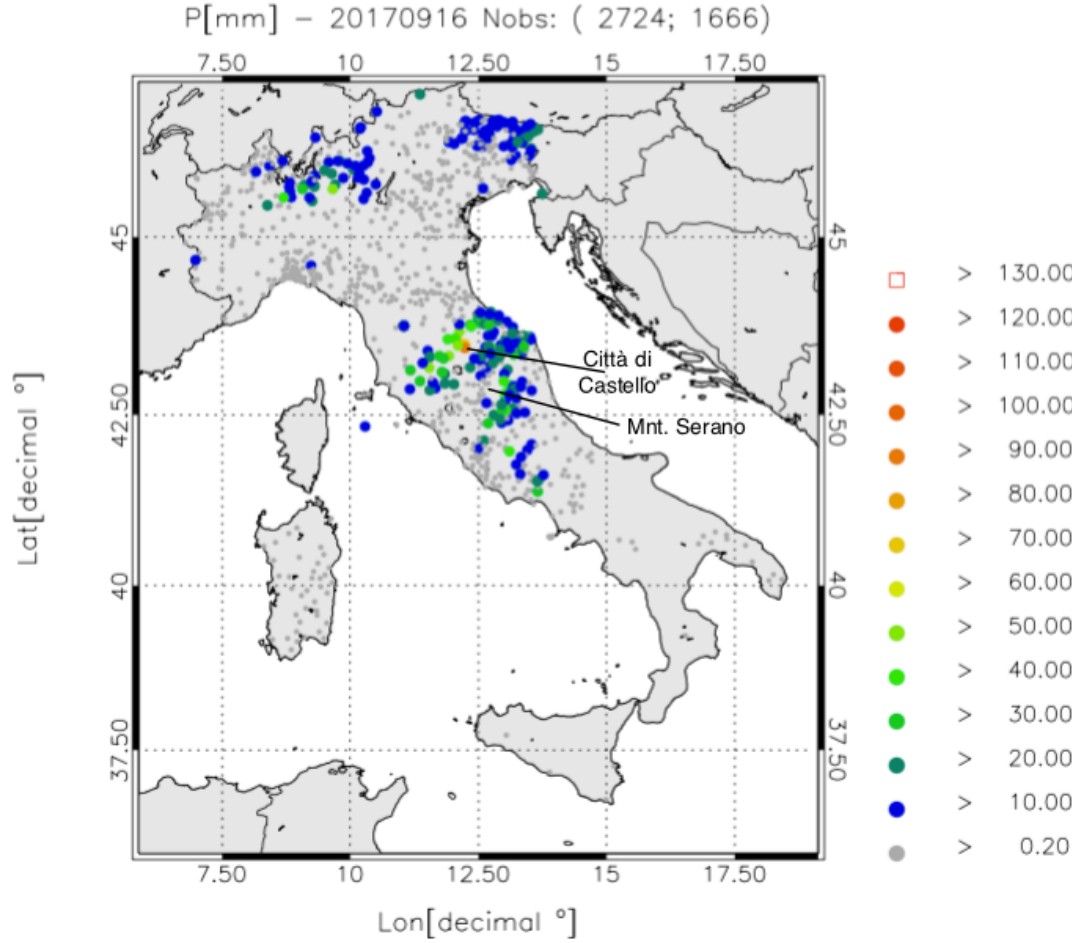


Figure 1: Daily precipitation (P) [mm] over Italy on 16 September 2017. Only raingauges observing at least 0.2 mm/day
are shown. The first number in the figure title within brackets represents the available raingauges, while the second
number represents raingauges observing at least 0.2 mm/day. The lowest precipitation class is represented by smaller
dots, the largest by a red square. The locations of Città di Castello and Mount Serano are indicated.












1093          a)

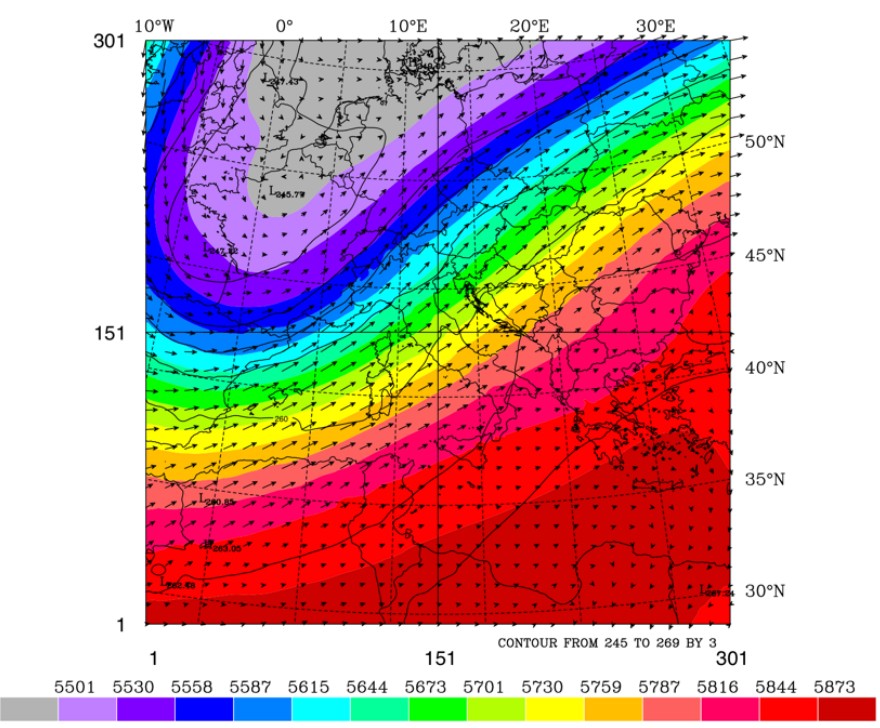

HGT[m] - WSP[m/s] - 20170916000000 - z=   500 hPa


b)

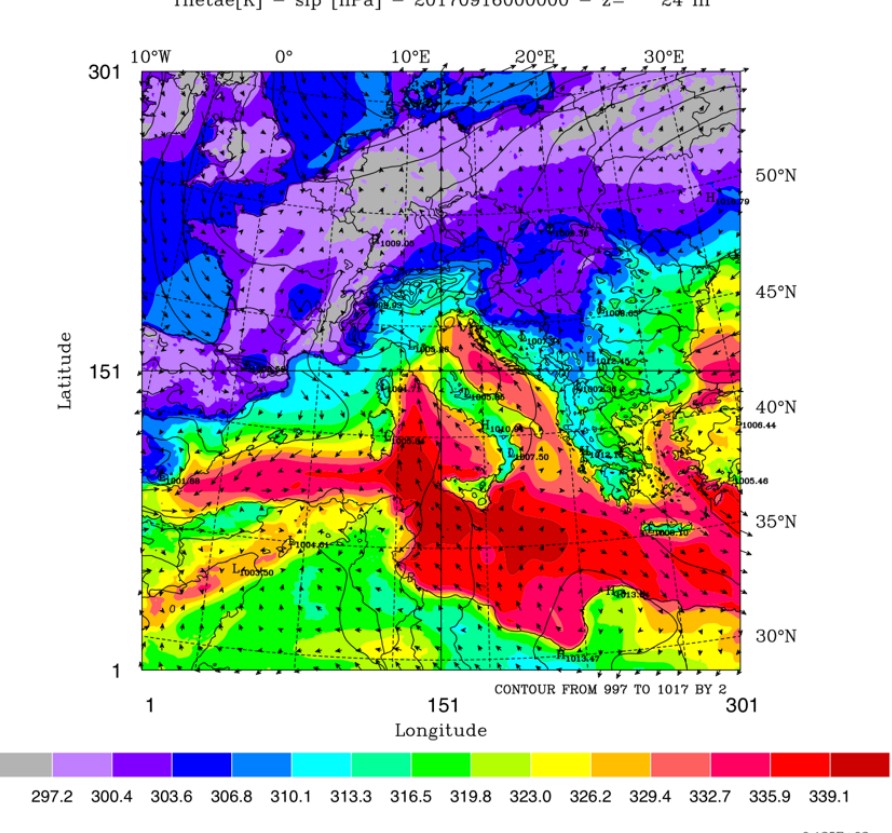

Thetae[K] − slp [hPa] − 20170916000000 − z=   24 m

CONTOUR FROM 997 TO 1017 BY 2

297.2  300.4  303.6  306.8  310.1  313.3  316.5  319.8  323.0  326.2  329.4  332.7  335.9  339.1

0.125E+02


Figure 2: a) Geopotential height (filled contours), temperature (contours) and wind vectors at 500 hPa on 16 September
2017 at 00 UTC. Maximum velocity is 31 m/s; b) equivalent potential temperature (filled contours), sea-level pressure
(contours) and wind vectors at 24 m above the surface (maximum value 13 m/s). A low-pressure patter is forming over
northern Italy, with a front in the western Mediterranean.

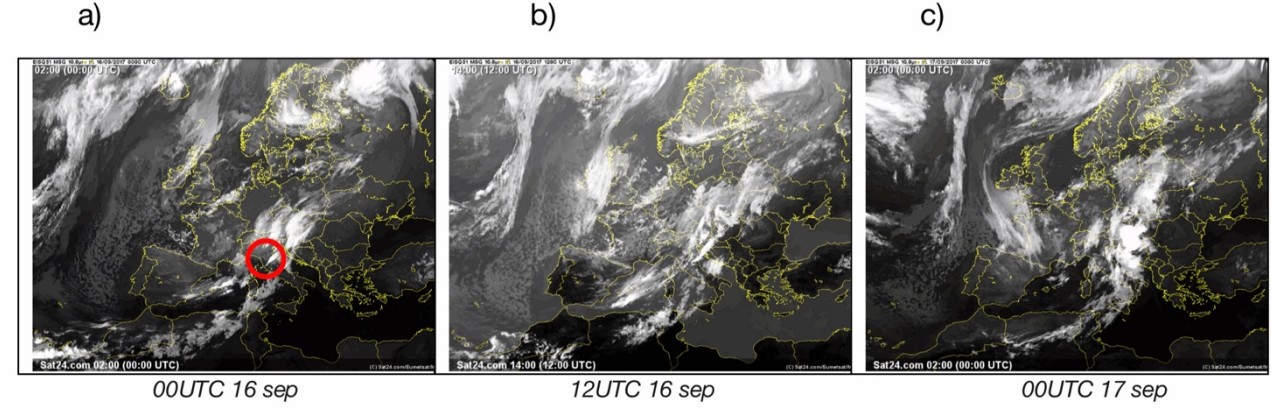

a)                              b)                              c)

00UTC 16 sep              12UTC 16 sep              00UTC 17 sep


Figure 3: a) Satellite images (METEOSAT second generation) of the infrared channel, 10.8 micron, at 00 UTC and 12 UTC
on 16 September, and at 00 UTC on 17 September 2017. A well-defined cloud system is apparent inside the red circle
of the image at 00 UTC on 16 September 2017.


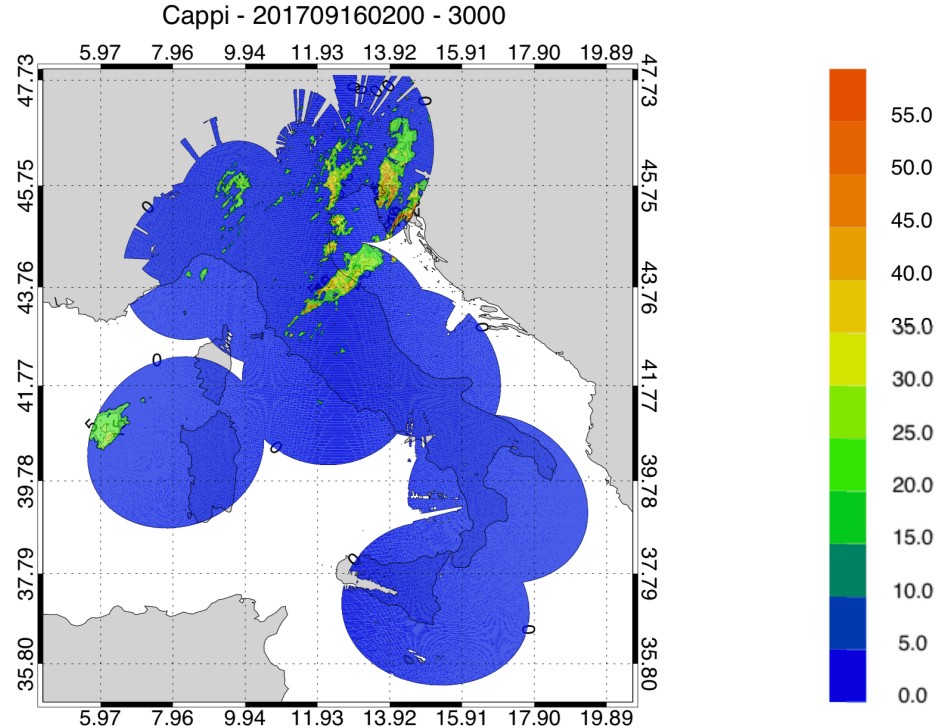

Figure 4: National radar mosaic at 3 km above the sea level observed at 02 UTC on 16 September 2017.

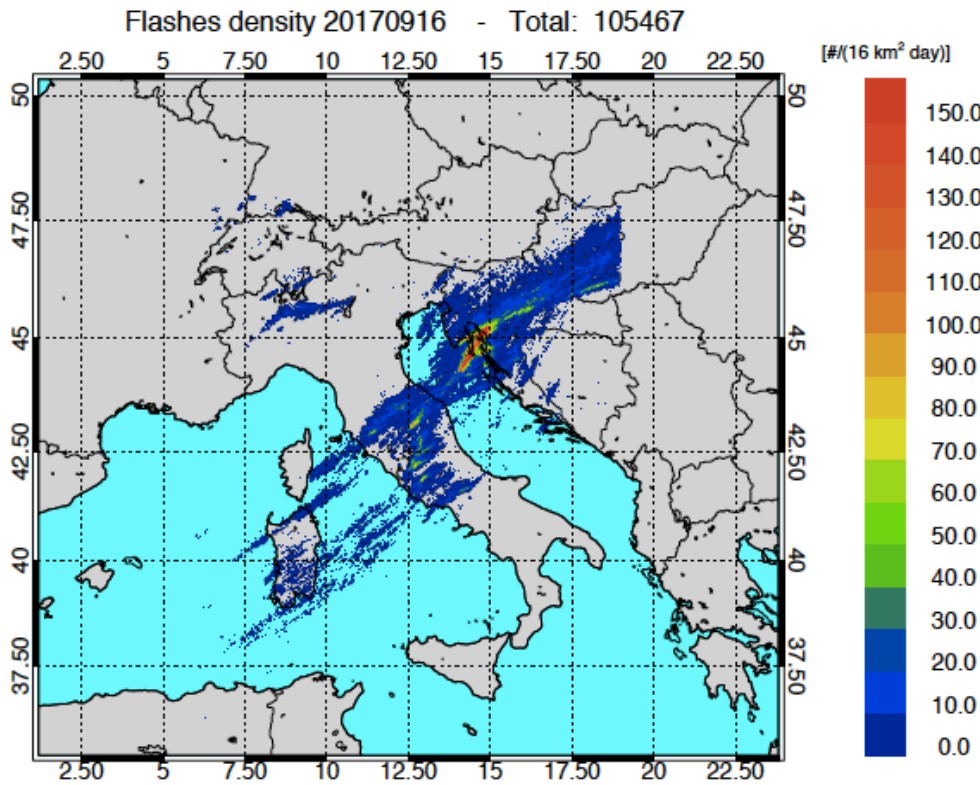


Figure 5: Lightning density (number of lighting per 16 km$^2$ for the whole day) recorded on 16 September 2017. The total
number of flashes is shown in the title.

1115        a)

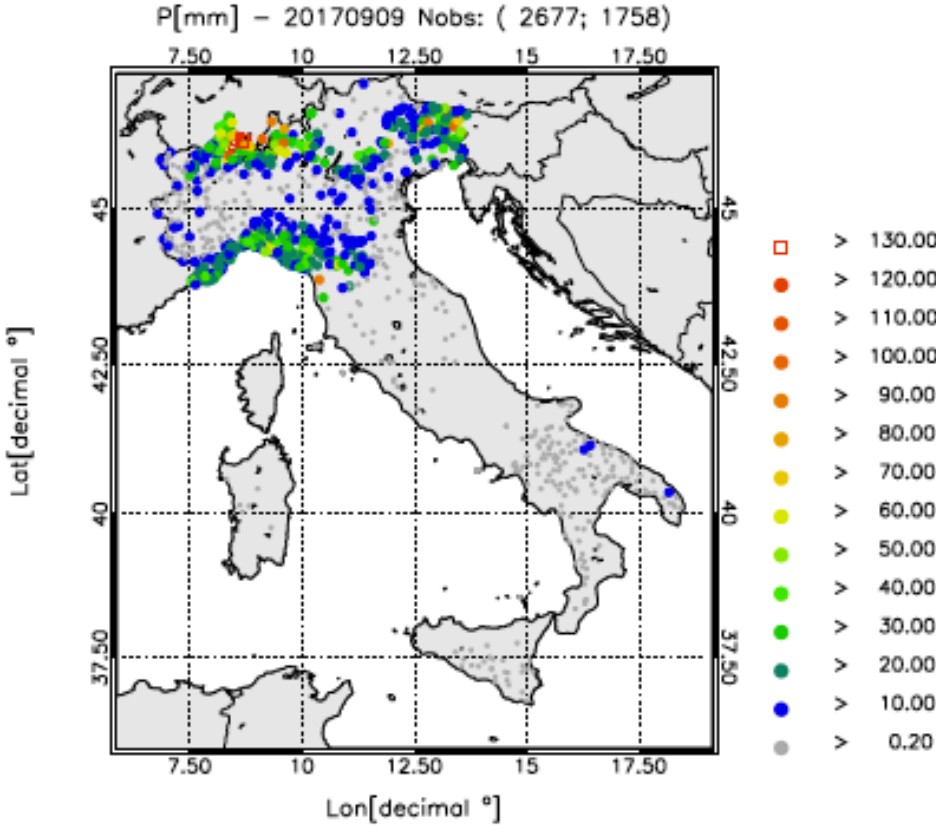


b)

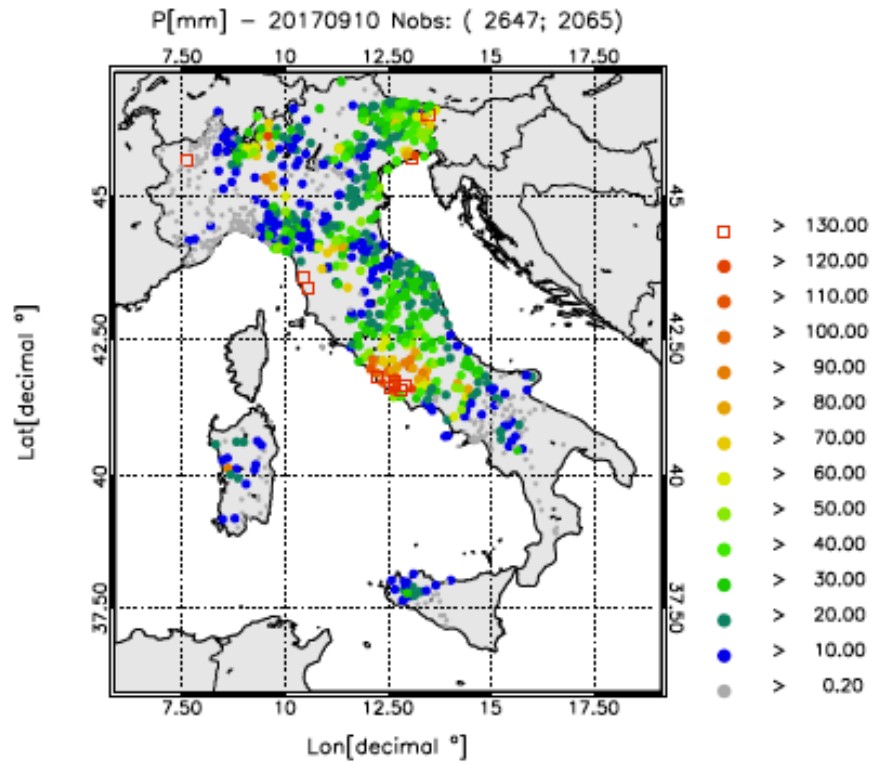


Figure 6: a) As in Figure 1 but for a) 9 September 2017 and b) 10 September 2017.

a)

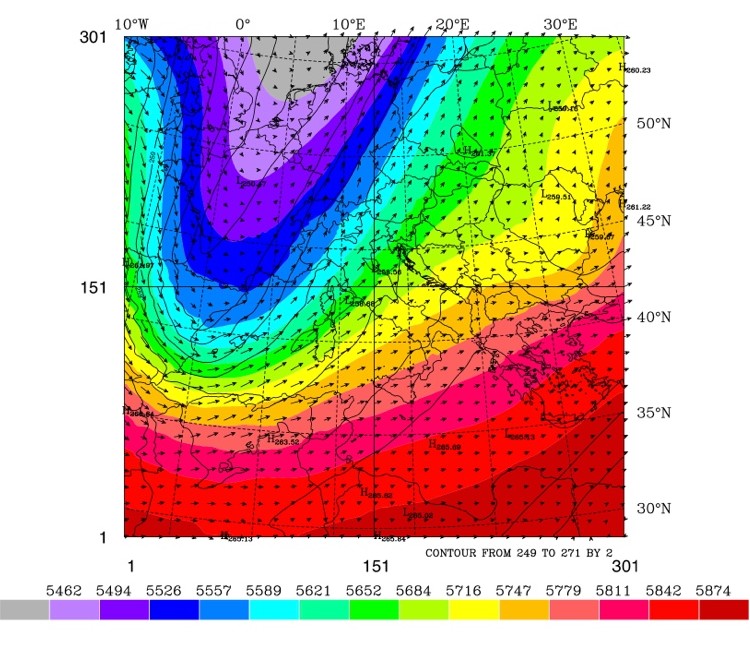


b)

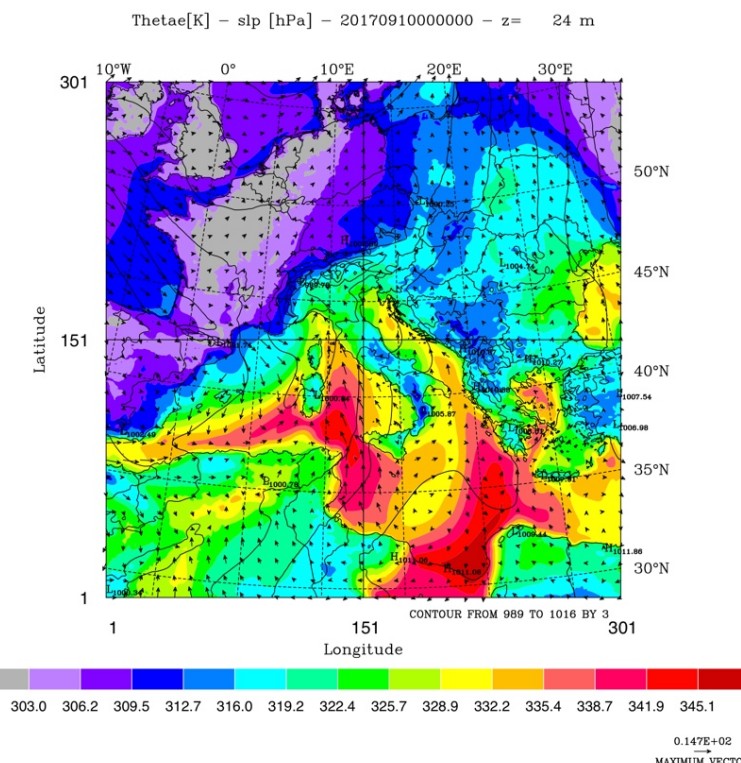


Figure 7: a) Geopotential height (filled contours), temperature (contours) and wind vectors at 500 hPa at 00 UTC on 10
September 2017. Maximum velocity is 37 m/s; b) equivalent potential temperature (filled contours), sea-level pressure
(contours) and wind vectors at 24 m above the surface (maximum value 15 m/s).

a)

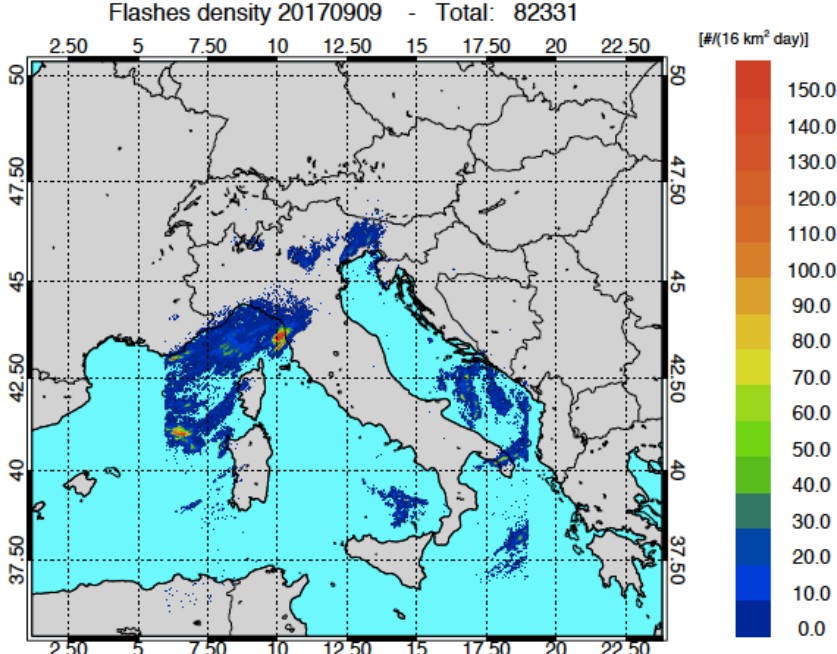




b)

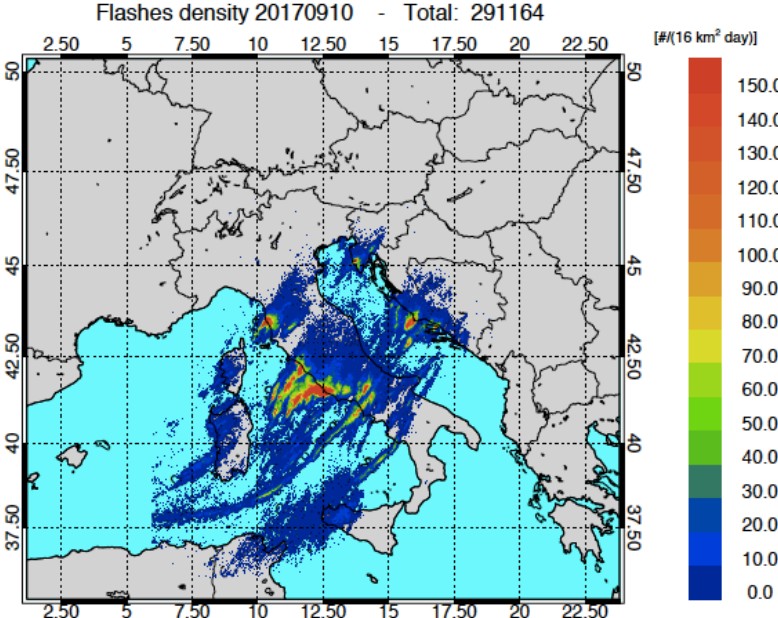


Figure 8: a) Lightning density (lightning number per 16 km$^2$ for the whole day) recorded on 09 September 2017; b) as in
a) on 10 September 2017. The number of flashes on each day is shown in the title.



a)                      b)                    c)

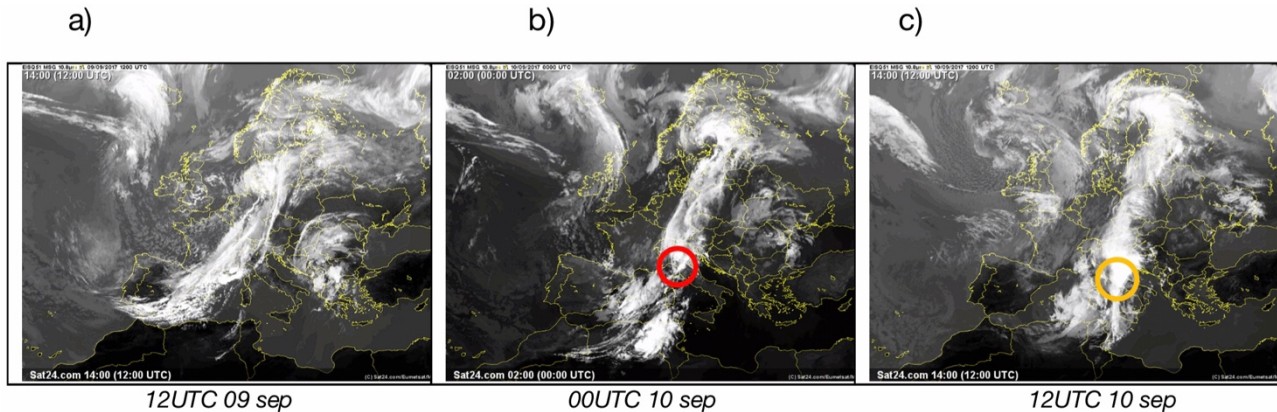

*12UTC 09 sep*             *00UTC 10 sep*             *12UTC 10 sep*

Figure 9: a) Satellite images (METEOSAT second generation) of the infrared channel, 10.8 micron, at 12 UTC on 9 September 2017, at 00 UTC and 12 UTC on 10 September 2017. The red circle in Figure 9b and the orange circle in Figure 9c show the Livorno and Lazio area, respectively.

a)

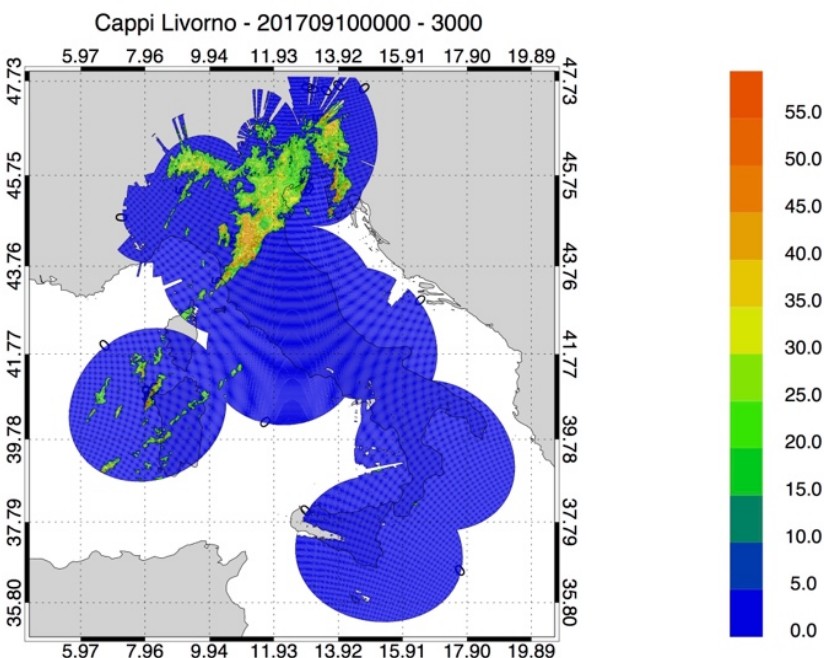

b)

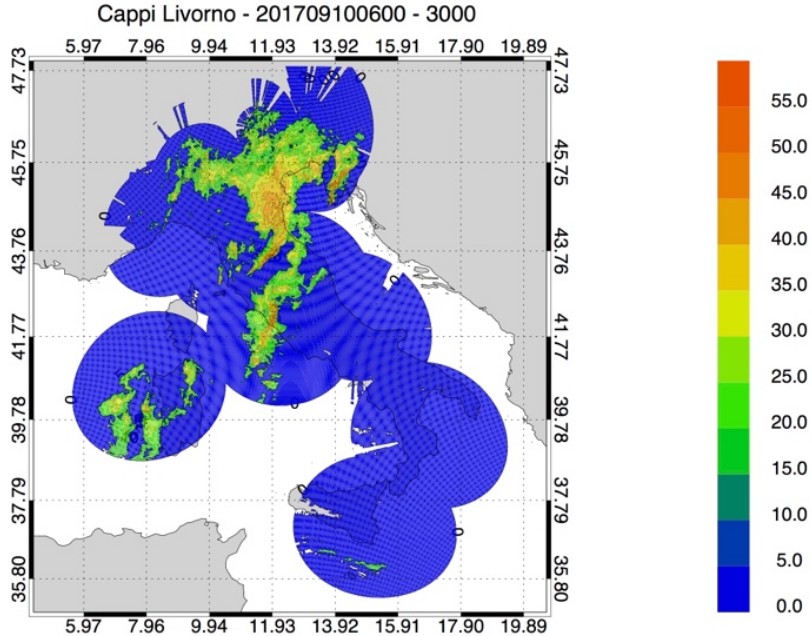


Figure 10: a) National radar mosaic at 3 km above the sea level observed at 00 UTC on 10 September 2017; b) as in a)
at 06 UTC.

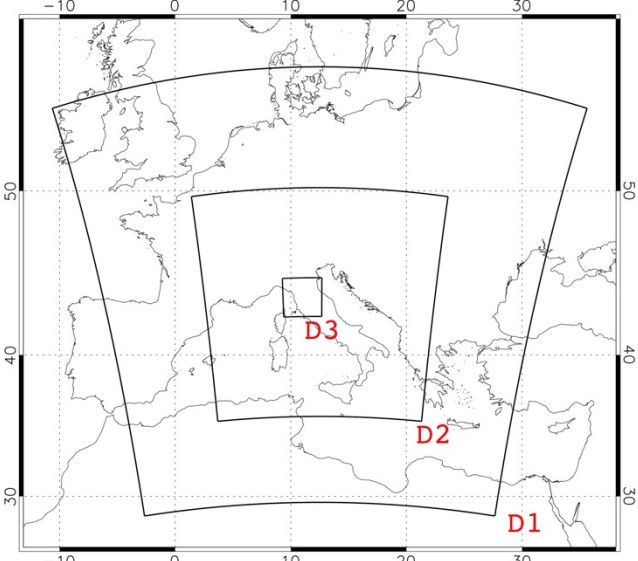


Figure 11: The three domains used in RAMS@ISAC. The model grid over domain D1 has 301 grid points in the NS and
WE directions and has 10 km horizontal resolution, the model grid over domain D2 has 401 grid points in the NS and
WE directions and has 4 km horizontal resolution. The model grid over domain D3 has 203 grid points in the NS and WE
directions and has 4/3 km horizontal resolution. All grids have the same thirty-six vertical levels spanning the 0-22.4 km
vertical layer.

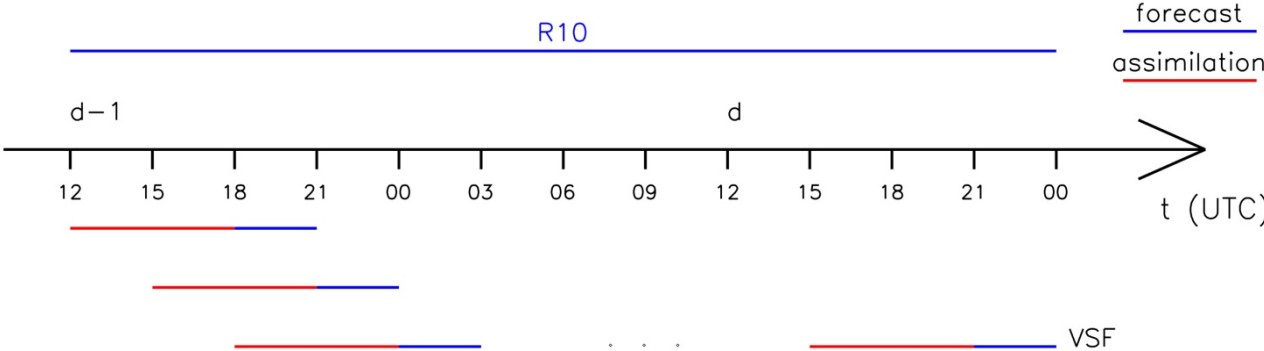


Figure 12: The implementation of RAMS@ISAC very short-term forecast.

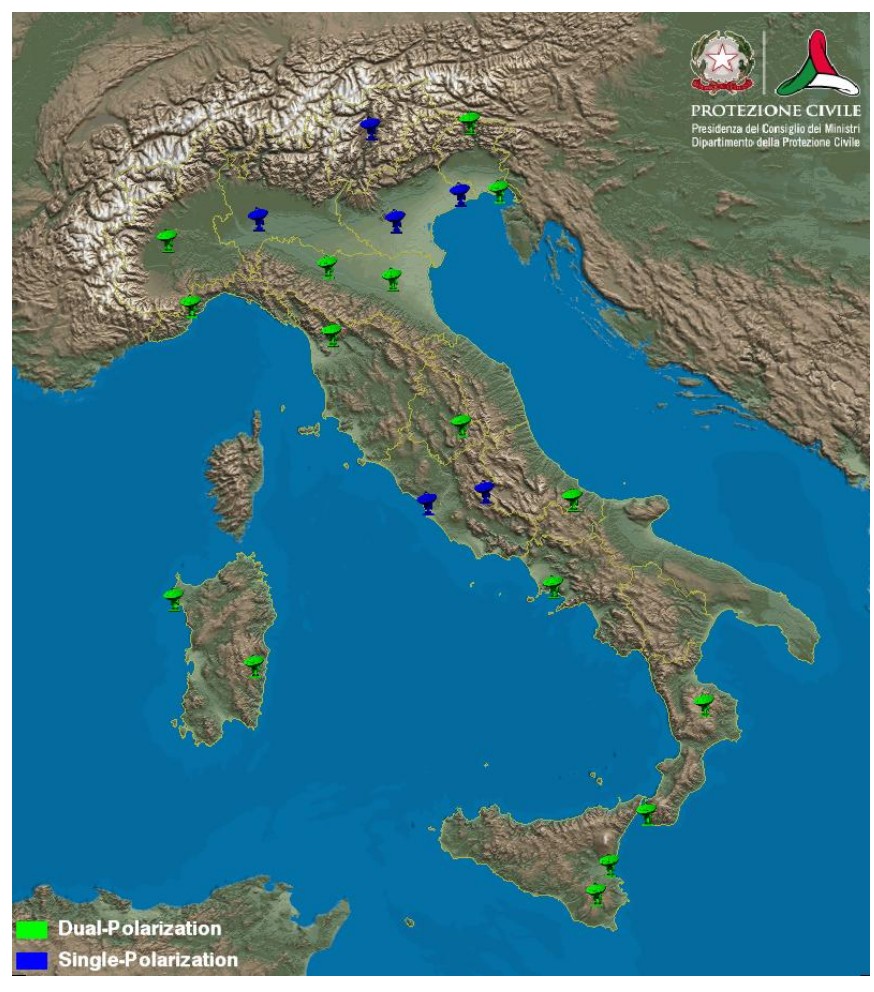


Figure 13: The radar network of the Department of Civil Protection. Green radars operate with dual-polarisation, blue
radars have single polarisation.







a)

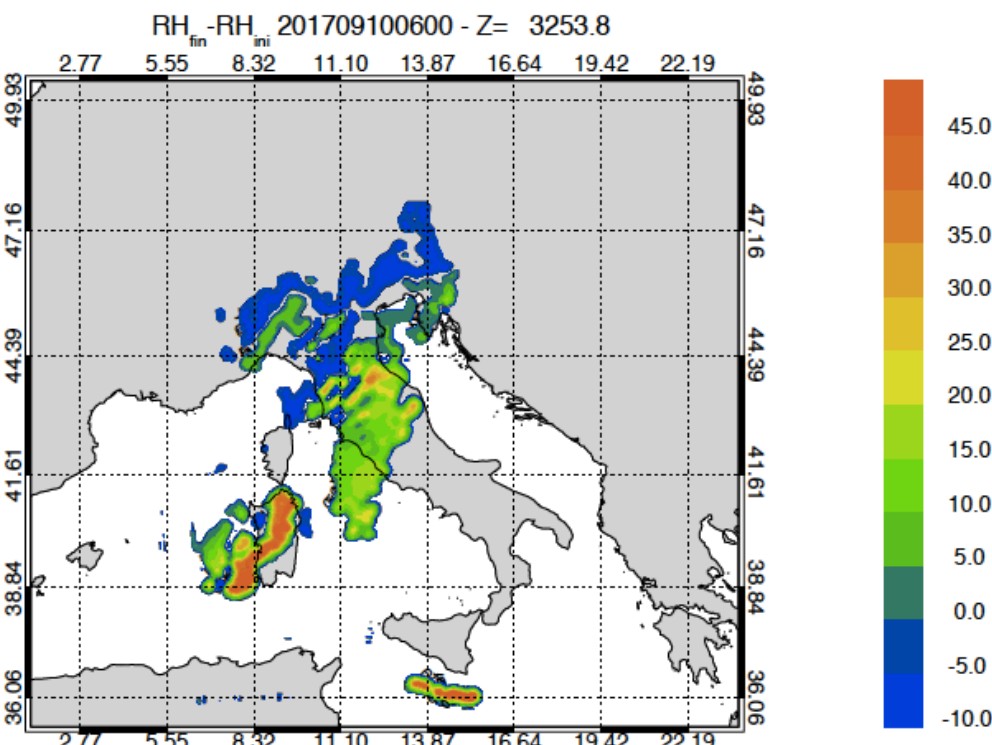


b)

Figure 14: a) RAMS@ISAC reflectivity factor simulated 3 km above sea level at 06 UTC on 10
September 2017; b) relative humidity difference between the analysis and the background at 06
UTC at 3.2 km level in the terrain following vertical coordinate of RAMS@ISAC.


a)

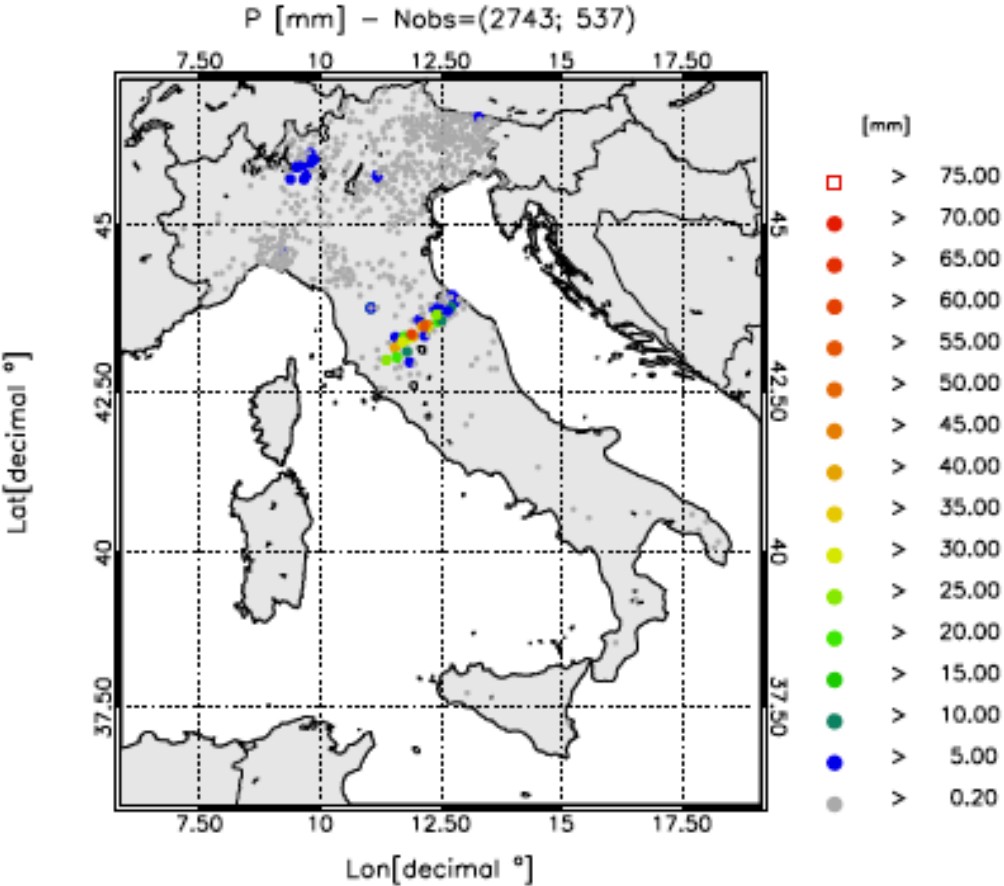



b)

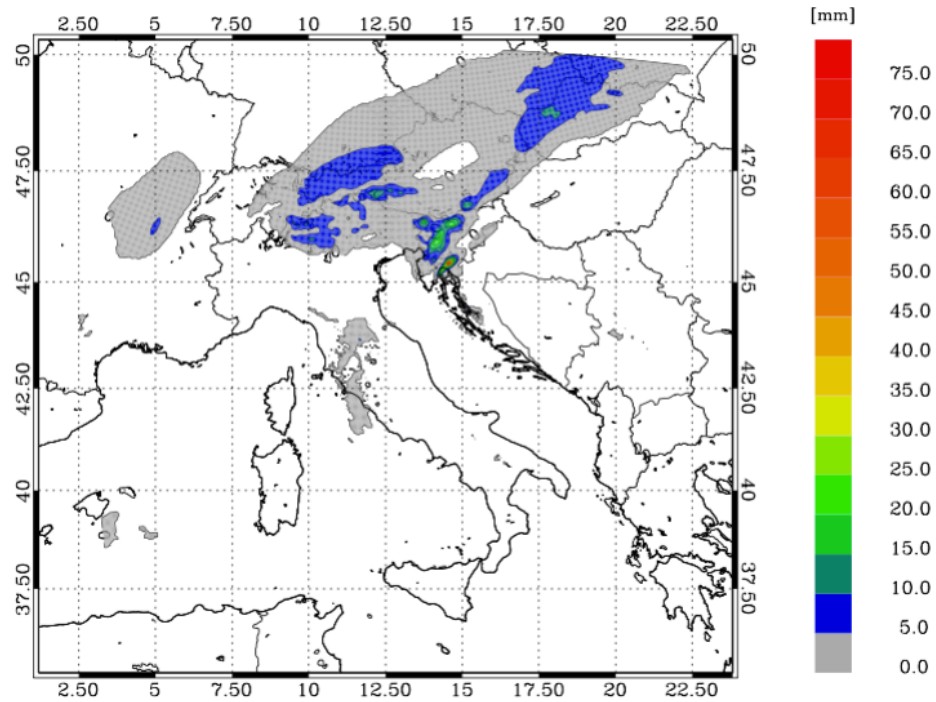



c)

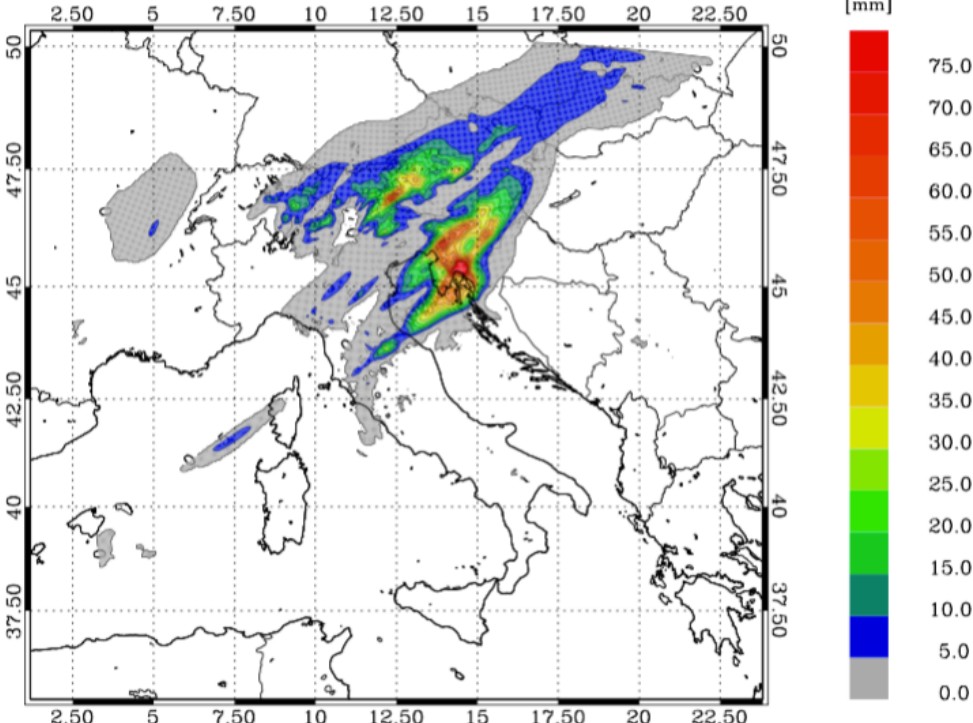




d)

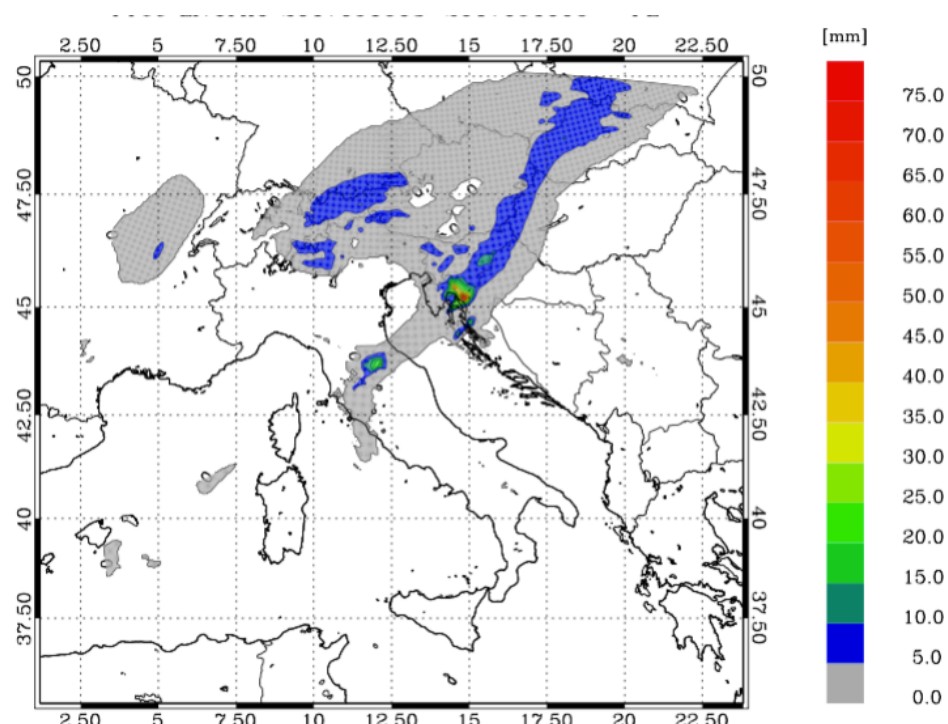



e)

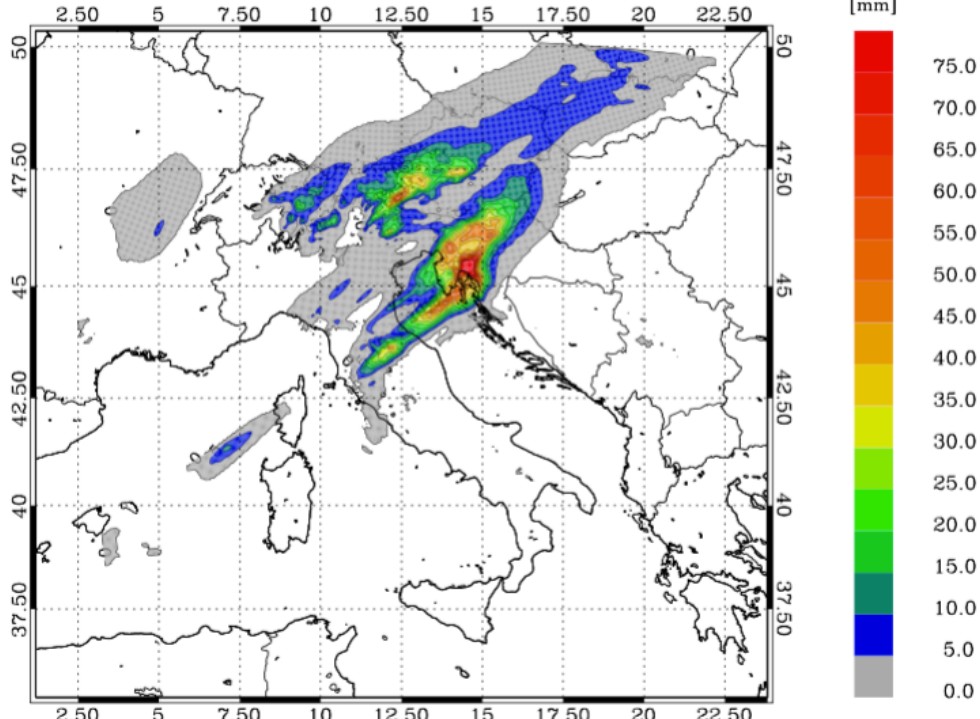


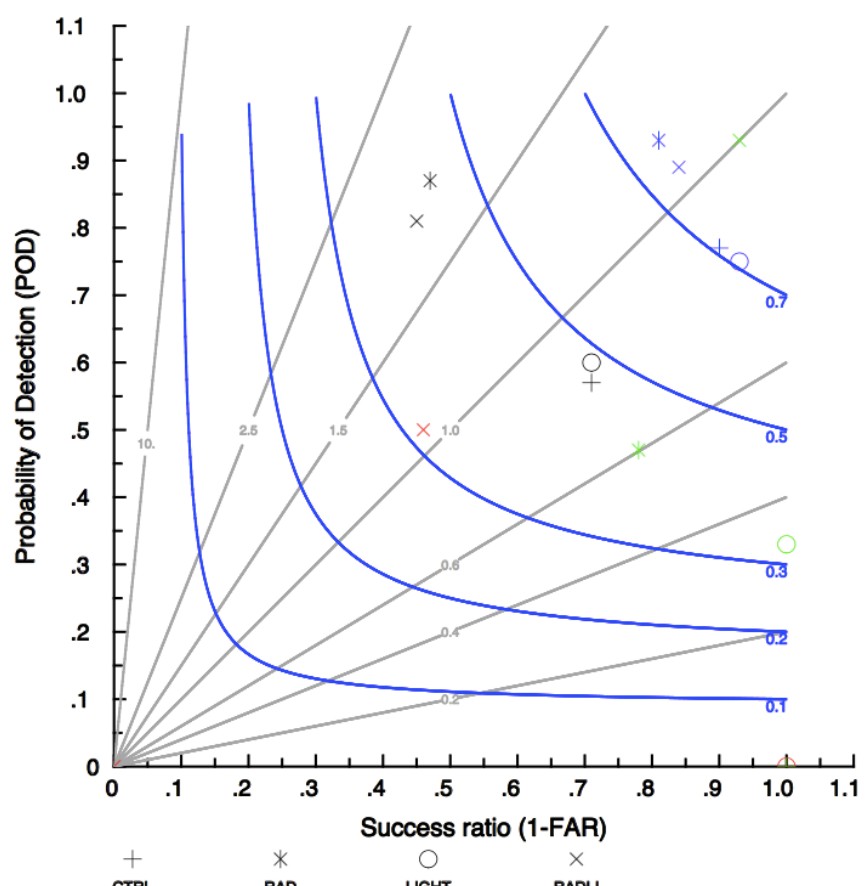



Figure 15: a) rainfall reported by raingauges between 03 and 06 UTC on 16 September 2017. Only raingauges observing at least 0.2 mm/day are shown. The first number in the title within brackets represents the available raingauges, while the second number represents those observing at least 0.2 mm/3h; b) rainfall  VSF of CTRL for the same time interval as in a); c) as in b) for RAD forecast; d) as in b) for LIGHT forecast; e) as in b) for RADLI forecast; f) performance diagram: black symbols are for the nearest neighbourhood and for 1mm/3h threshold; red symbols are for  the nearest neighbourhood and for 30 mm/3h threshold; blue symbols are for 25 km neighbourhood radii and for 1 mm/3h threshold; green symbols are for 25 km neighbourhood radii and for 30 mm/3h threshold.




























a)

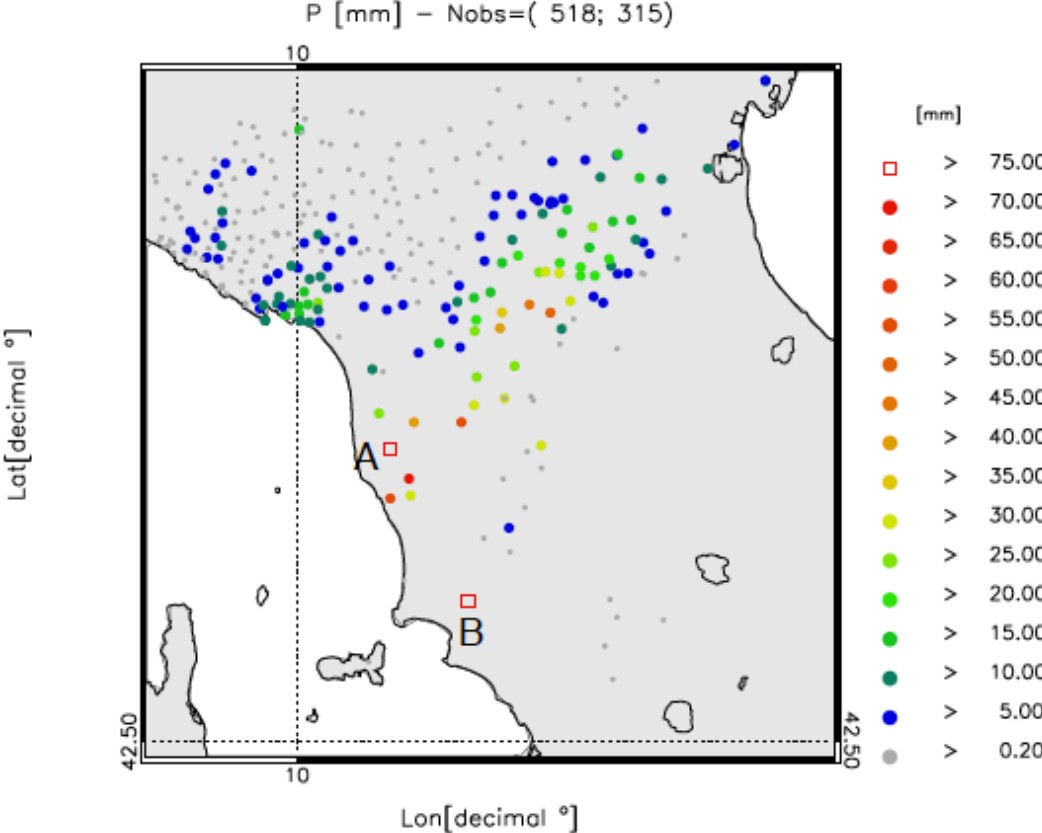

b)

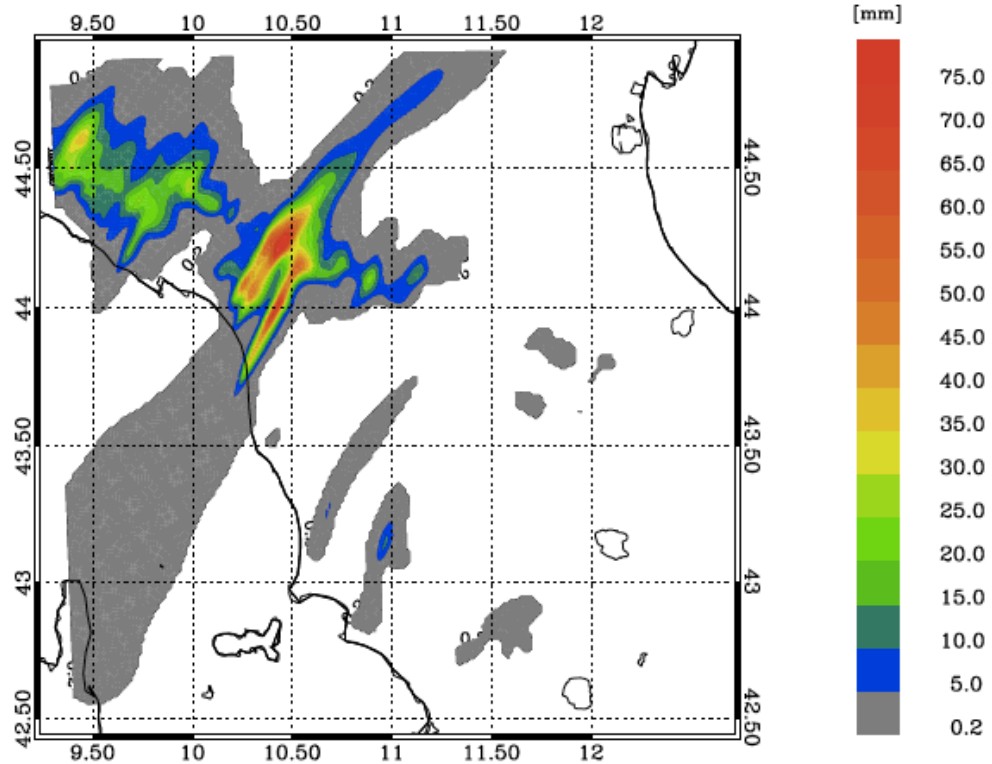


c)

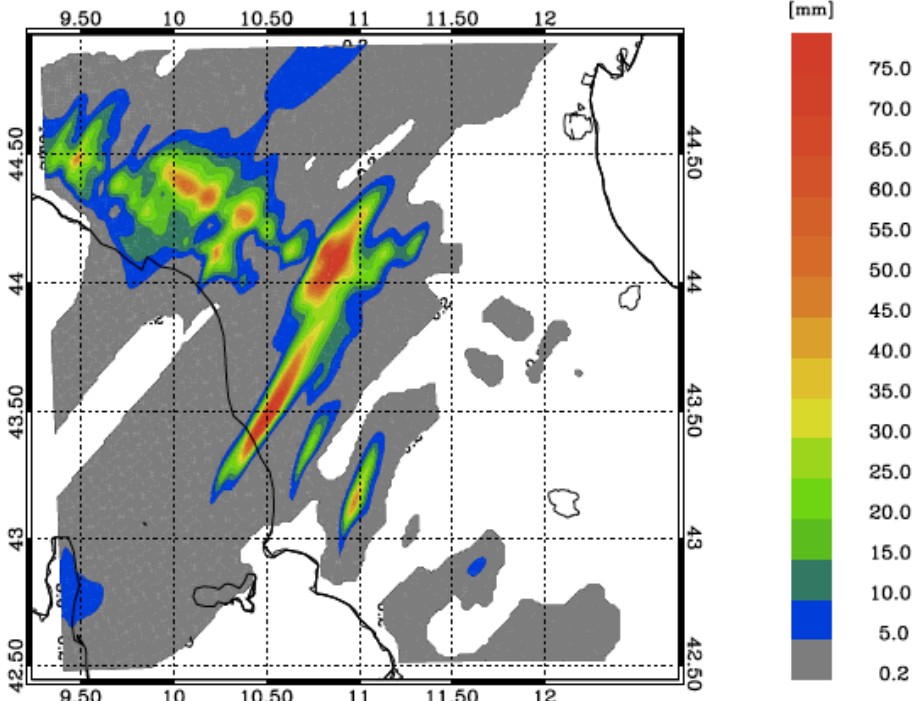



d)

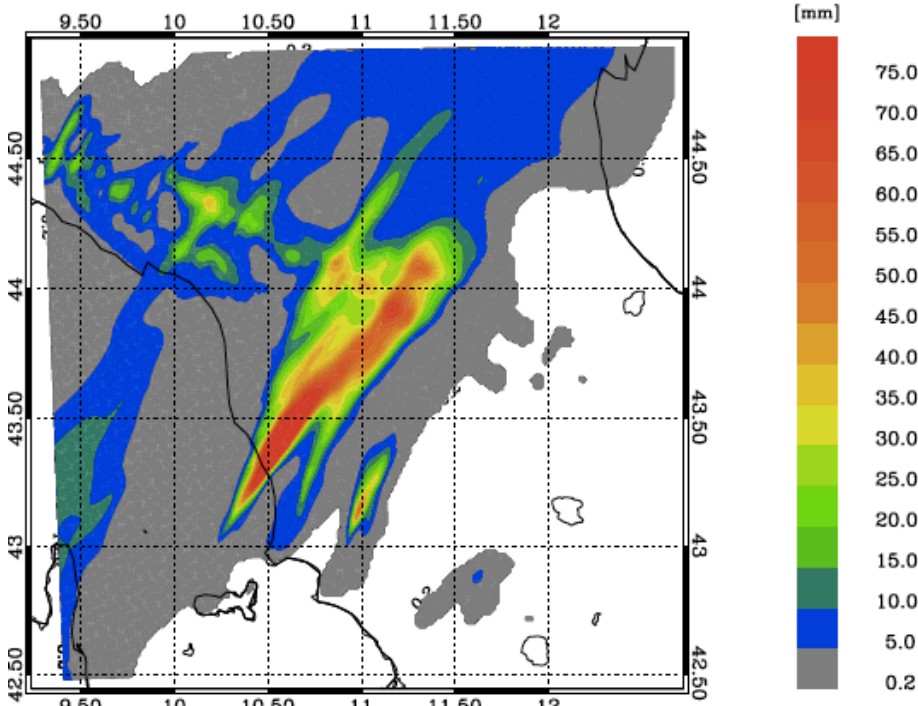







e)

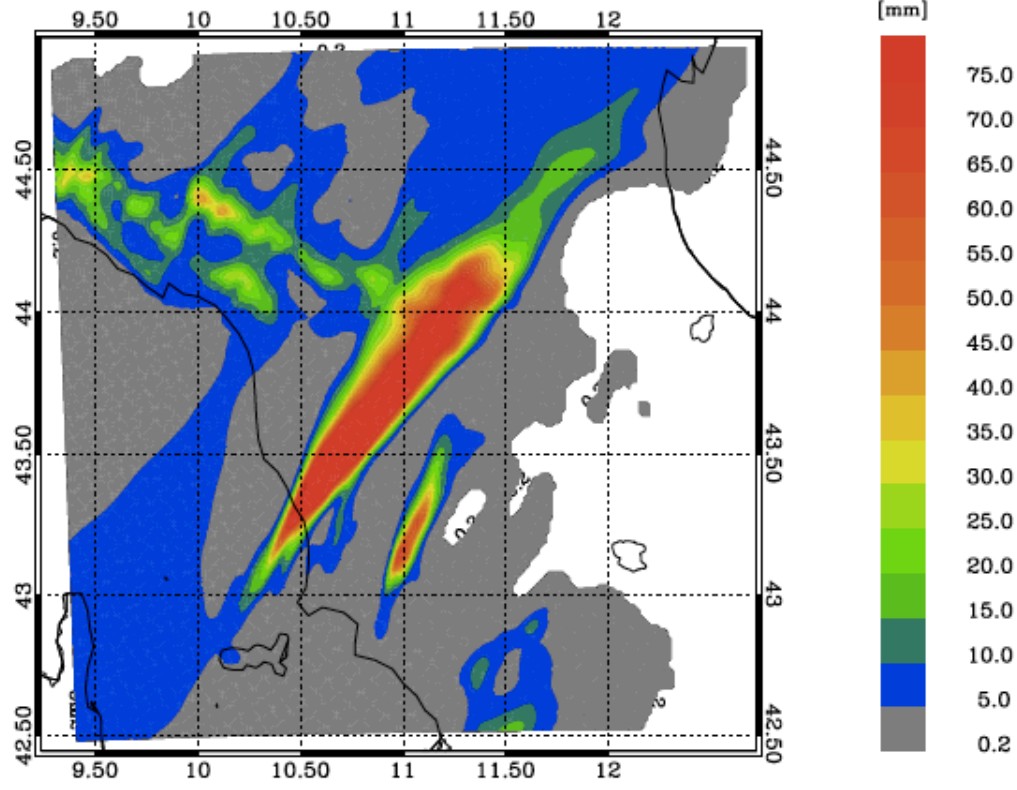


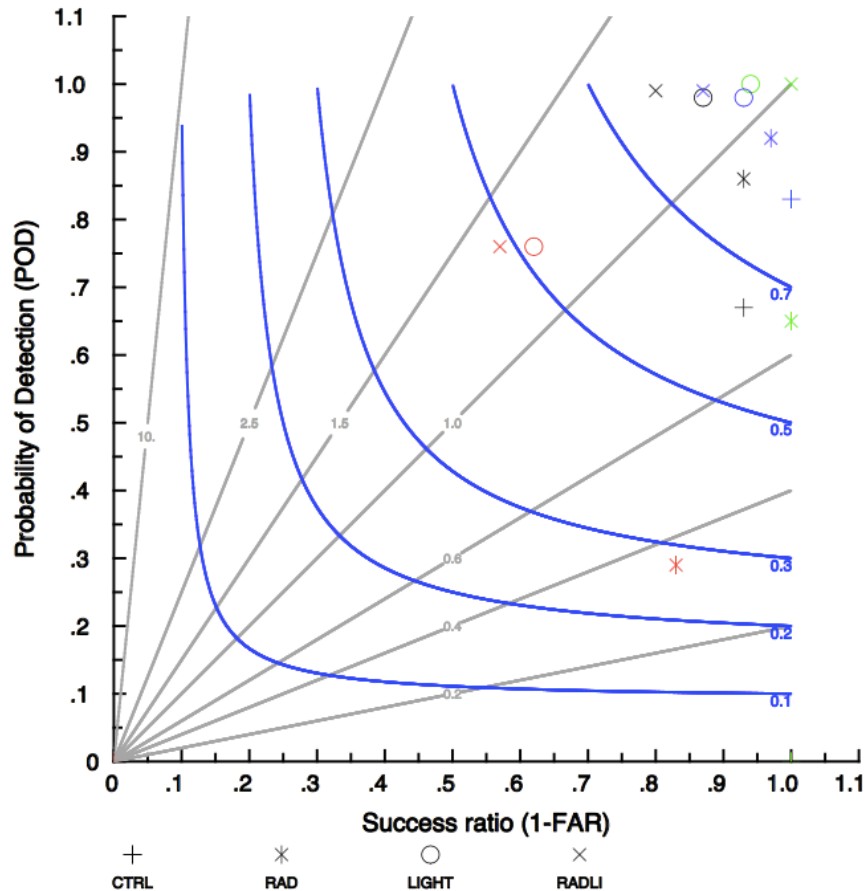

Figure 16: a) rainfall reported by raingauges between 00 and 03 UTC on 10 September 2017. Only stations reporting at
least 0.2 mm/3h are shown. The first number in the title within brackets represents the number of raingauges available
over the domain, while the second number shows those observing at least 0.2 mm/3h; b) rainfall VSF of CTRL for the
same time interval as in a); c) as in b) for RAD forecast; d) as in b) for LIGHT forecast; e) as in b) for RADLI forecast. Labels
A and B help to identify the positions of two rainfall maxima discussed into the text; f) performance diagram: black
symbols are for the nearest neighbourhood and for 1mm/3h threshold; red symbols are for the nearest neighbourhood
and for 30 mm/3h threshold; blue symbols are for 25 km neighbourhood radii and for 1 mm/3h threshold; green symbols
are for 25 km neighbourhood radii and for 30 mm/3h threshold.

1299       a)

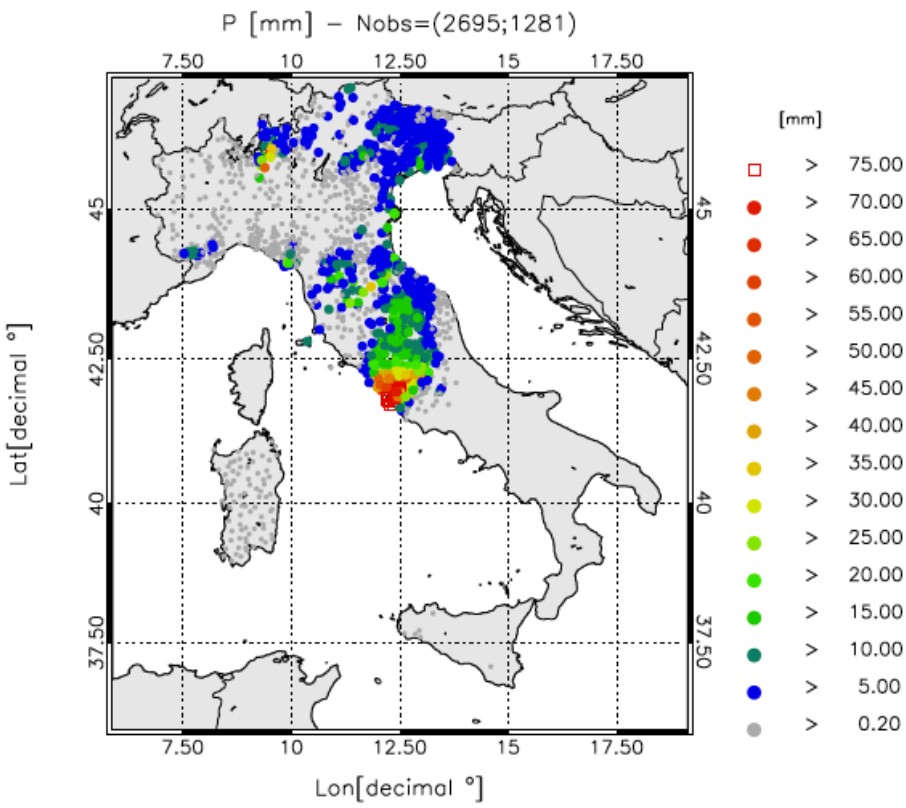


b)

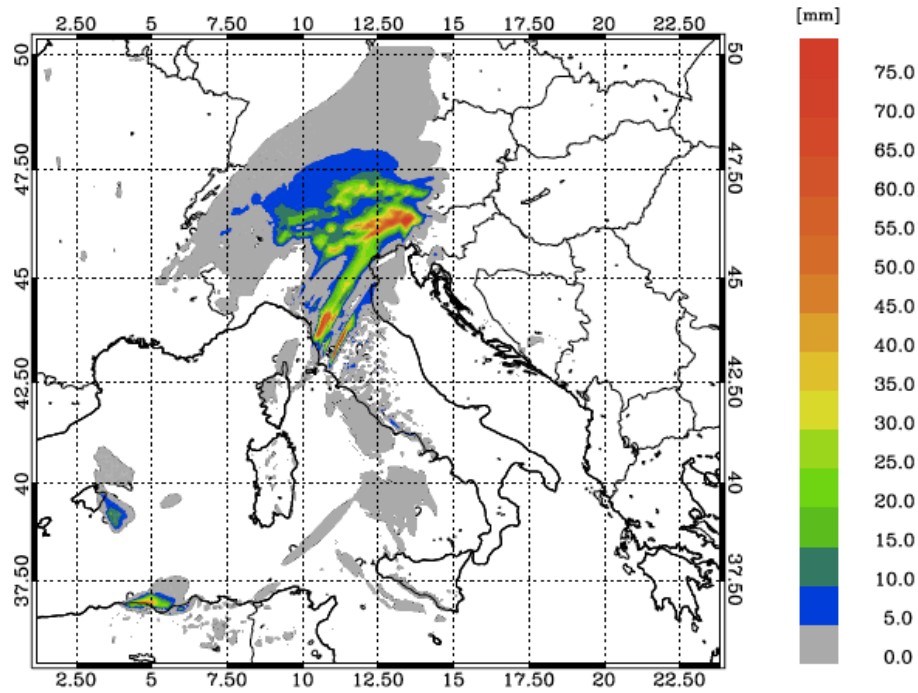





c)

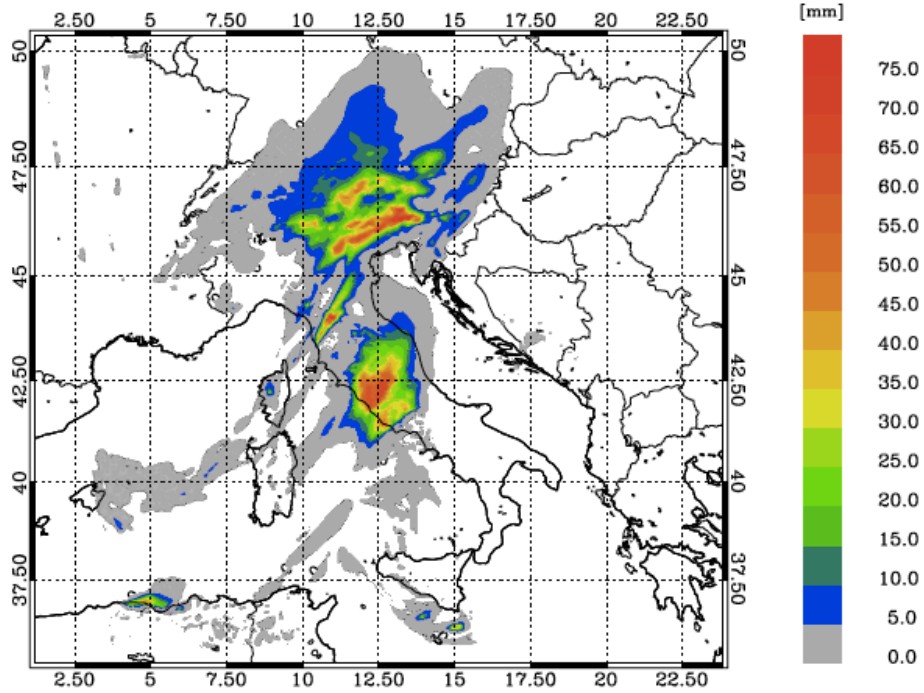



d)

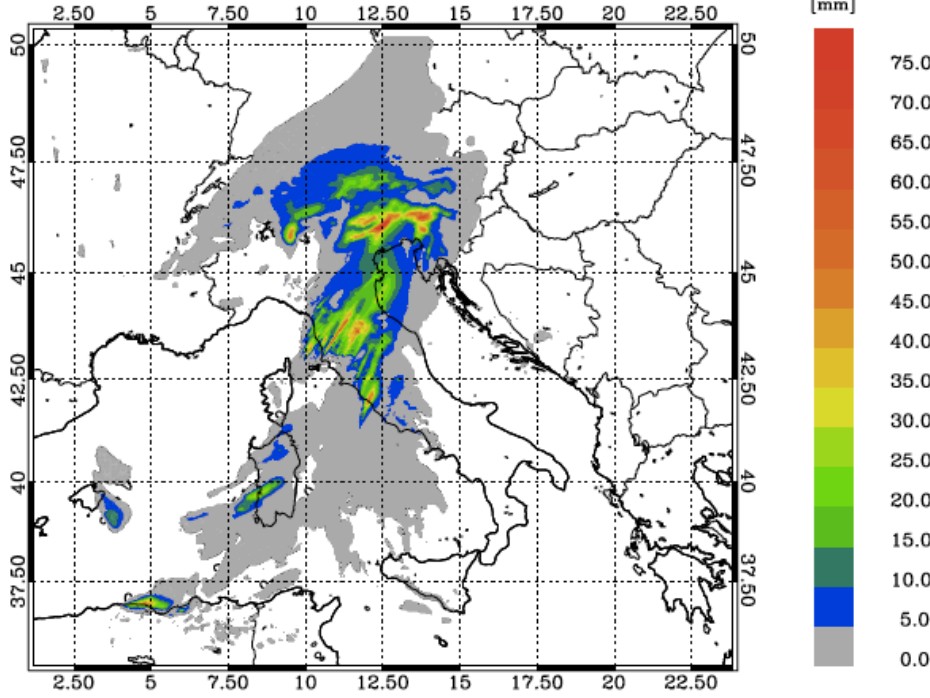



e)

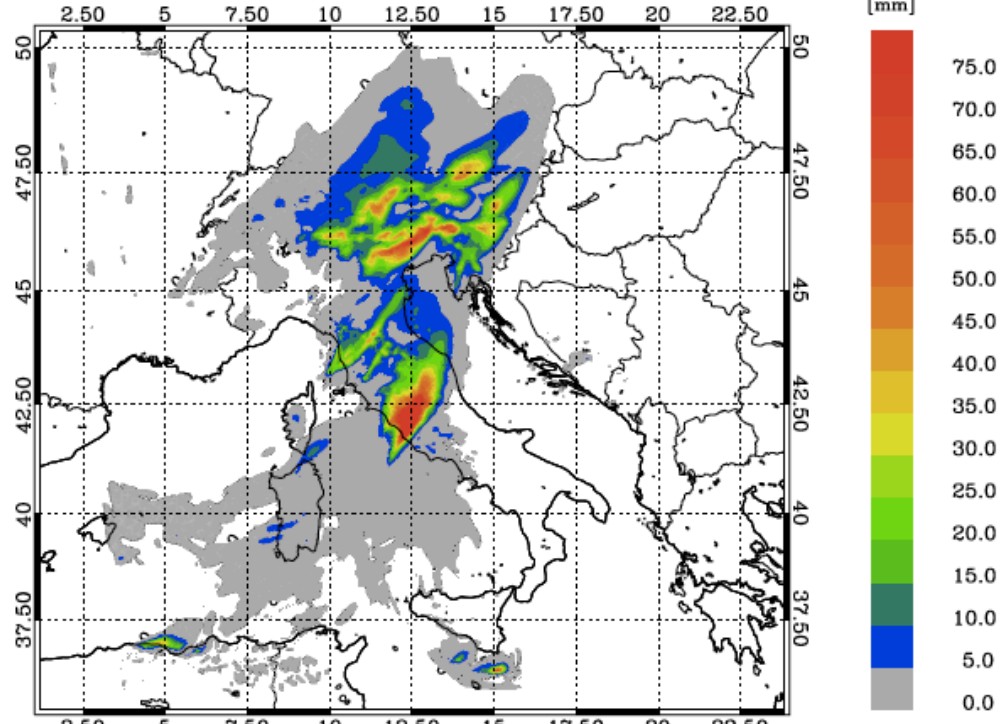









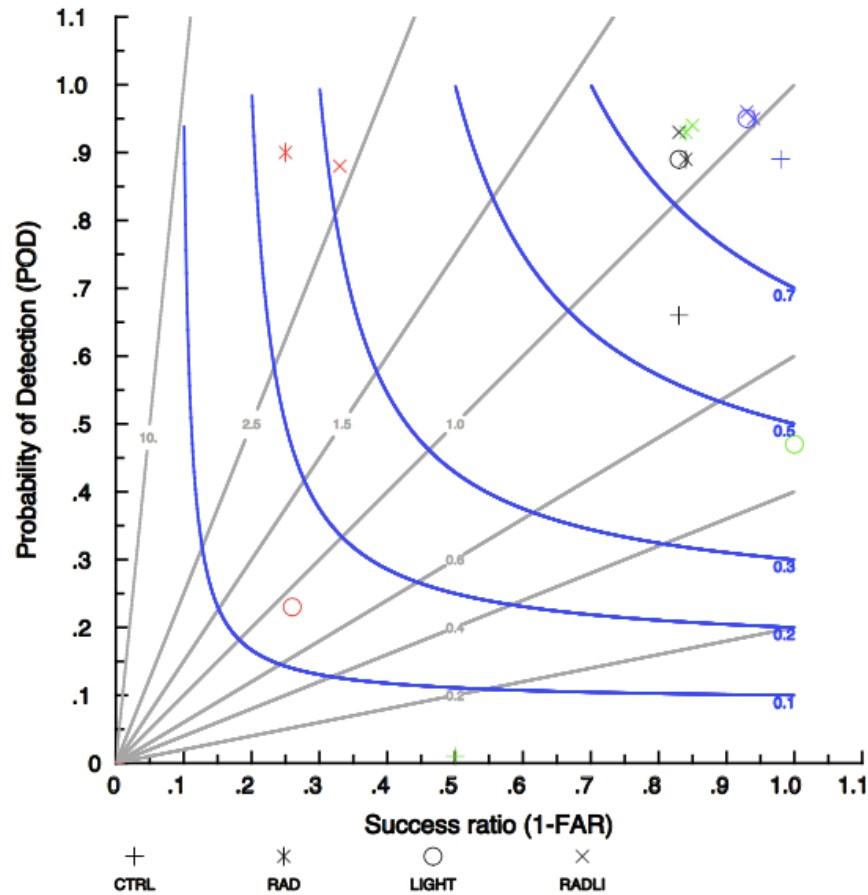

Figure 17: a) rainfall reported by raingauges between 06 - 09 UTC on 10 September 2017. For this time period 2695
raingauges reported valid observations in the domain, however only stations reporting at least 0.2 mm/3h are shown
The first number in the title within brackets represents the number of raingauges available over the domain, while the
second number shows those observing at least 0.2 mm/3h; b) rainfall VSF of CTRL in the same time interval as a); c) as
in b) for RAD forecast; d) as in b) for LIGHT forecast; g) as in b) for RADLI forecast; f) performance diagram: black symbols
are for the nearest neighbourhood and for 1mm/3h threshold; red symbols are for the nearest neighbourhood and for
30 mm/3h threshold; blue symbols are for 25 km neighbourhood radii and for 1 mm/3h threshold; green symbols are
for 25 km neighbourhood radii and for 30 mm/3h threshold.

a)

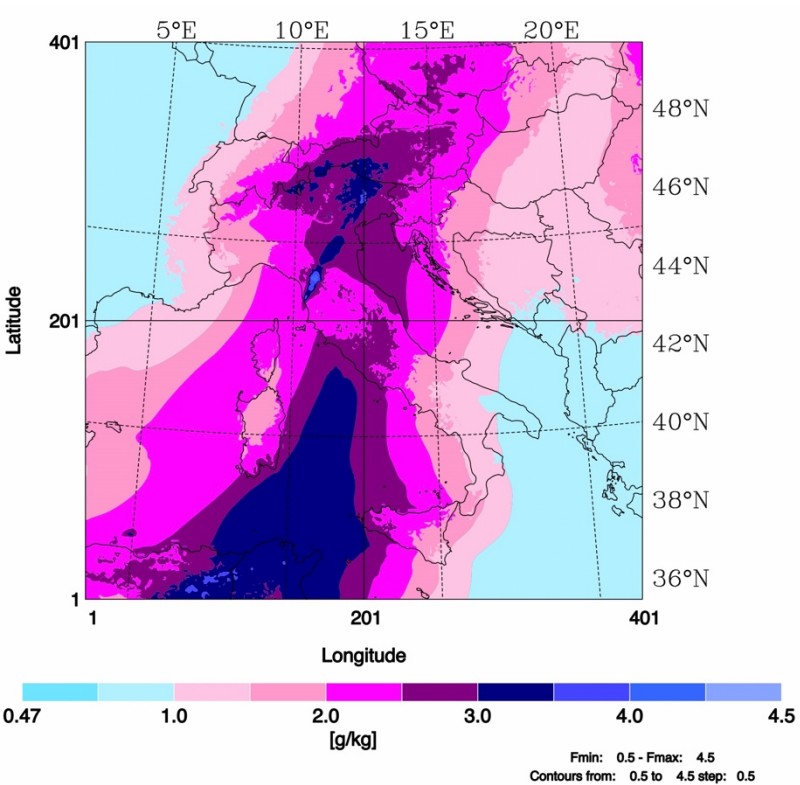

b)

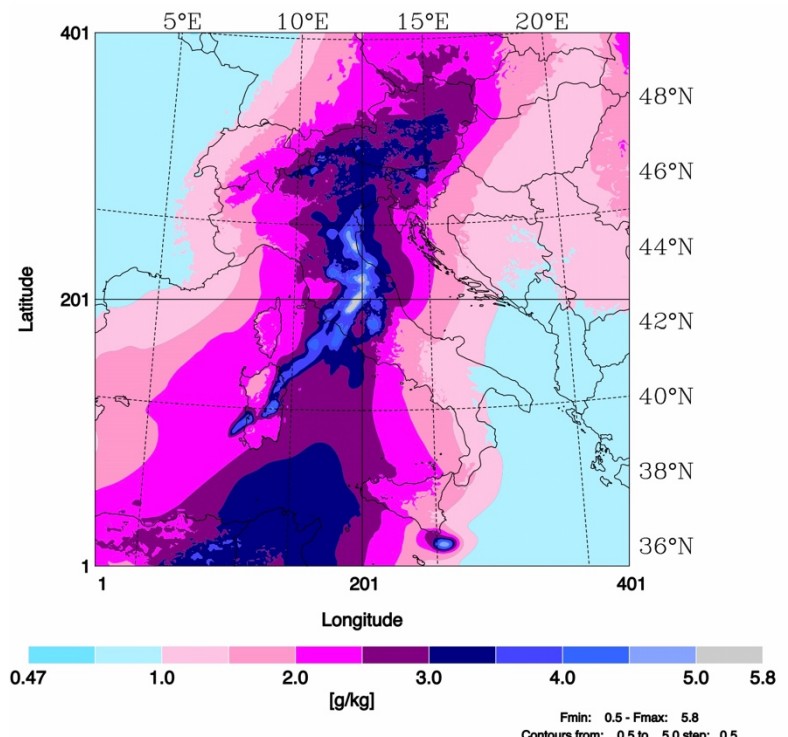

c)

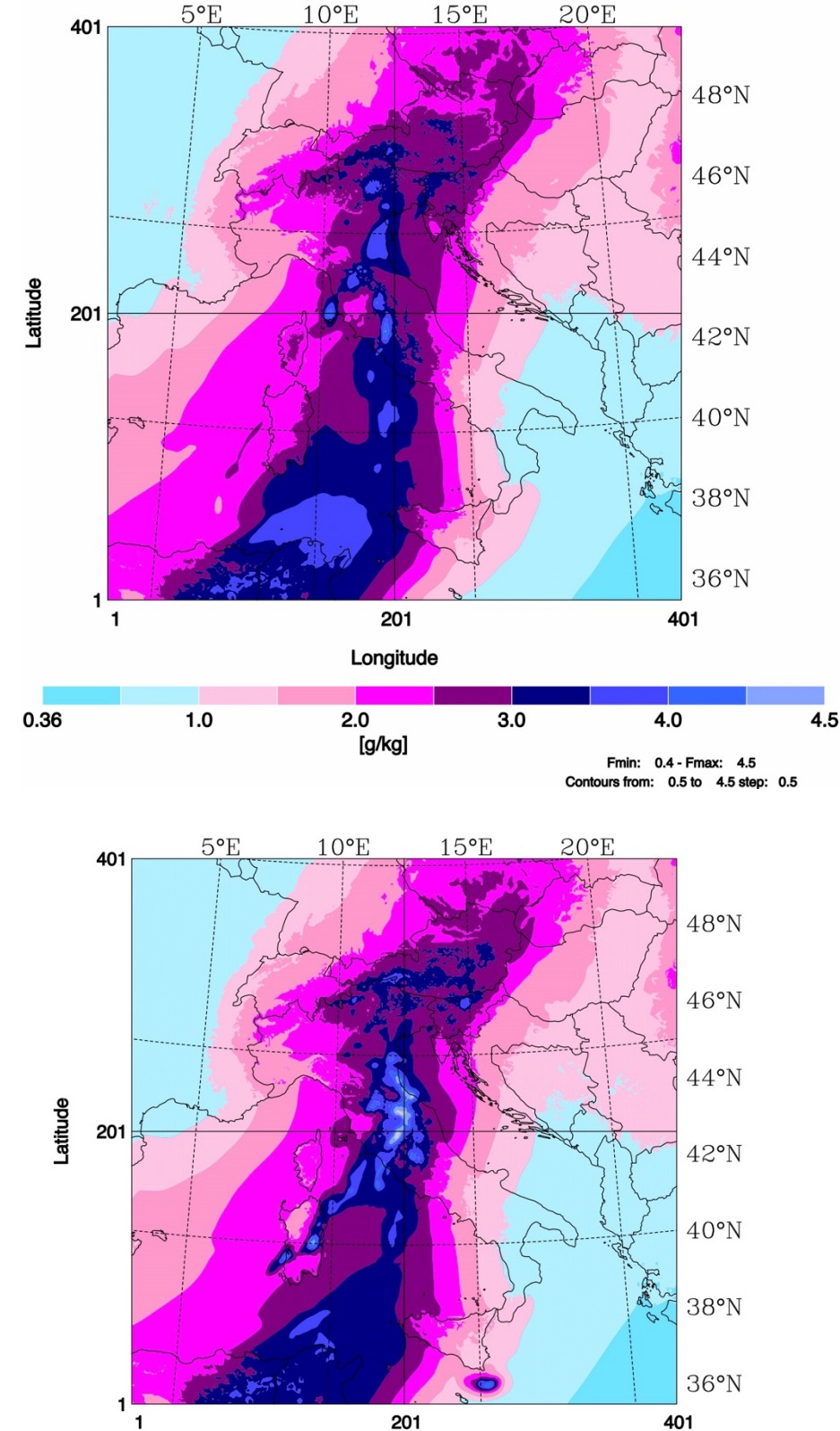

d)

Figure 18: Water vapour mixing ratio averaged between 3 and 10 km at 06 UTC on 10 September
2017 for: a) CTRL; b) RAD; c) LIGHT; d) RADLI.

a)

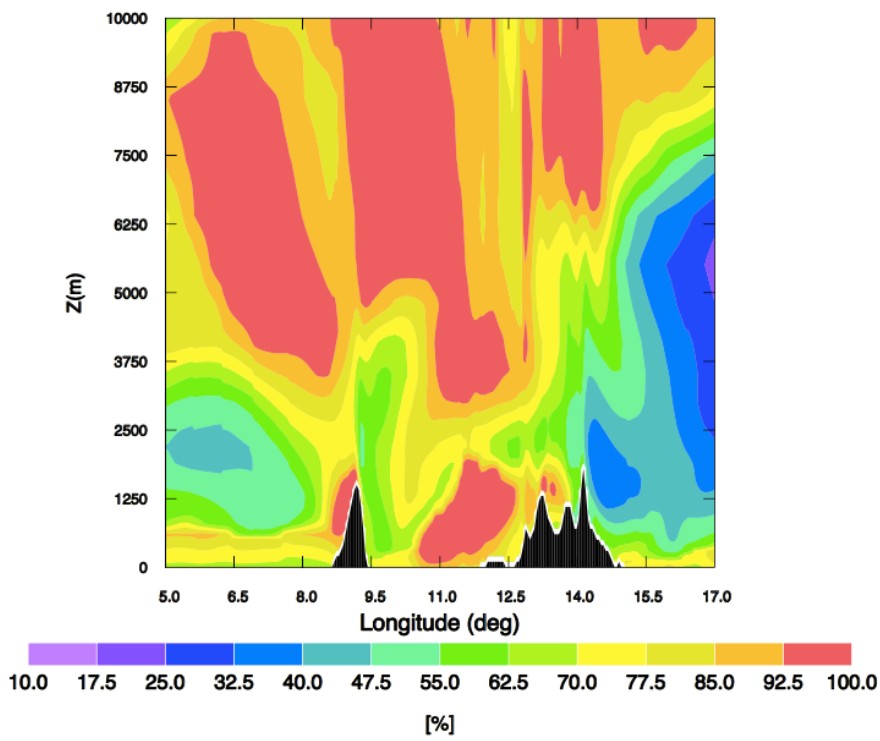

b)

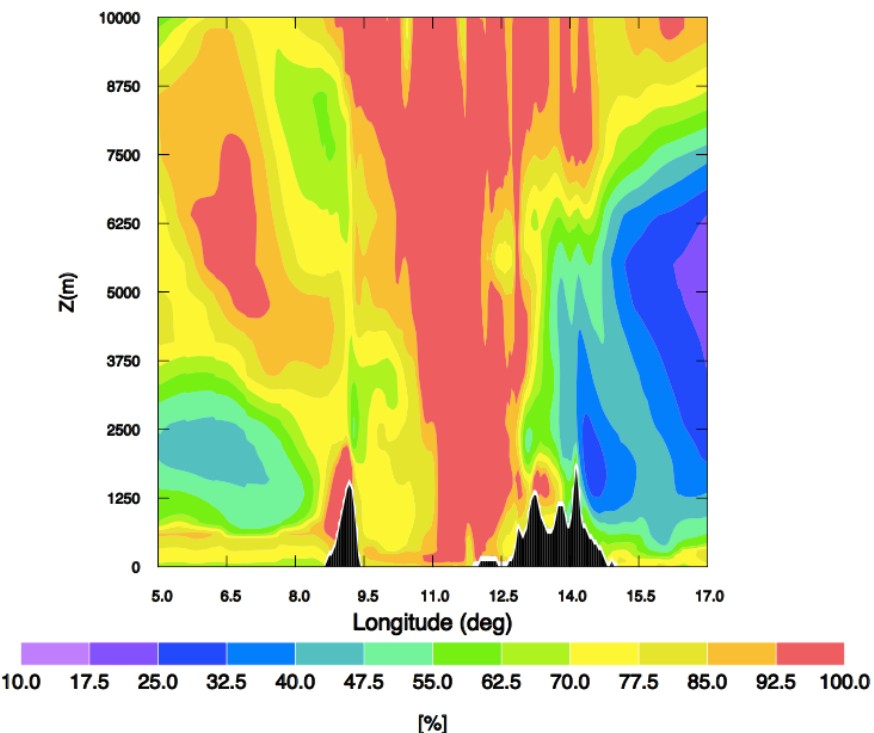

c)

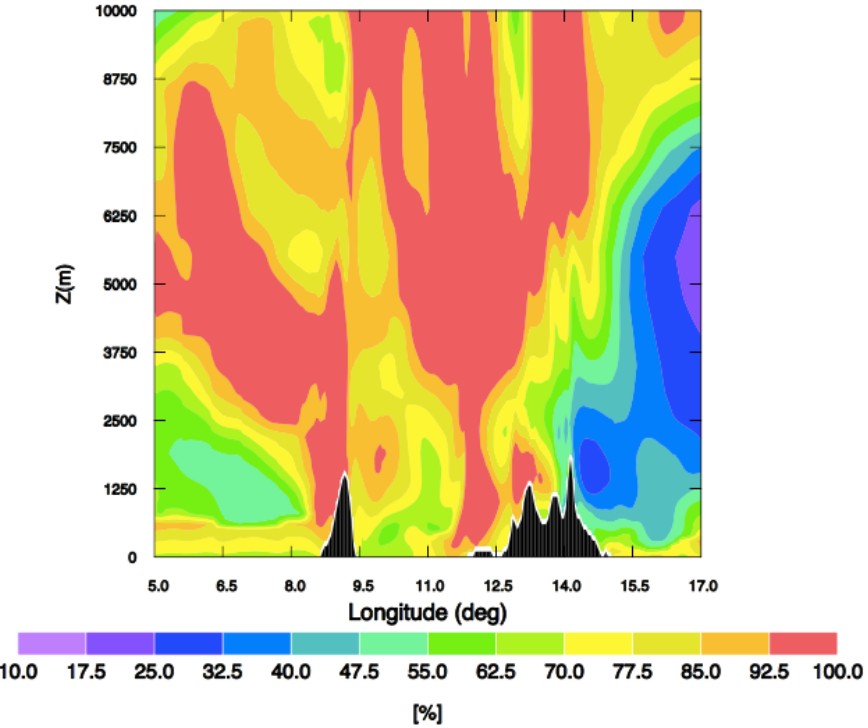

d)

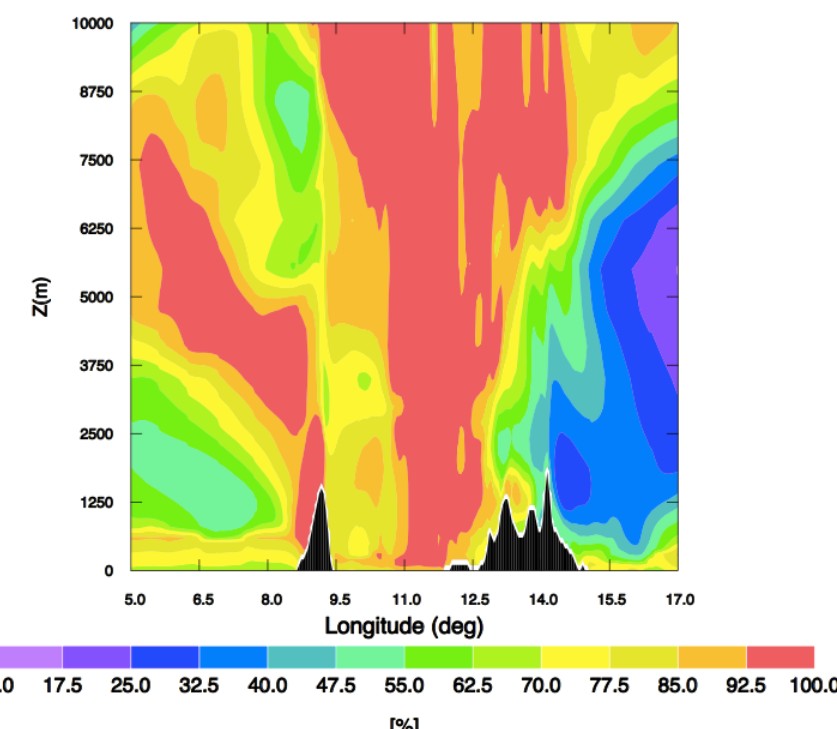

Figure 19: Relative humidity longitude-height cross-section at 42°N and at 06 UTC on 10
September 2017 for: a) CTRL; b) RAD; c) LIGHT; d) RADLI. Only the longitude range between 5 E
and 17 E and the vertical range between 0 and 10 km are shown for clarity.