# Peer review of "Preliminary results of the impact of lightning and radar reflectivity factor data"

_Natural Hazards and Earth System Sciences, 2018_

## Referee Comment (RC1) · Anonymous Referee #1 · 27 Nov 2018

Review of nhess-2018-319: Title: "The impact of lightning and radar data assimilation on the performance of very short term rainfall forecast for two case studies in Italy." by Federico et al.

Summary: The authors utilize a cloud-scale functional relationship between lightning and water vapor mass mixing ratio published in the literature and applied it to a home-grown 3DVAR framework at the convection-allowing scale to evaluate the analysis and short term forecast of two selected high impact weather events over Italy.

Recommendation: reject and, eventually, re-submit.

Main Comments: While the manuscript could eventually offer some merit for this jour-

nal, I found the analysis generally very rudimentary with the authors going at length in describing in excruciable level of details individual figures/panels in a repetitive and redundant manner without distilling the content into concise arguments/hypotheses. Given its repetitive nature, the entire results section could, in fact, easily be condensed into a 2-3 pages. Most importantly, the manuscript (hereafter, m/s) lacks rigor and rationales for the set ups and methods put forth for each, respective DA approaches. Salient Major issues are itemized below.

(1) As far as the scientific content is concerned, the core ideas and notions of this lightning data assimilation (LDA) method are conceptually similar to those from many existing studies, which fundamentally aim at promoting convective development through the introduction of latent heating within a prescribed neighborhood region/column centered at observed lightning locations. Past works from Benjamin et al. (2004), Alexander et al. (1999), Chang et al. (2001), Papadopoulos et al. (2005), Pessi and Businger (2009), have used empirical relationships between lightning-rainfall rates-latent heating or lightning-reflectivity rates-latent heating [e.g., in the HRRR]. Following a similar idea, recent works such as Machand and Fuelberg (2014), Lynn et al. (2015), Lynn (2017), Fierro et al. (2012; 2014, 2015), Wang et al. (2017, 2018) proposed LDA means that essentially boost the local thermal buoyancy where lightning is observed. A very limited portion of these techniques, however, offer alternative approaches to address spurious convection (i.e., removal) – which is a far more challenging problem to tackle. For completeness and given the relatively limited advances in LDA relative to radar DA, the authors should do a better job in discussing and including all the aforementioned references in their text. I was in fact astonished to notice that the integrity of the Results section in section 4 is completely devoid of references to previous works. In particular, since they opted to borrow an LDA method from one of these investigators, comparisons with their study should be performed more systematically throughout the m/s. For instance, the works of Federico et al. 2017b is invoked when referring to multi-day forecast statistics using the Fierro et al. method without mentioning that, such a study, was already conducted by the same author over a larger domain and using nearly three

times more forecast days/cases (Fierro et al. 2015 study). Given this omission, their study (Federico et al. 2017b) inadequately state that such multi-day statistics for this LDA have never been conducted. In a similar manner, it is of relevance to underline whenever appropriate that, in this work: (i) radial velocity is not included (specify why), (ii) only cloud-to-ground lightning data are considered and (iii) spurious convection is not addressed. In the light of (i) and (ii), one on the recent studies they cite (Fierro et al. 2016) not only assimilated level II radar data (radial velocity + reflectivity factor) but used total lightning data. This needs to be clearly stated, for completeness (Cf comment 3 below for rationales).

(2). In term of DA methodology, I found one major drawback, which is never discussed, nor evaluated. Given that both the LDA and their "RAD" experiment make adjustments to the relative humidity (RH) field, it is expected that both techniques will overlap in their adjustments over all the (many) grid points characterized by observed lightning flash rates exceeding zero. This is because changing RH is equivalent to adjusting $Qv$ as $RH \sim Qv/Qv\_saturation$. A more self-consistent DA approach would adjust the pseudo-observations for the $Qv$ or RH field in a manner that eliminates any possibility of overlap during the minimization. Toward that end, the authors should include soundings and/or horizontal cross sections of RH/$Qv$ that shows, quantitively, how the RH field is adjusted by each respective DA approach (radar vs lightning). Second, given that lightning is a cloud-scale observation, I cannot find any justifications for not conducting the 3DVAR analysis on the innermost, higher resolution domain. Instead, the method minimizes the cost function on the intermediate domain and, later, projects the innovations on the coarser-scale domain. This needs to be addressed. Third, the radius of influence/decorrelation length scale chosen for radar reflectivity factor (50 km) is far too large for convective scale applications and would incur unrealistically large amount of $Qv$ mass added into the domain – which will undoubtedly yield to spin-up issues and the generation of convective-scale gravity waves that will degrade longer term (>= 3h) solutions (please provide plot of perturbation pressure in your response). In that regard, the authors should indicate and contrast the total amount of $Qv$ mass

added by RAD and LIGHT.

(3). In the context of forecast improvements, the Qv-based method they borrowed/adapted was scaled for total lightning data (> 50% detection efficiency of intracloud [IC] flashes). I was surprised to find that absolutely no information on the detection efficiency and geolocation accuracy of the lightning network used (LINET) is provided in the text [no figures either]. Given the large area covered by this study, it is thus very likely that the geolocation accuracy of this network remains very poor for low amplitude flashes and for all flashes over oceanic regions. Given the low sferics amplitudes of IC flashes, the VLF portion of the sensor will miss nearly all these flashes, while the VHF portion only is able to detect some of the IC flashes within a few tens of kilometers away from the station [e.g., Rison, MacGorman works]. Thus, it is relevant to state and underscore that LINET only detects a very small portions of the total IC flashes in the study domain (likely < 5%). Motivation for scaling the F12 method for IC flashes (in lieu of cloud-to-ground [CG] flashes), lies in the well-documented finding that, in contrast to CGs, ICs are well correlated with thunderstorm kinematic and microphysical evolution (updraft strength, updraft volume, graupel mass etc, see Wiens et al. 2005, Schultz et al. 2011 among many others). CGs, on the other hand, were found to be correlated with the descent of reflectivity cores and the onset of the demise of the storm's updraft core [MacGorman and Nielsen 1991, MacGorman et al. 1989, Rutledge and Lang's seminal works etc]. Not surprisingly, ICs were found to lag CG by an average of 15 min [see one of the recent MacGorman study]. Moreover, Boccippio et al. 2001 and Medici et al. 2017 found that in deep continental convection, IC flashes always outnumber CGs by a ratio sometime exceeding 10:1. Based on these facts, it becomes clear why the Fierro method emphasized the use of IC flashes [or total lightning] for their application. Further motivation arises from the recent successful launches of the GLM instrument aboard GEOS-16/17, which will provide continuous day/night coverage of total lightning at ∼90% detection efficiency (DE) over a large domain covering the Americas (Gurka et al. 2006; Goodman et al. 2012, 2013, Rudlosky et al. 2018). Note that GLM will provide flash extent information of lightning, while the

metric derived from the (limited) point flash data in this study can only provide a very rough surrogate for CG flash location density at best. Similar space-borne technology to detect lightning have been developed by China (Feng-Yun-4, yang et al. 2016) with these data being assimilated in recent works by Wang et al. (2017, 2018) – which were never referenced either. Apart from their propensity to detect total lightning at a high DE, the chief advantage of this technology lies in its ability to retrieve lightning over remote oceanic regions. (4) The following key information pertaining to the respective DA methods are missing/never discussed: (a) What are the background/observation errors for reflectivity/lightning? (b) What statistics are used for model error ? (c) How is the adjoint for the lightning data assimilation operator derived ? (d) What assumptions are made for grid points with zero lightning or zero reflectivity observations ? Does the radar DA or LDA treat those as missing observations or equate those to the background values to reduce spread ? (e) What Gaussian decorrelation length scales are assumed for each observation ? Please specify/justify/explain. How would the selection of a given length scale value, influence the results ?

(f) Is spurious convection addresses by either DA method ? Please elaborate.

(g) Does the variational minimization set use a multi-scale approach ? If yes, what influence radii are chosen and how many cycles ?

(5) Why did the authors not include the fractions skill score FSS as the main evaluation metric for their forecast? Several works have posited that, in contrast to ETS, FSS does not penalize displacement errors as much and, arguably, FSS offers a more accurate measure of skill on convection-allowing grids (Mittermaier et al. 2011).

Additionally, more recent studies evaluating forecast performance have been making usage of the so-called performance diagrams, which conveniently merge several key contingency table elements into one single diagram (Roebber 2009). The authors should show such diagrams to provide a more complete and succinct view of the overall forecast performance of the case they selected.

[Figure]

(6) The case studies selected are cherry-picked given the confession that CTRL generally failed to provide reasonable forecast estimates of precipitation for both cases herein. For good measure, fairness and to better underscore the performance of the DA method, the authors should show the results for one case in which CTRL did not perform well and contrast it to one case where CTRL did preform reasonably well.

(7) The authors omit to mention that the degradation of the forecast at >= 3h is mainly due to saturation of the model solution by errors and biases within the initial / boundary conditions derived from large scale models or re-analysis datasets. This needs to be shown for both cases, especially given the unrealistically large (50 km) decorrelation length scale used for radar reflectivity factor.

(8) Title: Revise to include that: (i) primarily CG flashes are assimilated and (2) the model vehicle is RAMS.

Because these issues are collectively substantial and would require thorough rewriting of the manuscript in many places, I opted not to dwell on editorial comments for the time being. Additionally, the level of English remains, in my view, unacceptable for publication.

Figures:

Figures 5, 6, 8, 12a, 13a, 14a, 15a, 16a, 17a, 18a: The use of colored dots makes it very difficult to effectively compare the observations with those of the simulations: For consistency, either both sets of plots should show colored dots or shaded contours. For lightning, the authors should effectively show the gridded lightning data that were used to create the Qv or RH pseudo-observations. Additional comments: General comment: What is the main rationale for using a model that is marginally known by the community (RAMS) versus a more commonly used, battle tested, publicly available model such as WRF-ARW ? The authors not only seem to re-invent the wheel here but render any potential future work dedicated to reproducibility of the results - to the least - very challenging.

[Figure]

(1) Bottom, page 2: what are "conventional data" ? Why are radial velocity data not used ? Line 70: the main advantage of using 3DVAR vs 4DVAR, EnKF or hybrid methods lies in their already low computational burden. Thus, I do not agree with this justification. Also, variables are not "perturbed"; but adjusted by VAR methods. (2) Pages 3 and 4: Please refer to Major Comments 1 and 3. Lines 105: Given that "Federico et al. (2017a) implemented the methodology of Fierro et al. (2012) ...", how come on line 112 "We use the method of Federico et al. (2017a) to assimilate lightning..." ? Please revise accordingly. (3) Line 124: c.f. end of Major Comment 1. (4) Line 240: RAMS used diagnostic relationships (vs explicit) to forecast lightning as it does not explicitly solves for the 3D electric field. Line 243: "Fourth" (5) Line 290: Delete equation set as these are considered basic/common knowledge. (6) Section 3.2, lines 300-312: Explicitly state and indicate that equation (2) is from Fierro et al. (2012, 2015) and not from Federico et al. Line 305: Please explain the rationales behind the choices of these constants: In particular, how are the forecast metrics affected for a 20% change in A, which has been shown to exhibit the most notable influence on the forced convection? (7) Line 316-317: c.f. Major Comment 2. (8) End of page 11: c.f. Major Comment 2 (9) Line 356: do the authors refer to the LFC or the LCL, (which may I add is an idea borrowed from Marchand and Fuelberg 2014 and Fierro et al. 2016). What is the top of the adjustment layer for lightning ? Please elaborate. (10) Line 410 and elsewhere. This is similar to the results of Fierro et al. 2016. C.f. Major Comment 1. Please establish comparisons with previous works throughout the manuscript. (11) Line 669: This statement is incorrect. The DE of ground based sensors levels off very rapidly with distance from land. This is where space-borne lightning detection systems such as the GLM or Feng Yun-4 can fill the gap. (12) Lines 716-725: c.f. Major Comment 1.

Please also note the supplement to this comment:
https://www.nat-hazards-earth-syst-sci-discuss.net/nhess-2018-319/nhess-2018-319-RC1-supplement.pdf

---

## Referee Comment (RC2) · Anonymous Referee #2 · 11 Dec 2018

Review of nhess-2018-319: "The impact of lightning and radar data assimilation on the performance of very short term rainfall forecast for two case studies in Italy." by Federico et al.

Recommendation: major revisions

General comments: this manuscript addresses an interesting and challenging topic, moreover rerepresents a substantial contribution to the understanding of natural hazards and their consequences matching the scope of NHESS. However the scientific and presentation quality are poor, above all because the results are presented in a "repetitive and heavy" manner and the English language needs a deep revision.

The scope of the paper is sufficiently clear, less robust is the novelty, although the introduction is rich of references. Chapter 2 on the case studies are widely described and supported by a lot of details and figures. Also chapter 3 about data and methods is enough detailed and quite well written. Results in chapter 4 should be rewritten and differently organized to make the speech more fluid and less "arduous" to read (sometimes it appears boring), for example showing less figures and, in some cases, summarizing the scores in tables. Specific comments: please see the notes on the pdf for each section and also for each figure's caption, moreover please deeply motivate the reason why: a) radial velocity is not assimilated: the operator is not implemented or the data are not available?; b) data assimilation is performed in the domain D02 only; c) you used a background error matrix of fall 2012 for case studies of late summer 2017. Technical corrections: please see the notes on the pdf for each section and also for each figure's caption.

Please also note the supplement to this comment:
https://www.nat-hazards-earth-syst-sci-discuss.net/nhess-2018-319/nhess-2018-319-RC2-supplement.zip

---

## Author Comment (AC1) · 5 Feb 2019

**Answer to the reviewer #2 comments on NHESS-2018-319.**

We acknowledge the reviewer for the useful comments on the paper, both in the general comments section and in the pdf file. Our answers are in red.

Before starting the discussion, we note that, in reviewing the paper, we found two errors: a) the length scale of the background error matrix in the x and y directions varies between 14 and 25 km and not, as stated into the paper, between 20 and 30 km; second the lightning number for each day written into the initial manuscript are wrong. The correct numbers are 82 331 for the 9 September, 291 164 for the 10 September (170 000 is written into the manuscript) and 105 467 for the 16 September (60 000 written in the manuscript). We apologize for these errors. However, the results shown in the paper were obtained using the correct number of flashes and the correct length scales in the background error matrix.

Extracted from the general comments:

This manuscript addresses an interesting and challenging topic, moreover represents a substantial contribution to the understanding of natural hazards and their consequences matching the scope of NHESS. However, the scientific and presentation quality are poor, above all because the results are presented in a "repetitive and heavy" manner and the English language needs a deep revision.

Specific comments: please see the notes on the pdf for each section and also for each figure's caption, moreover please deeply motivate the reason why: a) radial velocity is not assimilated: the operator is not implemented or the data are not available?; b) data assimilation is performed in the domain D02 only; c) you used a background error matrix of fall 2012 for case studies of late summer 2017. Technical corrections: please see the notes on the pdf for each section and also for each figure's caption.

Comment "…in a repetitive and heavy manner". In the first submission of the paper we stressed the important improvement given by the data assimilation at the local scale on the precipitation VSF (Very Short term Forecast, 0-3h). To highlight this point, we showed the many ways in which the forecast is improved by the assimilation of lightning, radar or both. For example, the two stages of the Serano case show that the radar (first phase 03-06 UTC on 16 September) or lightning (second phase of the event, 18-21 UTC on 16 September) were the key observation to assimilate in order to improve the precipitation VSF. Also other stages had some specific aspects that we discussed.

Our attempt, however, was not successful, given the comments of both reviewers and the results section (Section 4) underwent a substantial rewriting.

In particular, in the revised version of the paper, we will delete the Section 4.1.2 (second phase of the Serano case) and Section 4.2.1 (first case of the Livorno case). The results Section 4.2.1 will be shortly commented in Section 5 (Discussion and conclusions) to highlight that there is space for improvement. Following the comments of the reviewer #2, the scores of the phases discussed into the paper will be put in three tables (Tables: 4-6) for specific thresholds (1, 6, 10, 20, 30, 40 mm/3h and, for Livorno, also 50 mm/3h) and, following the remarks of the reviewer #1, different neighborhood radii will be used to compute ETS and POD scores.

The space gained by deleting the two sub-sections stated above will be used to extend the discussion about the methods of assimilating lightning and radar in the RAMS@ISAC model and to add two short sections to the result section. In particular, we will extend the section "Lightning data assimilation" to include a discussion of the useful comments raised by the Reviewer#1, we will extend the section "Radar data assimilation" to show an example of 3D-Var assimilation of reflectivity factor (this will also answer to few comments of the Reviewer #1). A draft of these revised sections will be reported at end of the answer to the Reviewer #1 comments.

Finally, we will add a section (Section 4.3) to show how the lightning and radar data assimilation works together, presenting the evolution of the total water averaged for all VFS of the two cases and including in this discussion the assimilation stage, as well as sensitivity tests of the precipitation scores (POD and ETS) to the nudging formulation (Section 4.4). Section 4.4 requested new simulations with different model settings (see Table 3 at the end of this answer).

A draft of the new Results section (Section 4), in clouding the new sections 4.3 and 4.4, is shown at the end of this answer. This could not be the final form because minor changes are still possible.

Comment "…the English language needs a deep revision". We will revise the English of the paper, also according to the suggestions of the reviewer #2 in the PDF file. Also, the copy-editing service of the journal will improve the quality of the English before the eventual publication of the paper.

Specific question a) We are working on the assimilation of the radial velocity but the operator is not yet implemented in the 3D-Var. Also, while the reflectivity factor measured by the radar network is operationally available, the product of radial velocity is under development. At the moment, it needs further research to solve some issues (complex orography, operations of the radars not optimal for the Doppler retrieval and others). For these reasons, the attention was on the assimilation of radar reflectivity factor. These motivations will be discussed in the revised version of the paper in Section 3.3 by writing:

"Radial velocity is not assimilated in the RAMS@ISAC model because the operational product of radial velocity needs research to solve issues (complex

orography, operation of radars non optimal for Doppler retrieval, not homogeneous coverage of the country), and it is not available for assimilation. Also, the implementation of radial velocity data assimilation is under development in RAMS-3DVar and it is not available for testing. For these reasons, we didn't consider the assimilation of radial velocity in this work. "

Specific question b) Data assimilation is not performed on domain D3 (R1) because we don't have background error statistics for this grid . The background error statistics are important in 3D-Var and, while radar data (we apply 3D-Var for radar only) are cloud-scale observations and could be used for the 3D-Var assimilation in domain D3, we cannot apply it on this domain.

Background error statistics for the domain D2 are computed by the NMC method, which, for this paper, is based on HyMeX-SOP1 simulations. The Appendix A and B of Federico (2013) shows the details. of the application of the method, which requires a number of simulations (see Barker et al., 2004 for the general discussion). These simulations are not available for the innermost grid of the Livorno case, which was introduced to better resolve the precipitation at the local scale and to show how precise can be the impact of the lightning and radar data assimilation on the VSF. These motivations will be clarified in the revised version of the paper.

Of course, this limitation is only for radar reflectivity factor because flashes are assimilated by nudging. Nevertheless, we could not compare simulations with or without data assimilation for a specific domain assimilating lightning in the innermost domain and for this reason we assimilated flashes over the D2 only.

In the paper we will specify better the role of the domain D3 and the reason for not assimilation lightning and radar reflectivity factor over the domain D3.

We will write in section 3.1 (RAMS@ISAC and simulations set-up)
"The third domain covers the Tuscany Region, has 4/3 km horizontal resolution (R1), and it is used for Livorno to represent with higher spatial detail the precipitation field over Tuscany and to show better the precision of the rainfall VSF using data assimilation at the local scale. The fine structures of the precipitation field are smeared out over Tuscany using only domains D1 and D2. The operational implementation of the RAMS@ISAC model uses the domains D1 and D2 and no refinement for specific areas of Italy are used because Italy is a complex orography country and grid refinements for a specific event can be done only a-posteriori, i.e. after the occurrence of the event."

And few lines below:

"It is noted that data assimilation is performed in the domain D2 (R4) only, and the innovations are transferred to the domain D3 (R1), for the Livorno case, by the two way-nesting. The domain D3 is used for the Livorno case to refine the resolution of the precipitation field over Tuscany and to show the spatial and temporal precision of the precipitation forecast over Tuscany using data assimilation. However, its usage is exceptional because, as stated above, Italy is a complex orography country and grid refinements over specific areas are used only after the occurrence of an event. For these reasons the domain D3 is usually not used in RAMS@ISAC and statistics about the background error aren't available for this grid. The background error in RAMS-3DVar is computed by the NMC method (Parrish and Derber, 1992), which requires a number of simulations (at least two-weeks) verifying at the same time but starting with a lag of 12 h. These simulations are not performed in this paper and background error statistics for the domain D3 are not available. Of course, being lightning assimilated by nudging, they could be assimilated over the domain D3. Nevertheless, to preserve the rationale of the paper, i.e. comparing simulation with or without data assimilation for specific domains, we didn't assimilate lightning over the domain D3.

Of course, being lightning and radar cloud scale observations, their assimilation at higher horizontal resolution is foreseeable in future works. "

Specific questions c) We chose the background error matrix computed for HyMeX-SOP1 because the period was characterized by several convective events over Italy, as documented in Ferretti et al., (2014), while the period preceding the convective events of this paper was characterized by fair weather, typical of the summer Mediterranean season. For this reason, we believe that the matrix for the HyMeX-SOP1 is more representative of convective events compared to the matrix computed for the period of the storms occurrence.  We will write:

"The background error matrix is computed using the NMC method (Parrish and Derber, 1992; Barker et al. 2004) applied to the HyMeX-SOP1 (Hydrological cycle in the Mediterranean Experiment – First Special Observing Period occurred in the period 6 September-6 November 2012; Ducroq et al., 2014). This choice is motivated by the fact that HyMeX-SOP1 contains several heavy precipitation events over Italy and the background error matrix is representative of the convective environment of the cases considered in this paper. In particular, 10 out of 20 declared IOP (Intense Observing Period) of HyMeX-SOP1 occurred in Italy (Ferretti et al., 2014). On the contrary, the period of September 2017, before and after the events selected in this study was characterized by fair and stable weather conditions over Italy and the background error matrix for September 2017 is less representative of the convective environments that characterise the events of this paper."

PDF file with technical corrections:

Considering the pdf file, all corrections will be accepted. There are, however, few points that need a short discussion. They are listed below.

Elements of novelty of the paper: we will highlight better the elements of novelty of this paper in the introduction. We will write:
"This paper presents for the first time the assimilation of the radar reflectivity factor in the RAMS@ISAC model and shows how the assimilation of the radar reflectivity factor works together with lighting data assimilation. Also, this paper shows how accurate in space and time can be the forecast of the precipitation field using cloud scale observations over complex terrain, contributing in this way to a number of works on the same subject."

Comment on Line 276: The frequency bias was not shown to keep the discussion concise. However, important to point out was that the model had a wet bias, especially when assimilating radar reflectivity factor. For this purpose, we introduced the score discussing it to highlight that the model has a wet bias, nevertheless we didn't show any graphs to keep the discussion more concise. In the revised version of the paper the wet bias of the model will be highlighted better and a discussion on how to reduce the wet bias will be added in the "Discussion and conclusions" section (Section 5).

In particular, at the end of Section 4.4 we will write:
"It is finally noted that RAD and RADLI have high POD values for all thresholds, nevertheless their ETS is below that of LIGHT and SAT up to 32 mm/3 h (RADLI) and 42 mm/3h (RAD). This behaviour is caused by the larger number of false alarms given by assimilating radar reflectivity factor compared to those assimilating lightning. This result shows again that the RAD and RADLI configurations have a wet frequency bias. In particular, the frequency bias of RAD and RADLI configuration is about 3 between 20 and 40 mm/3h."

and in the "Discussion and Conclusions " section:

"The wet bias of RAD and RADLI forecast is the main drawback of the results of this paper. To reduce the moisture added by radar and lightning data assimilation further research is needed and different approaches are possible (Fierro et al., 2016). In particular: a) assimilating for a shorter time period (0-6h in this paper); b) reducing the length-scales of the 3D-Var in the horizontal directions to limit the spreading of the innovations (or assuming an innovation equal to zero for grid points without lightning and with zero reflectivity factor observed and simulated); c) reducing the amount of water vapour added to the model (for example reducing the values of A and B constants for lightning data assimilation or relaxing the request od saturation when radar reflectivity is observed in areas where the model has zero reflectivity); d) adding moisture to a shallower vertical level.

It is also noted that a combination of heating and moistening could provide the same buoyancy with less water vapour addition (Marchand and Fulberg, 2014)."

Line 306: There was a typing error into equation 2. The correct equation is:

$$q_v = A q_s + B q_s \tanh(CX)(1 - \tanh(D q_g^t))$$

and $q_g$ is in the last term. We also note that the number of lightning assimilated by the model is larger than that reported in the first submission of the paper for both cases for a mistake we did in typing the numbers. The correct number of assimilated flashes is reported at the start of this answer. The results, however, were obtained using the correct number of flashes.

Line 314: "The check and eventual substitution of the water vapor is performed every five minutes and it is made only in the charging zone (0 °C, -25°C)." To better explain this choice, we will reformulate it adding references to the charging zone:

"The check and eventual substitution of the water vapor is performed every five minutes and it is made within the mixed phase layer zone (0 °C, -25°C), wherein electrification processes are the most active (Takahashi 1978, Emersic and Sounders, 2010; Fierro et al., 2015). It is also noted that some authors use the layer (0 °C, -20°C) (Fierro et al., 2012; 2015), however specific tests for the case studies of this paper did not show significant changes of the simulated precipitation to this choice."

Line 314: the comment is: could you add some more details about this quality control?
We will add two references. We will write:
"The processing chain aims at identifying most of the uncertainty sources as clutter, partial beam blocking and beam broadening. The radar observations are processed according to nine steps detailed in Vulpiani et al. (2014), Petracca et al. (2018) and references therein."

In the following we show the new section Results (Section 4). Note that sections 4.3 and 4.4 are completely new. Note also that minor changes are possible before the eventual submission of the revised version of the paper.

**4. Results**

In this section, we discuss the most intense phase of the Serano case, 03-06 UTC on 16 September, and two VSF forecasts, 00-03 UTC and 06-09 UTC on 10 September, for the Livorno case. The two VSF for Livorno correspond to the most intense phase of the storm in Livorno and to a very intense phase over Lazio region, Central Italy. The aim of the section is to show the notable improvement given to the very short term forecast by the lightning and radar reflectivity factor data assimilation. In the discussion paper two additional VSF are discussed, one for Serano and one for Livorno; also,

the discussion paper shows the behaviour of the scores for a number of rainfall thresholds larger than those shown in this section.

In particular, we consider four types of VSF: a) CTRL, without radar reflectivity factor and lightning data assimilation; b) LIGHT, assimilating lightning but not radar reflectivity factor; c) RAD, assimilating radar reflectivity factor but not lightning; d) RADLI, assimilating both lightning and radar reflectivity factor. A_76 and SAT show the sensitivity of the results to the nudging formulation. Table 3 shows the types of simulations considered in this paper.

[revised manuscript text omitted]

**4.3 Evolution of total water**

Because lightning data assimilation and radar reflectivity factor data assimilation both adjust the water vapour mixing ratio ($q_v$), it is interesting to evaluate the contribution of each data source to the $q_v$ adjustment including in that evaluation the assimilation phase (0-6 h).

For the 3D-Var approach the impact of the contribution of data assimilation on $q_v$ can be done using maps similar to Figure 14b. For example, Fierro et al. (2016), using a 3D-Var approach to assimilate lightning, used the layer averaged $q_v$ between 3 and 10 km to quantify the water vapour added to the WRF model by lightning data assimilation. However, because in this paper lightning are assimilated by nudging, this kind of representation is not practicable because it is difficult to separate the contribution of the nudging from other processes in the evolution of $q_v$.

Fierro et al. (2015) used the total water substance mass (forecasted accumulated precipitation + total hydrometeors and water vapour mass) to quantify the impact of lightning data assimilation by

nudging. In this paper, a similar approach is used. More specifically, we consider the forecasted accumulated precipitation and the total hydrometeors and water vapour mass in the atmosphere averaged over the grid columns. Also, we averaged all VSFs for Serano and Livorno. The evolution of the forecasted accumulated precipitation is shown in Figure 18a, while the evolution of the total hydrometeors and water vapour mass in the atmosphere is shown in Figure 18b.

Considering the Figures 18a and 18b it is apparent that flashes add less water vapour compared to radar reflectivity factor data assimilation and, of course, RADLI has the largest impact. In particular, the total water mass added to the background is 2.5%, 5.7% and 7.4% for LIGHT, RAD and RADLI, respectively. Importantly, the total water substance mass added by RADLI to the background is less than the sum of the total water substance mass added by RAD and LIGHT. This happens because 3D-Var adds water to the background limiting the impact of nudging during the simulation. For example, in an already saturated atmosphere the nudging of Eqn. (2) doesn't have any impact.

Accumulated precipitation accounts for the largest part of the water vapour added to the simulation, similarly to Fierro et al. (2015). At the end the assimilation phase (6h), the evolution of the total water vapour and hydrometeors mass in the atmosphere converges towards the background as boundary conditions propagates into the domain.

*4.4 Sensitivity to nudging formulation*

As stated in Section 3.2, there are limitations when applying the nudging method of Fierro et al. (2012) to RAMS@ISAC. Also, the optimal setting of the coefficients of Eqn. (2) depends on the case study. For these reasons, it is interesting to evaluate the sensitivity of the results to changes in nudging formulation. For this purpose, we show the variability of ETS and POD scores to changes in the A and B coefficients of Eqn. (1). The scores are computed considering all the VSF for the two case studies for different configurations: A_76 has the coefficients A=0.76 and B=0.25; LIGHT has A=0.86 and B=0.15 (default setting), SAT has A=1.01 and B=0; RADLI has A=0.86 and B=0.15 (default setting); CTRL, and RAD are as defined in Table 3.

The scores are computed for the second RAMS@ISAC domains and are shown for the nearest neighbourhood. ETS score (Figure 19a) shows that all configurations assimilating either lightning or radar reflectivity factor alone or a combination of lightning and radar reflectivity factor improves the forecast for all thresholds. RADLI has the best ETS for rainfall intensity larger than 32 mm/3h in line with the results of the three VSF discussed above.

For rainfall lower than 32 mm/3h, the simulations assimilating lightning perform better, because they have less false alarms compared to those assimilating radar reflectivity factor (not shown). From the comparison of LIGHT and SAT with A_76, it is apparent that the latter has the worst score. The comparison between LIGHT and SAT shows mixed results: SAT performs better up to

38 mm/3h, while LIGHT is better for higher thresholds. This result is confirmed by POD, Figure 19b, which shows that SAT performs better up to 32 mm/3h, while LIGHT is better for higher thresholds. A visual inspection of the model output reveals that SAT can generate spurious convection in some areas while missing convection in other areas that are correctly forecast by LIGHT or even A_76, i.e. adding less water vapour to the model because of the different trajectories in the phase space followed by the model using different settings.

Lynn et al. (2015) implemented a method suggested by Fierro et al. (2012) to suppress spurious convection in WRF model. The method compares the lightning forecast during the assimilation period with lightning observations to filter out spurious convection. The application of the methodology on 10 July 2013 improved the forecast of the squall line from Texas to Iowa, which was the focus of the forecast on that day; however, the application of the method to 19 and 21 March 2012 over the US gave mixed results, improving the forecast in the first 6h and worsening it after 6h.

The implementation of this method in the RAMS@ISAC could be used to suppress spurious convection in simulations assimilating lightning, especially SAT.

It is finally noted that RAD and RADLI have high POD values for all thresholds, nevertheless their ETS is below that of LIGHT and SAT up to 32 mm/3 h (RADLI) and 42 mm/3h (RAD). This behaviour is caused by the larger number of false alarms given by assimilating radar reflectivity factor compared to those assimilating lightning. This result shows again that the RAD and RADLI configurations have a wet frequency bias. In particular, the frequency bias of RAD and RADLI configuration is greater than 3 between 20 and 40 mm/3h.

Table 3: Types of simulations performed.

[revised manuscript text omitted]

---

## Author Comment (AC2) · 8 Feb 2019

**Answer to the reviewer #1 comment of NHESS-2018-319.**

First of all we acknowledge the reviewer for the fast and complete review of the paper. In the following we will give answers/actions to improve the paper. Our comments are in red.

Before starting the discussion, we note that, in reviewing the paper, we found two errors: a) the length scale of the background error matrix in the x and y direction varies between 14 and 25 km and not, as stated into the paper, between 20 and 30 km; second the lightning number for each day written into the initial manuscript are wrong. The correct numbers are 82 331 for the 9 September, 291 164 for the 10 September (170 000 is written into the manuscript) and 105 467 for the 16 September (60 000 written in the manuscript). We apologize for these errors. However, the results shown in the paper were obtained using the correct number of flashes and the correct length scales in the background error matrix.

In the first submission we stressed the improvement given by the data assimilation at the local scale on the precipitation VSF (Very Short term Forecast, 0-3h). To highlight this point, we showed the many ways in which the forecast could be improved by the assimilation of lightning, radar or both. For example, the two stages of the Serano case show that the radar (first phase 03-06 UTC on 16 September) or lightning (second phase of the event, 18-21 UTC on 16 September) were the key observation to assimilate in order to improve the precipitation VSF. Also other stages had some specific aspects that we discussed.

Our attempt, however, was not successful, given the comments of both reviewers and the results section (Section 4) underwent a substantial rewriting. In particular, in the revised version of the paper, we will delete the Section 4.1.2 (second phase of the Serano case) and Section 4.2.1 (first case of the Livorno case). The results Section 4.2.1 will be shortly commented in Section 5 (Discussion and conclusions) to highlight that there is space for improvement. Following the comments of the reviewer #2, the scores of the phases commented into the paper will be put in three tables (Tables: 4-6) for specific thresholds (1, 6, 10, 20, 30, 40 mm/3h and, for Livorno, also 50 mm/3h). This will limit the number of precipitation thresholds considered but will increase the readability of the paper.

The space gained by deleting the two sub-sections stated above will be used to extend the discussion about the methods of assimilating lightning and radar in the RAMS@ISAC and to add two short sections to the result paragraph. In particular, we will extend the section "Lightning data assimilation" to include a discussion of the useful comments raised by the Reviewer#1, we will extend the section "Radar data assimilation" to show an example of 3D-Var assimilation of reflectivity factor (this should also answer to few comments of the Reviewer #1). A draft of these revised sections is reported at the end of this answer.

Finally, we will add a section (Section 4.3) to show how the lightning and radar data assimilation works together, presenting the evolution of the total water mass averaged for all VSF of the two cases and including in this discussion the assimilation stage, as well as sensitivity tests for the nudging formulation of lightning data assimilation (Section 4.4). The latter point requested new simulations with different model settings (see Table 3 at the end of this answer).

A draft of the new Results section (Section 4) is shown at the end of this answer. This could not be the final form because minor changes are still possible.

Summary: The authors utilize a cloud-scale functional relationship between lightning and water vapor mass mixing ratio published in the literature and applied it to a homegrown 3DVAR framework at the convection-allowing scale to evaluate the analysis and short term forecast of two selected high impact weather events over Italy.

Recommendation: reject and, eventually, re-submit.

Main Comments:
While the manuscript could eventually offer some merit for this journal, I found the analysis generally very rudimentary with the authors going at length in describing in excruciable level of details individual figures/panels in a repetitive and redundant manner without distilling the content into concise arguments/hypotheses. Given its repetitive nature, the entire results section could, in fact, easily be condensed into a 2-3 pages. Most importantly, the manuscript (hereafter, m/s) lacks rigor and rationales for the set ups and methods put forth for each, respective DA approaches. Salient Major issues are itemized below.

(1) As far as the scientific content is concerned, the core ideas and notions of this lightning data assimilation (LDA) method are conceptually similar to those from many existing studies, which fundamentally aim at promoting convective development through the introduction of latent heating within a prescribed neighborhood region/column centered at observed lightning locations. Past works from Benjamin et al. (2004), Alexander et al. (1999), Chang et al. (2001), Papadopoulos et al. (2005), Pessi and Businger (2009), have used empirical relationships between lightning-rainfall rates-latent heating or lightning- reflectivity rates-latent heating [e.g., in the HRRR]. Following a similar idea, recent works such as Machand and Fuelberg (2014), Lynn et al. (2015), Lynn (2017), Fierro et al. (2012; 2014, 2015), Wang et al. (2017, 2018) proposed LDA means that essentially boost the local thermal buoyancy where lightning is observed. A very limited portion of these techniques, however, offer alternative approaches to address spurious convection (i.e., removal) – which is a far more challenging problem to tackle. For completeness and given the relatively limited advances in LDA relative to radar DA, the authors should do a better job in discussing and including all the aforementioned references in their text. I was in fact astonished to notice that the integrity of the Results section in section 4 is completely devoid of references to previous works.
In particular, since they opted to borrow an LDA method from one of these investigators, comparisons with their study should be performed more systematically throughout the m/s. For instance, the works of Federico et al. 2017b is invoked when referring to multi-day forecast statistics using the Fierro et al. method without mentioning that, such a study, was already conducted by the same author over a larger domain and using nearly three times more forecast days/cases (Fierro et al. 2015 study). Given this omission, their study (Federico et al. 2017b) inadequately state that such multi-day statistics for this LDA have never been conducted. In a similar manner, it is of relevance to underline whenever appropriate that, in this work: (i) radial velocity is not included (specify why), (ii) only cloud-to-ground lightning data are considered and (iii) spurious convection is not addressed. In the light of (i) and (ii), one on the recent studies they

cite (Fierro et al. 2016) not only assimilated level II radar data (radial velocity + reflectivity factor) but used total lightning data. This needs to be clearly stated, for completeness (Cf comment 3 below for rationales).

In the revised version of the paper we will extend the discussion of the LDA in the introduction in order to include all the above papers.
Considering the other points:

(i)     We are working on the assimilation of the radial velocity but the operator is not yet implemented in the 3D-Var. Also, while the reflectivity factor measured by the radar network is operationally available, the product of radial velocity is under development. At the moment, it needs further research to solve some issues (complex orography, operations of the radars not optimal for the Doppler retrieval and others). For these reasons, the attention was on the assimilation of reflectivity factor. These motivations will be discussed in the revised version of the paper in Section 3.3 by writing:

"Radial velocity is not assimilated in the RAMS@ISAC model because the operational product of radial velocity needs research to solve issues (complex orography, operation of radars non optimal for Doppler retrieval, not homogeneous coverage of the country), and it is not available for assimilation. Also, the implementation of radial velocity data assimilation is under development in RAMS-3DVar and it is not available for testing. For these reasons, we didn't consider the assimilation of radial velocity in this work. "

Considering the point (ii) in the paper we will write that total lightning are assimilated, not only CG. For these events the fraction of IC strokes to the total number of strokes detected by LINET is about 30% (22% on 09 September, 30% on 10 September and 35% on 16 November). There are cases when the IC strokes recorded by LINET are more than 50% of the total number of stokes over Italy. In general the Section on LDA will be extended to consider this point and others; (iii) The spurious convection is not considered by the LDA but it is considered in the assimilation of radar reflectivity factor. We will specify better this point in future version of the paper, but the comment is already present in the first submission version.

(2). In term of DA methodology, I found one major drawback, which is never discussed, nor evaluated. Given that both the LDA and their "RAD" experiment make adjustments to the relative humidity (RH) field, it is expected that both techniques will overlap in their adjustments over all the (many) grid points characterized by observed lightning flash rates exceeding zero. This is because changing RH is equivalent to adjusting Qv as RH ~ Qv/Qv_saturation. A more self-consistent DA approach would adjust the pseudo- observations for the Qv or RH field in a manner that eliminates any possibility of overlap during the minimization. Toward that end, the authors should include soundings and/or horizontal cross sections of RH/Qv that shows, quantitively, how the RH field is adjusted by each respective DA approach (radar vs lightning).

Second, given that lightning is a cloud-scale observation, I cannot find any justifications for not conducting the 3DVAR analysis on the innermost, higher resolution domain. Instead, the method minimizes the cost function on the intermediate domain and, later, projects the innovations on the coarser-scale domain. This needs to be addressed.

First: we will add a complete new section (Section 4.3) to address this point. In this section we will show the evolution of the accumulated precipitation and total water mass in the atmosphere (i.e. water vapour mass+mass oh hydrometeors) as a function of time (including the spin-up period). A draft of this new section is attached at the end of this answer.

Second:

Data assimilation is not performed on domain D3 (R1) because we don't have background error statistics for this grid.

Background error statistics for the domain D2 are computed by the NMC method, which, for this paper, is based on HyMeX-SOP1 simulations. The Appendix A and B of Federico (2013) shows the detail of the application of the method, which require a number of simulations (see also Barker et al., 2004 for the general discussion). Because the application of the domain D3 is exceptional the background error matrix was not computed for this domain and no data assimilation was performed.

Of course, this limitation is only for radar reflectivity factor because lightning are assimilated by nudging. Nevertheless, we could not reproduce the rationale of the paper, i.e. compare simulations with or without data assimilation for a specific domain, assimilating lightning in the innermost domain and for this reason we assimilated flashes over the D2 only.

In the paper we will specify better the role of the domain D3 and the reason for not assimilating lightning and radar reflectivity factor over the domain D3.

We will write in section 3.1

"The third domain covers the Tuscany Region, has 4/3 km horizontal resolution (R1), and it is used for Livorno to represent with higher spatial detail the precipitation field over Tuscany and to show better the precision of the rainfall VSF using data assimilation at the local scale. The fine structures of the precipitation field are smeared out over Tuscany using only domains D1 and D2. The operational implementation of the RAMS@ISAC model uses the domains D1 and D2 and no refinement for specific areas of Italy are used because Italy is a complex orography country and grid refinements for a specific event can be done only after the occurrence of the event."

And few lines below:

"It is noted that data assimilation is performed in the domain D2 (R4) only, and the innovations are transferred to the domain D3 (R1), for the Livorno case, by the two way-nesting. The domain D3 is used for the Livorno case to refine the resolution of the precipitation field over Tuscany and to show the spatial and temporal precision of the precipitation forecast over Tuscany using data assimilation. However, its usage is exceptional because, as stated above, Italy is a complex orography country and grid refinements over specific areas are used only after the occurrence of an event. For these reasons the domain D3 is usually not used in RAMS@ISAC simulations and no statistics about the background error are available for this grid. Because lightning are assimilated by nudging, they could be easily assimilated over the domain D3. Nevertheless, to preserve the rationale of the paper, i.e. comparing simulation with or without data assimilation for specific domains, we didn't assimilate lightning for domain D3.

Of course, being lightning and radar cloud scale observations, their assimilation at higher horizontal resolution is foreseeable in future works. "

Third, the radius of influence/decorrelation length scale chosen for radar reflectivity factor (50 km) is far too large for convective scale applications and would incur unrealistically large amount of Qv mass added into the domain – which will undoubtedly yield to spin-up issues and the generation of convective-scale gravity waves that will degrade longer term (>= 3h) solutions (please provide plot of perturbation pressure in your response). In that regard, the authors should indicate and contrast the total amount of Qv mass added by RAD and LIGHT.

The 50 km length is not a distance to spread the innovation introduced by radar reflectivity factor data assimilation. It represents a search radius to compute the pseudo-profile of relative humidity used in 3D-Var. A discussion about this point will be introduced in the new section on radar reflectivity factor data assimilation.

In particular we will write:

"It is important to point out that the 50 km length-scale of the above step doesn't represent the horizontal correlation length-scale of the background error, which determines the horizontal

spreading of the innovations in the 3D-Var data assimilation (the latter length-scale is between 14 and 25 km depending on the level). The 50 km length-scale is used to set a square for computing the pseudo-profile of relative humidity (Eqn. (2)). This profile is given by a weighted average whose weights are determined by the agreement between the simulated and observed reflectivity factor. The larger the agreement the larger the weight. This distance seems appropriate because the spatial error of meteorological models in simulating meteorological features, for example fronts, can be of this order. The control simulation for the two events considered in this paper confirms this choice."

(3). In the context of forecast improvements, the Qv-based method they borrowed/adapted was scaled for total lightning data (> 50% detection efficiency of intra-cloud [IC] flashes). I was surprised to find that absolutely no information on the detection efficiency and geolocation accuracy of the lightning network used (LINET) is provided in the text [no figures either]. Given the large area covered by this study, it is thus very likely that the geolocation accuracy of this network remains very poor for low amplitude flashes and for all flashes over oceanic regions. Given the low sferics amplitudes of IC flashes, the VLF portion of the sensor will miss nearly all these flashes, while the VHF portion only is able to detect some of the IC flashes within a few tens of kilometers away from the station [e.g., Rison, MacGorman works]. Thus, it is relevant to state and underscore that LINET only detects a very small portions of the total IC flashes in the study domain (likely < 5%). Motivation for scaling the F12 method for IC flashes (in lieu of cloud-to-ground [CG] flashes), lies in the well-documented finding that, in contrast to CGs, ICs are well correlated with thunderstorm kinematic and microphysical evolution (updraft strength, updraft volume, graupel mass etc, see Wiens et al. 2005, Schultz et al. 2011 among many others). CGs, on the other hand, were found to be correlated with the descent of reflectivity cores and the onset of the demise of the storm's updraft core [MacGorman and Nielsen 1991, MacGorman et al. 1989, Rutledge and Lang's seminal works etc]. Not surprisingly, ICs were found to lag CG by an average of 15 min [see one of the recent MacGorman study]. Moreover, Boccippio et al. 2001 and Medici et al. 2017 found that in deep continental convection, IC flashes always outnumber CGs by a ratio sometime exceeding 10:1. Based on these facts, it becomes clear why the Fierro method emphasized the use of IC flashes [or total lightning] for their application. Further motivation arises from the recent successful launches of the GLM instrument aboard GEOS-16/17, which will provide continuous day/night coverage of total lightning at ~90% detection efficiency (DE) over a large domain covering the Americas (Gurka et al. 2006; Goodman et al. 2012, 2013, Rudlosky et al. 2018). Note that GLM will provide flash extent information of lightning, while the metric derived from the (limited) point flash data in this study can only provide a very rough surrogate for CG flash location density at best. Similar space-borne technology to detect lightning have been developed by China (Feng-Yun-4, yang et al. 2016) with these data being assimilated in recent works by Wang et al. (2017, 2018) – which were never referenced either. Apart from their propensity to detect total lightning at a high DE, the chief advantage of this technology lies in its ability to retrieve lightning over remote oceanic regions.

LINET has been started and used operationally since 2004. Since then, more than 100 publication have appeared that give evidence about both DE and LA. In particular, since the beginning in 2004 LINET exhibited a statistical average location accuracy of some 100 m. Because a minimum of 5 sensor reports are exploited for each stroke solution, the LA does not deteriorate within several 100 km from a sensor. Thus, the LA is excellent all over the present study region.

LINET Europe comprises more than 200 sensors and provides more extensive stroke data than any other VLF/LF system in the region.

LINET detects and records stroke signals down to currents of a few kA (CG normalization). This is the reason why LINET ranges are large enough to exploit >=5 sensors for geolocation without reducing the typical baselines of 250-300 km. The resulting DE is good enough to detect any CG. Over the Mediterranean the stroke DE diminishes due to larger baselines. However, the flash detection is less sensitive because of the stronger strokes that characterize a flash.

Like any other VLF/LF system signals are recorded whether CG or IC. Thus, the detected IC portion is certainly not lower than in any other VLF/LF system. As a consequence, total lightning is reported at least as efficient than in any other VLF/LF system, and will be beneficial for the purposes in the present paper.

IC discrimination of LINET is based on TOA analysis. The advantage is a unique discrimination when the detection geometry is within certain ranges; the disadvantage is decreasing discrimination power when the distance to the closest sensor become too large, because of too small TOA differences between CG and IC at the same 2D location. Thus, over water far from land the identified IC fraction decreases, though total lightning counts remain relevant.

We emphasize that the time evolution of IC reports in considered area (not too far from land) signify very well the change of meteorological condition, especially with respect to severe weather. Note that the relatives changes (including lightning jumps in rate and altitude) are indicative, without the need to have absolute event numbers. See for example ref. "Thunderstorm Nowcasting" in Met. Tech. Int., Sept. 2017, p.109-112.

It is true that leader steps signify discharge processes (see, e.g., well-known LMA results). However, it is well-proven that VLF/LF detects pulses from IC activity that are very similar to CG strokes; this is why CG-IC discrimination is very challenging for VLF/LF systems. We think, though, that any VHF issue is not relevant here, because there is no large-scale VHF system that covers Italy and the surrounding sea with baselines of a few 10 km.

Observations from global networks or satellites may be a point of future concern, but do not represent any focus in the present paper; also, IC discrimination is either not yet possible or poor. It may be mentioned that GLM lightning data is not yet an issue; interestingly, Eumetsat/NASA on behalf of NOAA have selected LINET to carry out the first evaluation of the new lightning data source. This has been communicated in Science Team Meetings and conferences (see GLM Cal/Val 2017 Ground Validation Field Campaign 2017).

These points will be discussed in the new section on lightning data assimilation (Section 3.2). A draft of this section is shown at the end of this answer, but it is still incomplete.

(4) The following key information pertaining to the respective DA methods are missing/never discussed:
(a) What are the background/observation errors for reflectivity/lightning? (b) What statistics are used for model error ?
Lightning are assimilated by nudging and no error is associated with them. The error matrix for model error will be clarified in the section on the radar reflectivity factor data assimilation (see the attached draft of Section 3.3).

(c) How is the adjoint for the lightning data assimilation operator derived ?
The derivation of the adjoint of lightning data assimilation was performed using two case studies of the HyMeX-SOP1 (unpublished work) as commented in Federico et al., (2017a). We will comment about this point in the new section on lightning data assimilation. Also, in future versions of this paper we will add a new section (Section 4.4) to show the sensitivity of the rainfall VSF score (POD and ETS) to the nudging formulation.

(d) What assumptions are made for grid points with zero lightning or zero reflectivity observations ? Does the radar DA or LDA treat those as missing observations or equate those to the background values to reduce spread ?
Lighting are assimilated by nudging and this comment doesn't apply. In the case of radar, grid points with zero reflectivity factor and zero simulated reflectivity factor are assumed missing observation, and the innovations can spread freely. Again this will be clarified in future versions of the paper (see the Section 3.3 draft).

(e) What Gaussian decorrelation length scales are assumed for each observation ? Please specify/justify/explain. How would the selection of a given length scale value, influence the results ?

The observation error matrix for radar reflectivity is diagonal (this was already stated in the first submission of the paper). We acknowledge that the sensitivity tests proposed by the reviewer are interesting, nevertheless they will be left for future studies. The importance of this point will be discussed shortly in the paper. We will write:

"The observation error matrix R in Eqn. (4) is diagonal and observations' errors are uncorrelated. This choice is partially justified by under sampling the radar reflectivity factor observation by choosing one point every five grid points in both horizontal directions of the radar observations Cartesian grid (Rohn et al., 2001) . However, correlation observations errors have significant impact on the final analysis, as shown for example in Fierro et al. (2016), and different choices of the matrix R will be considered in future studies.
The value of the elements on the diagonal of R depends on the vertical level and are 1/4 of the diagonal element of the $B_z$ matrix at the corresponding height. By this choice, we give more credit to the observations than to the background and analyses strongly adjust the background towards observations."

(f) Is spurious convection addresses by either DA method ? Please elaborate.

Yes, in the radar reflectivity factor data assimilation, but not in the lightning data assimilation. The point is already present in the discussion paper, but it will be better clarified in future versions of the paper in the section dedicated to the radar data assimilation.

(g) Does the variational minimization set use a multi-scale approach ? If yes, what influence radii are chosen and how many cycles ?

We don't use the multi-scale approach. This will be clarified in the paper in the section dedicated to the radar data assimilation.

(5) Why did the authors not include the fractions skill score FSS as the main evaluation metric for their forecast? Several works have posited that, in contrast to ETS, FSS does not penalize displacement errors as much and, arguably, FSS offers a more accurate measure of skill on convection-allowing grids (Mittermaier et al. 2013).
Additionally, more recent studies evaluating forecast performance have been making usage of the so-called performance diagrams, which conveniently merge several key contingency table elements into one single diagram (Roebber 2009). The authors should show such diagrams to provide a more complete and succinct view of the overall forecast performance of the case they selected.

Considering POD and ETS gives the possibility to show the many facets of a forecast, and this, in our opinion, is important. These scores are also widely used in the bibliography and this make the results of this paper comparable with other papers. We, of course, acknowledge that there are other interesting measurements of the model performance, as FSS, that could be considered. Taking into consideration this comment and the comment of the reviewer #2 about the score we propose the following solution: we consider three neighborhood radii to take into account for displacement errors; nearest neighborhood (as in the first submission), 25 km and 50 km. We will put the scores in three tables (Tables 4-6 attached at the end of this answer) following a remark of the second reviewer.

(6) The case studies selected are cherry-picked given the confession that CTRL generally failed to provide reasonable forecast estimates of precipitation for both cases herein. For good measure, fairness and to better underscore the performance of the DA method, the authors should show the results for one case in which CTRL did not perform well and contrast it to one case where CTRL did preform reasonably well.

The events were selected because they were missed by several forecasts and, for this reason, they are challenging. Moreover, they had important consequences because nine people died and damage to properties was extensive. We will stress better this point in the introduction by writing:

"The forecast of severe events at the local scale still remains a challenge because of the multitude of physical processes involved over a wide range of scales (Stensrud et al., 2009). The Serano case, being localised in space, poses challenges in forecasting the exact position and timing of convection initiation; the Livorno event involves the interaction between a high impact storm with the complex orography of Italy, which is difficult to simulate at the local scale. For the above reasons the forecast of both events was challenging, as confirmed by the poor forecast of RAMS@ISAC. The difficulty to forecast timely and accurately the precipitation fields of the two cases is the main reason for choosing them as test cases for testing the lightning and radar data assimilation."

(7) The authors omit to mention that the degradation of the forecast at >= 3h is mainly due to saturation of the model solution by errors and biases within the initial / boundary conditions derived from large scale models or re-analysis datasets. This needs to be shown for both cases, especially given the unrealistically large (50 km) decorrelation length scale used for radar reflectivity factor.
Ok we will consider this point in the revised version of the paper. However, model errors plays an important role in the degradation of the forecast in addition to IC/BC. Again 50 km is not the decorrelation length scale for radar reflectivity factor.

(8) Title: Revise to include that: (i) primarily CG flashes are assimilated and (2) the model vehicle is RAMS.

We will include in the title that RAMS@ISAC is the model vehicle. As stated above, the IC flashes for the case studies considered in this paper is about 30%, which is not a small fraction of the total lightning. The discussion on the method assimilating lightning will be widened to consider this and other points.

Because these issues are collectively substantial and would require thorough rewriting of the manuscript in many places, I opted not to dwell on editorial comments for the time being. Additionally, the level of English remains, in my view, unacceptable for publication.

We will revise the English of the paper, also according to the suggestions of the reviewer 2 in the PDF file. The copy-editing service of the journal will also improve the quality of the English.

Figures:
Figures 5, 6, 8, 12a, 13a, 14a, 15a, 16a, 17a, 18a: The use of colored dots makes it very difficult to effectively compare the observations with those of the simulations: For consistency, either both sets of plots should show colored dots or shaded contours. For lightning, the authors should effectively show the gridded lightning data that were used to create the Qv or RH pseudo-observations.

Ok. It is always difficult to choose the right representation of the precipitation field when comparing model output with raingauges. We acknowledge that the solution suggested by the reviewer is a good one, however we also like our representation because: a) rainfall at the raingauges is not interpolated, avoiding in this way errors introduced by the interpolation process; b) the rainfall predicted by the model shown as a field gives the possibility to see the behavior of the model also in parts uncovered by raingauges. We propose the following solution: a) redraw the RAMS@ISAC rainfall field changing the colorbar to match exactly the raingauges colorbar; b)

adding the representation suggested by the reviewer as supplemental material to the paper (Figures S1-S3 at the end of this answer); c) recalling the supplemental material when discussing the second VSF of Livorno to highlight the wet frequency bias when assimilating radar reflectivity factor (see Figure S3 at the end of this answer).
Ok for the Figures about lightning. They will be redrawn according to the reviewer remark.

Additional comments:
General comment: What is the main rationale for using a model that is marginally known by the community (RAMS) versus a more commonly used, battle tested, publicly available model such as WRF-ARW ? The authors not only seem to re-invent the wheel here but render any potential future work dedicated to reproducibility of the results - to the least - very challenging.

RAMS@ISAC is used/maintained/developed at ISAC-CNR since several years (and it is also operational over Italy since 2000, in different versions/adaptations etc), and it is important for us to test our tool for challenging cases, as those considered in this work.
Also, we are WRF users too (see, for example, Avolio and Federico, 2018) and for the cases of this paper no specific differences were found for the performance of WRF and RAMS@ISAC models (using the same initial and dynamic boundary conditions). The performance of WRF model for the Livorno case is shown, for example, in Ricciardelli et al. (2018) and the reviewer can see that the comments given in this paper about the performance of RAMS@ISAC for the most intense phase of the Livorno case can also be applied to the WRF model (see specifically their Figures 11 and 12 for the most intense phases in Livorno). Consider also that Ricciardelli et al. used ECMWF IC/BC, which are different from that used in this paper. So, the results of this paper could be even more valuable because they are "more general" and not linked to the specific modelling tool.
We will add a reference to the above cited paper and a short discussion in Section 5 (Discussion and Conclusions).

(1) Bottom, page 2: what are "conventional data" ? Why are radial velocity data not used ? Line 70: the main advantage of using 3DVAR vs 4DVAR, EnKF or hybrid methods lies in their already low computational burden. Thus, I do not agree with this justification. Also, variables are not "perturbed"; but adjusted by VAR methods.

For radial velocity we already answered. We will change the paper according to the reviewer suggestion for the other parts of the comment.

(2) Pages 3 and 4: Please refer to Major Comments 1 and 3. Lines 105: Given that "Federico et al. (2017a) implemented the methodology of Fierro et al. (2012) ...", how come on line 112 "We use the method of Federico et al. (2017a) to assimilate lightning..." ? Please revise accordingly.
Ok for this comment. We will add the reference to Fierro et al. (2012). The comment of line 112 come from the fact that we intended to cite the adaptation of the methodology, that is discussed in Federico et al. (2017a).

(3) Line 124: c.f. end of Major Comment 1.
Ok.

(4) Line 240: RAMS used diagnostic relationships (vs explicit) to forecast lightning as it does not explicitly solves for the 3D electric field. Line 243: "Fourth"
For the first comment we wrote: "Second, it predicts the occurrence of lightning following the diagnostic methodology of Dahl et al. (2012),…."

(5) Line 290: Delete equation set as these are considered basic/common knowledge.

In some papers, where we omitted the equations, we had the opposite comment. However, for this paper, to reduce length and to give more space to the important points raised by the reviewer the equations will be deleted.

(6) Section 3.2, lines 300-312: Explicitly state and indicate that equation (2) is from Fierro et al. (2012, 2015) and not from Federico et al. Line 305: Please explain the rationales behind the choices of these constants: In particular, how are the forecast metrics affected for a 20% change in A, which has been shown to exhibit the most notable influence on the forced convection?
Ok for the reference. The functional form is that of Fierro et al. (2012, 2015), but the coefficients were adapted for RAMS@ISAC as shown in Federico et al (2017a). In Federico et al. (2017a) it is clearly stated that the method is that of Fierro et al. (2012), the only difference being the adaptation to RAMS@ISAC model. Sensitivity tests to the nudging formulation will be shown in Section 4.4.

(7) Line 316-317: c.f. Major Comment 2.
Ok.

(8) End of page 11: c.f. Major Comment 2
Ok.

(9) Line 356: do the authors refer to the LFC or the LCL, (which may I add is an idea borrowed from Marchand and Fuelberg 2014 and Fierro et al. 2016). What is the top of the adjustment layer for lightning ? Please elaborate.
It is the LCL. The idea is of Caumont et al. (2010), we didn't add the reference to this point of the paper because the whole methodology is taken from Caumont et al. (2010), already cited several times. The top adjustment for lightning is -25°C. However, this is already stated in the paper (Lines 314-315 "The check and eventual substitution of the water vapor is performed every five minutes and it is made only in the charging zone (0 °C, -25°C).").

(10) Line 410 and elsewhere. This is similar to the results of Fierro et al. 2016. C.f. Major Comment 1. Please establish comparisons with previous works throughout the manuscript.
Ok.

(11) Line 669: This statement is incorrect. The DE of ground based sensors levels off very rapidly with distance from land. This is where space-borne lightning detection systems such as the GLM or Feng Yun-4 can fill the gap.
Ok, however the good coverage of the LINET network for some important areas, as between Corsica and Italian mainland (both Liguria and Tuscany) makes this point "less problematic" for the Livorno case.

(12) Lines 716-725: c.f. Major Comment 1.
Ok.

Hereafter we show the new sections on lightning data assimilation (Section 3.2) on radar data assimilation (Section 3.3) and the new results section (Section 4).

*3.2 Lightning data assimilation*

Lightning data are provided by LINET (LIghtning detection NETwork; Betz et al., 2009; www.nowcast.de) which has more than 500 sensors worldwide with the greatest density over Europe (more than 200 sensors). The network has a good coverage over Central Europe and

Western Mediterranean (from 10 W to 35 E and from 30 N to 60 N). The area of good coverage includes the region considered in this paper.

LINET exploits the VLF/LF electromagnetic bands and provides measurements of both IC (intra-cloud) and CG (cloud to ground) discharges. IC strokes are detected as long as lightning occurs within 120 km from the nearest sensor thanks to an optimised hardware and advanced techniques to process the data (TOA-3D, Betz et al., 2004). According to Betz et al. (2009), LINET has a location accuracy of 100 m (since 2004) for an average distance of 200 km among the sensors verified by strikes into towers of known positions.

The good performance of the LINET network and its ability to detect IC strokes is shown in Lagouvardos et al. (2009) for a storm in southern Germany, while the good performance over Italy, including both CG and IC strokes, is discussed in Petracca et al. (2014).

The lightning data assimilation scheme is that of Fierro et al. (2012; 2014) and uses the total lightning, i.e. intra-cloud plus cloud to ground flashes.

The method starts by computing the water vapour mixing ratio $q_v$:

$$q_v = Aq_s + Bq_s \tanh(CX)(1 - \tanh(Dq_g^\alpha))$$

(1)

Where coefficients are set to A=0.86, B=0.15, C=0.30, D=0.25, $\alpha$=2.2, $q_s$ is the saturation mixing ratio at the model atmospheric temperature, and $q_g$ is the graupel mixing ratio (g kg$^{-1}$). $X$ is the number of total flashes (IC+CG) falling in a grid box of domain D2 (R4) in the past five minutes. The mixing ratio $q_v$ of Eq. (1) is computed only for grid points where flashes are recorded. More specifically, for each grid point we consider the number of flashes falling in a grid box centred at the grid point in the last five minutes. The mixing ratio of Eqn. (1) is compared with that predicted by the model. If the mixing ratio of Eqn. (1) is larger than the simulated one, the latter is changed with the value given by Eqn. (1), otherwise the modelled mixing ratio is left unchanged. This method can only add water vapour to the forecast.

The check and eventual substitution of the water vapor is performed every five minutes and it is made within the mixed phase layer zone (0 °C, -25°C), wherein electrification processes are the most active (Takahashi 1978, Emersic and Sounders, 2010; Fierro et al., 2015). It is also noted that some authors use the layer (0 °C, -20°C) (Fierro et al., 2012; 2015).

The scheme of Fierro et al. (2012; 2014) was adapted to RAMS@ISAC in Federico et al. (2017a). In particular, the coefficient C of Eqn. (1) was rescaled from that of Fierro et al. (2012) considering the different spatial and temporal resolution of the gridded lightning data; then the coefficient C was tuned (increased) by trials and errors considering two case studies of HyMeX-SOP1 (15 and 27 October 2012). The C constant was adapted subjectively considering two opposite requests:

increasing the hits and minimising (or not increasing substantially) the false alarms. POD and ETS scores were considered as metrics for this purpose. Then, Eqn. (1) was applied to twenty case studies of HyMeX-SOP1 giving a statistically significant (90, or 95% depending on the rainfall threshold) improvement of the RAMS@ISAC precipitation VSF (3h).

Nevertheless, an exhaustive statistics on the performance of rainfall VSF to the nudging formulation in RAMS@ISAC is missing and further studies are needed in this direction. Also, the optimal choice of the coefficients A, B, C, D and $\alpha$ are case dependent.

In addition to the above issues there is another important point for the application of the Fierro et al. (2012) method to RAMS@ISAC. Fierro et al (2012) applied the method using the ENTLN network, which has a detection efficiency (DE) greater than 50% for IC over Oklahoma, where the ENTLN data were used. The emphasis on IC flashes in the set-up of Fierro et al. (2012) method is given because observational and model studies have provided evidence that IC flashes are better correlated than CG flashes with various measures of intensifying convection (updraft strength, volume, graupel mass flux etc.; Carey and Rutledge 1998; MacGorman et al. 2005; Wiens et al. 2005; Fierro et al. 2006; Deierling and Petersen 2008; MacGorman et al. 2011). For this reason methods that use both IC and CG flashes performs better than those using CG, the latter being correlated with the descent of reflectivity cores and the onset of the demise of the storm' s updraft core.

A direct DE for IC strokes cannot be reliably compared with that of ENTLN, because the area is different and the technical details about IC detection remain unclear (type of signals, VLF/LF or VHF, discrimination IC-CG). An analysis for the case studies shows that IC strokes are about 30% of the total number of strokes reported. Also, the fraction of IC strokes to the total strokes depends on the position. For example, for the Serano case, the fraction of IC strokes detected by LINET over the area hit by the largest precipitation is more than 50% while over the Adriatic Sea it decreases to 10%-15%.

For all the above reasons there are limitations to the application of Eqn. (1) to RAMS@ISAC 
[revised manuscript text omitted]
 toward observations. We could choose to give more credit to the background compared to the observations, nevertheless the poor performance of the control forecast for the selected cases justifies this choice. The background error matrix is computed using the NMC method (Parrish and

Derber, 1992; Barker et al. 2004) applied to the HyMeX-SOP1 (Hydrological cycle in the Mediterranean Experiment – First Special Observing Period occurred in the period 6 September-6 November 2012; Ducroq et al., 2014). This choice is motivated by the fact that HyMeX-SOP1 contains several heavy precipitation events over Italy and the background error matrix is representative of the convective environment of the cases considered in this paper. In particular, 10 out of 20 declared IOP (Intense Observing Period) of HyMeX-SOP1 occurred in Italy (Ferretti et al., 2014). On the contrary, the period of September 2017, before and after the events selected in this study was characterized by fair and stable weather conditions over Italy and the background error matrix for September 2017 is less representative of the convective environments that characterise the events of this paper. It is also important to highlight that the dependence of the results on the choice of the background error matrix is mainly determined by the choice of the horizontal and vertical length scales of the background error correlation because the observation error matrix (**R**) is ¼ of the background error at the same level to give more credit to the observations than to the background at this level (comparison at the levels above an below that of Figure 14a shows that the method was able to dry the model west of Sardinia).

Because it is the first time that the assimilation of radar reflectivity factor in RAMS@ISAC model is shown it is useful to discuss an example of analysis. We select the analysis for the Livorno case at 06 UTC. The observed CAPPI at 3km above sea level is shown in Figure 10b. The corresponding CAPPI simulated by the background is shown in Figure 14a. In general, the comparison between simulated and observed reflectivity factor shows the difficulty of the model to represent convection properly. In particular, the model is able to represent the convection over Northern Italy but it has poor performance over Sardinia, south of Sicily and over Central Italy. The difference between the analysis and background relative humidity after and before the analysis is shown in Figure 14b (absolute values less than 1% are suppressed in the figure for clarity). Both positive (convection enhancing) and negative (convection suppressing) adjustments can be found. Over Central Italy, Sardinia and South of Sicily relative humidity is increased because the model doesn't simulate the observed reflectivity (Figure 10b). Over northern Italy the model is partially dried for two different reasons: over northwest of Italy because RAMS@ISAC simulates unobserved reflectivity, over north and northeast of Italy because the model simulates larger values of reflectivity factor compared to the observations. The RAMS-3DVar is able to dry the relative humidity field north of Corsica island, where the RAMS@ISAC predicts unobserved reflectivity, while RAMS-3DVar didn't suppress the unobserved convection west of Sardinia because the pseudo profiles computed over this area weren't appreciably drier than the

background. Cross correlations among variables are neglected in this study and the applications of the RAMS-3DVar affects the water vapour mixing ratio only.

Because the lightning data assimilation perturbs the water vapour mixing ratio, it follows that the data assimilation presented in this study changes only this parameter.

**4. Results**

In this section, we discuss the most intense phase of the Serano case, 03-06 UTC on 16 September, and two VSF forecasts, 00-03 UTC and 06-09 UTC on 10 September, for the Livorno case. The two VSF for Livorno correspond to the most intense phase of the storm in Livorno and to a very intense phase over Lazio region, Central Italy. The aim of the section is to show the notable improvement given to the very short term forecast by the lightning and radar reflectivity factor data assimilation. In the discussion paper two additional VSF are discussed, one for Serano and one for Livorno; also, the discussion paper shows the behaviour of the scores for a number of rainfall thresholds larger than those shown in this section.

In particular, we consider four types of VSF: a) CTRL, without radar reflectivity factor and lightning data assimilation; b) LIGHT, assimilating lightning but not radar reflectivity factor; c) RAD, assimilating radar reflectivity factor but not lightning; d) RADLI, assimilating both lightning and radar reflectivity factor. A_76 and SAT show the sensitivity of the results to the nudging formulation. Table 3 shows the types of simulations considered in this paper.

[revised manuscript text omitted]

*4.3 Evolution of total water*

Because lightning data assimilation and radar reflectivity factor data assimilation both adjust the water vapour mixing ratio ($q_v$), it is interesting to evaluate the contribution of each data source to the $q_v$ adjustment including in that evaluation the assimilation phase (0-6 h).

For the 3D-Var approach the impact of the contribution of data assimilation on $q_v$ can be done using maps similar to Figure 14b. For example, Fierro et al. (2016), using a 3D-Var approach to assimilate lightning, used the layer averaged $q_v$ between 3 and 10 km to quantify the water vapour added to the WRF model by lightning data assimilation. However, because in this paper lightning are assimilated by nudging, this kind of representation is not practicable because it is difficult to separate the contribution of the nudging from other processes in the evolution of $q_v$.

Fierro et al. (2015) used the total water substance mass (forecasted accumulated precipitation + total hydrometeors and water vapour mass) to quantify the impact of lightning data assimilation by nudging. In this paper, a similar approach is used. More specifically, we consider the forecasted accumulated precipitation and the total hydrometeors and water vapour mass in the atmosphere averaged over the grid columns. Also, we averaged all VSFs for Serano and Livorno. The evolution of the forecasted accumulated precipitation is shown in Figure 18a, while the evolution of the total hydrometeors and water vapour mass in the atmosphere is shown in Figure 18b.

Considering the Figures 18a and 18b it is apparent that flashes add less water vapour compared to radar reflectivity factor data assimilation and, of course, RADLI has the largest impact. In particular, the total water mass added to the background is 2.5%, 5.7% and 7.4% for LIGHT, RAD and RADLI, respectively. Importantly, the total water substance mass added by RADLI to the

background is less than the sum of the total water substance mass added by RAD and LIGHT. This happens because 3D-Var adds water to the background limiting the impact of nudging during the simulation. For example, in an already saturated atmosphere the nudging of Eqn. (2) doesn't have any impact.

Accumulated precipitation accounts for the largest part of the water vapour added to the simulation, similarly to Fierro et al. (2015). At the end the assimilation phase (6h), the evolution of the total water vapour and hydrometeors mass in the atmosphere converges towards the background as boundary conditions propagates into the domain.

*4.4 Sensitivity to nudging formulation*

As stated in Section 3.2, there are limitations when applying the nudging method of Fierro et al. (2012) to RAMS@ISAC. Also, the optimal setting of the coefficients of Eqn. (2) depends on the case study. For these reasons, it is interesting to evaluate the sensitivity of the results to changes in nudging formulation. For this purpose, we show the variability of ETS and POD scores to changes in the A and B coefficients of Eqn. (1). The scores are computed considering all the VSF for the two case studies for different configurations: A_76 has the coefficients A=0.76 and B=0.25; LIGHT has A=0.86 and B=0.15 (default setting), SAT has A=1.01 and B=0; RADLI has A=0.86 and B=0.15 (default setting); CTRL, and RAD are as defined in Table 3.

The scores are computed for the second RAMS@ISAC domains and are shown for the nearest neighbourhood. ETS score (Figure 19a) shows that all configurations assimilating either lightning or radar reflectivity factor alone or a combination of lightning and radar reflectivity factor improves the forecast for all thresholds. RADLI has the best ETS for rainfall intensity larger than 32 mm/3h in line with the results of the three VSF discussed above.

For rainfall lower than 32 mm/3h, the simulations assimilating lightning perform better, because they have less false alarms compared to those assimilating radar reflectivity factor (not shown). From the comparison of LIGHT and SAT with A_76, it is apparent that the latter has the worst score. The comparison between LIGHT and SAT shows mixed results: SAT performs better up to 38 mm/3h, while LIGHT is better for higher thresholds. This result is confirmed by POD, Figure 19b, which shows that SAT performs better up to 32 mm/3h, while LIGHT is better for higher thresholds. A visual inspection of the model output reveals that SAT can generate spurious convection in some areas while missing convection in other areas that are correctly forecast by LIGHT or even A_76, i.e. adding less water vapour to the model because of the different trajectories in the phase space followed by the model using different settings.

Lynn et al. (2015) implemented a method suggested by Fierro et al. (2012) to suppress spurious convection in WRF model. The method compares the lightning forecast during the assimilation period with lightning observations to filter out spurious convection. The application of the methodology on 10 July 2013 improved the forecast of the squall line from Texas to Iowa, which was the focus of the forecast on that day; however, the application of the method to 19 and 21 March 2012 over the US gave mixed results, improving the forecast in the first 6h and worsening it after 6h.

The implementation of this method in the RAMS@ISAC could be used to suppress spurious convection in simulations assimilating lightning, especially SAT.

It is finally noted that RAD and RADLI have high POD values for all thresholds, nevertheless their ETS is below that of LIGHT and SAT up to 32 mm/3 h (RADLI) and 42 mm/3h (RAD). This behaviour is caused by the larger number of false alarms given by assimilating radar reflectivity factor compared to those assimilating lightning. This result shows again that the RAD and RADLI configurations have a wet frequency bias. In particular, the frequency bias of RAD and RADLI configuration is about 3 between 20 and 40 mm/3h.

Table 3: Types of simulations performed.

[revised manuscript text omitted]

Pessi, A. T., and S. Businger, 2009: The impact of lightning data assimilation on a winter storm simulation over the North Pacific Ocean. Mon. Wea. Rev., 137, 3177–3195, doi:10.1175/ 2009MWR2765.1

Petracca, M., L. P. D'Adderio, F. Porcù, G. Vulpiani, S. Sebastianelli, and S. Puca, 2018: Validation of GPM Dual-Frequency Precipitation Radar (DPR) rainfall products over Italy. J. Hydrometeor., 19, 907–925, https://doi.org/10.1175/JHM-D-17-0144.1

Ricciardelli, E.; Di Paola, F.; Gentile, S.; Cersosimo, A.; Cimini, D.; Gallucci, D.; Geraldi, E.; Larosa, S.; Nilo, S.T.; Ripepi, E.; Romano, F.; Viggiano, M. Analysis of Livorno Heavy Rainfall Event: Examples of Satellite-Based Observation Techniques in Support of Numerical Weather Prediction. Remote Sens. 2018, 10, 1549.

Rohn, M., Kelly, G., Saunders, R. W.: Impact of a New Cloud Motion Wind Product from Meteosat on NWP Analyses and Forecasts, Monthly Weather Review, 129, 2392-2403, 2001.

Stensrud, D. J., and Fritsch, J. M.: Mesoscale convective systems in weakly forced large-scale environments. Part II: Generation of a mesoscale initial condition, Mon. Weather Rev., 122, 2068-2083, 1994.

Takahashi, T., 1978: Riming electrification as a charge generation mechanism in thunderstorms. J. Atmos. Sci., 35, 1536–1548, doi:https://doi.org/10.1175/1520 0469(1978)0352.0.CO;2.

Vulpiani, G., A. Rinollo, S. Puca, and M. Montopoli, 2014: A quality-based approach for radar rain field reconstruction and the H-SAF precipitation products validation. Proc. Eighth European Radar Conf., Garmish-Partenkirchen, Germany, ERAD, Abstract 220, 6 pp., http://www.pa.op.dlr.de/erad2014/programme/ ExtendedAbstracts/220_Vulpiani.pdf (last access January 2019).

Wiens, K. C., S. A. Rutledge, and S. A. Tessendorf, 2005: The 29 June 2000 supercell observed during STEPS. Part II: Lightning and charge structure. J. Atmos. Sci., 62, 4151–4177, doi:10.1175/JAS3615.1

---

## Author Comment (AC3) · 11 Feb 2019

Just to let you know that the discussion about lightning data assimilation has been updated. Please, refer to the reviewer#1 answer for this subject.

---

## Author Response (AR1)

**Answer to the reviewer 1 comment of NHESS-2018-319.**

We acknowledge the reviewer for the fast and complete review of the paper. In the following we show the actions taken to improve the paper quality. The Author Replies (AR) to the Reviewer Comments (RC) are in blu.

Before discussing in details the RCs, it's import to outline that the subject paper contained two typos. The first refers to the length scale of the background error matrix in the x and y directions that varies between 14 and 25 km and not , between 20 and 30 km, as erroneously reported in the manuscript. The second refers to the correct lightning number for each day are 82 331 for the 9 September, 291 164 for the 10 September (170 000 is written into the manuscript) and 105 467 for the 16 September (60 000 written in the manuscript). Despite the typos, the results shown in the paper were obtained using the correct number of flashes and the correct length scales in the background error matrix.

In the first submission, we stressed the improvement given by the data assimilation at the local scale on the precipitation VSF (Very Short term Forecast, 0-3h). To highlight this point, we showed several ways to improve the forecast by the assimilation of lightning, radar data or both, as it's evident for the Serano case study, for which the radar assimilation impacted the forecast of the first phase (03-06 UTC) whereas lightning impacted the second one (18-21 UTC).

Notwithstanding, given the comments of reviewers 1 and 2 and the results section (Section 4) underwent a substantial rewriting. In particular, in the revised version of the paper, we deleted Section 4.1.2 (second phase of the Serano case study) and Section 4.2.1 (first case of the Livorno case study). The results of Section 4.2.1 are now shortly commented in Section 5 (Discussion and conclusions) to highlight that there is space for improvement. Following the comments of the reviewer #2, the scores of the phases commented in the paper were summarized in three tables (Tables: 4-6) for specific thresholds (1, 6, 10, 20, 30, 40 mm/3h and, for Livorno, also 50 mm/3h). This limited the number of precipitation thresholds considered but increased the readability of the paper.

The space gained by deleting the aforementioned sub-sections was used to extend the discussion about the adopted assimilation methodologies. In particular, we extended the section "Lightning data assimilation" to include a discussion on the useful comments raised by the reviewer 1; we extended the section "Radar data assimilation" to show an example of 3D-Var assimilation of the reflectivity factor (this should also answer to few comments of the reviewer 1).

Finally, we added supplemental material to the paper discussing the following two points: a) the relative contribution to the total water mass given by lightning and radar reflectivity factor data assimilation (Section S1); b) the sensitivity of the precipitation VSF to the nudging formulation (Sections S2). In addition, the supplemental material provides different plots of Figures 15-17 (Section S3, as requested by reviewer 1) and the forward radar operator used in RAMS-3DVar (Section S4), as requested by the reviewer 3.
The important points considered in the supplemental material weren't included into the paper to avoid exceeding the length limit. However, the supplemental material is recalled in several parts of the paper to help the reader to consider it for reading.

**Reviewer's preamble**
Summary: The authors utilize a cloud-scale functional relationship between lightning and water vapor mass mixing ratio published in the literature and applied it to a homegrown 3DVAR

framework at the convection-allowing scale to evaluate the analysis and short term forecast of two
selected high impact weather events over Italy.
Recommendation: reject and, eventually, re-submit.
Main Comments:
While the manuscript could eventually offer some merit for this journal, I found the analysis
generally very rudimentary with the authors going at length in describing in excruciable level of
details individual figures/panels in a repetitive and redundant manner without distilling the content
into concise arguments/hypotheses. Given its repetitive nature, the entire results section could, in
fact, easily be condensed into a 2-3 pages. Most importantly, the manuscript (hereafter, m/s) lacks
rigor and rationales for the set ups and methods put forth for each, respective DA approaches.
Salient Major issues are itemized below.
**RC(1).** As far as the scientific content is concerned, the core ideas and notions of this lightning
data assimilation (LDA) method are conceptually similar to those from many existing studies,
which fundamentally aim at promoting convective development through the introduction of latent
heating within a prescribed neighborhood region/column centered at observed lightning locations.
Past works from Benjamin et al. (2004), Alexander et al. (1999), Chang et al. (2001), Papadopoulos
et al. (2005), Pessi and Businger (2009), have used empirical relationships between lightning-
rainfall rates-latent heating or lightning- reflectivity rates-latent heating [e.g., in the HRRR].
Following a similar idea, recent works such as Machand and Fuelberg (2014), Lynn et al. (2015),
Lynn (2017), Fierro et al. (2012; 2014, 2015), Wang et al. (2017, 2018) proposed LDA means that
essentially boost the local thermal buoyancy where lightning is observed. A very limited portion of
these techniques, however, offer alternative approaches to address spurious convection (i.e.,
removal) – which is a far more challenging problem to tackle. For completeness and given the
relatively limited advances in LDA relative to radar DA, the authors should do a better job in
discussing and including all the aforementioned references in their text. I was in fact astonished to
notice that the integrity of the Results section in section 4 is completely devoid of references to
previous works.
In particular, since they opted to borrow an LDA method from one of these investigators,
comparisons with their study should be performed more systematically throughout the m/s. For
instance, the works of Federico et al. 2017b is invoked when referring to multi-day forecast
statistics using the Fierro et al. method without mentioning that, such a study, was already
conducted by the same author over a larger domain and using nearly three times more forecast
days/cases (Fierro et al. 2015 study). Given this omission, their study (Federico et al. 2017b)
inadequately state that such multi-day statistics for this LDA have never been conducted. In a
similar manner, it is of relevance to underline whenever appropriate that, in this work: (i) radial
velocity is not included (specify why), (ii) only cloud-to-ground lightning data are considered and
(iii) spurious convection is not addressed. In the light of (i) and (ii), one on the recent studies they
cite (Fierro et al. 2016) not only assimilated level II radar data (radial velocity + reflectivity factor)
but used total lightning data. This needs to be clearly stated, for completeness (Cf comment 3 below
for rationales).
AR: In the revised version of the paper we extended the discussion of the LDA in the introduction,
in the data and method section and in the discussion of the results in order to include most of the
mentioned references.. The problem caused by the missed reference in Federico et al. (2017b) study
was corrected in the reviewed manuscript.
Regarding the specific comment (i): it's worth mentioning that we are working on the assimilation
of the radial velocity but the operator is not yet implemented in the 3D-Var. Besides, while the
reflectivity factor measured by the radar network is operationally available, the radial velocity is not operationally processed. Currently, it needs further research to manage some issues (complex orography, scan strategies optimized for rainfall estimation). For these reasons, we focused on the assimilation of the reflectivity factor. These motivations are discussed in the revised version of the paper in Section 3.3 by writing:

"Radial velocity is not assimilated within the RAMS@ISAC model because it is not operationally processed , the scan strategy being optimized for QPE purposes. Furthermore, the implementation of a radial velocity data assimilation scheme  is under development in RAMS-3DVAR and it is not currently available for testing. For these reasons, we didn't consider the assimilation of this parameter. "

Regarding point (ii) in the paper we wrote that total lightning is assimilated, not only CG. For the events analyzed in the paper the fraction of IC strokes to the total number of strokes detected by LINET is about 30% (22% on 09 September, 30% on 10 September and 35% on 16 November). There are cases when the IC strokes recorded by LINET are more than 50% of the total number of stokes over Italy. In general, the Section on LDA has been extended to consider this point and others; (iii) The spurious convection is not considered by the LDA but it is considered in the assimilation of radar reflectivity factor. We specified better this point in the revised version of the paper, but the comment is already present in the first submitted version.

**RC (2).** In term of DA methodology, I found one major drawback, which is never discussed, nor evaluated. Given that both the LDA and their "RAD" experiment make adjustments to the relative humidity (RH) field, it is expected that both techniques will overlap in their adjustments over all the (many) grid points characterized by observed lightning flash rates exceeding zero. This is because changing RH is equivalent to adjusting Qv as RH ~ Qv/Qv_saturation. A more self-consistent DA approach would adjust the pseudo- observations for the Qv or RH field in a manner that eliminates any possibility of overlap during the minimization. Toward that end, the authors should include soundings and/or horizontal cross sections of RH/Qv that shows, quantitively, how the RH field is adjusted by each respective DA approach (radar vs lightning).

Second, given that lightning is a cloud-scale observation, I cannot find any justifications for not conducting the 3DVAR analysis on the innermost, higher resolution domain. Instead, the method minimizes the cost function on the intermediate domain and, later, projects the innovations on the coarser-scale domain. This needs to be addressed.

AR: First: we added a complete new section (Section S1 of the supplemental material) to address this point. In this section, we show the evolution of the accumulated precipitation and total water mass in the atmosphere (i.e. water vapour mass+mass of hydrometeors) as a function of time (including the spin-up period).

Second:

Data assimilation is not performed on domain D3 (R1) because the use of this domain is exceptional and we don't have background error statistics for this grid.

The background error statistics for the domain D2 is computed by the NMC method, which, for this paper, is based on HyMeX-SOP1 simulations. The Appendix A and B of Federico (2013) show the details of the application of the method, which requires a large number of simulations (see Barker et al., 2004 for the general discussion). These simulations are not available for the innermost grid of
the Livorno case, which was introduced to better resolve the precipitation at the local scale and to
show how precise can be the impact of lightning and radar data assimilation on the VSF. These
motivations have been clarified in the revised version of the paper.

It is worth specifying that this limitation refers only to the assimilation of the radar reflectivity
factor because flashes are assimilated by nudging. Nevertheless, we could not compare simulations
with or without data assimilation for a specific domain assimilating lightning in the innermost
domain and for this reason we assimilated flashes over the D2 only.

In the revised version of the paper, we specified better the role of the domain D3 and the reason for
not assimilating lightning and radar reflectivity factor over the domain D3.

It is specified in section 3.1 (RAMS@ISAC and simulations set-up) as follows:

"The third domain covers the Tuscany Region, has 4/3 km horizontal resolution (R1), and it is used
for Livorno to represent with higher spatial detail the precipitation field over Tuscany. The fine
structures of the precipitation field are smeared out over Tuscany using only domains D1 and D2.
The operational implementation of the RAMS@ISAC model uses the domains D1 and D2 and no
refinements for specific areas of Italy are used because Italy is a complex orography country and
grid refinements for a specific event can be done only a-posteriori, i.e. after the occurrence of the
event."

And few lines below:

"It is noted that data assimilation is performed over the domain D2 (R4) only, and the innovations
are transferred to domain D3 (R1), for the Livorno case, by the two way-nesting. The domain D3 is
used for the Livorno case to refine the resolution of the precipitation field over Tuscany and to
show the spatial and temporal precision of the precipitation forecast over Tuscany using data
assimilation. However, its usage is exceptional because, as stated above, Italy is a complex
orography country and grid refinements over specific areas are used only after the occurrence of an
event. For these reasons the domain D3 is usually not used in RAMS@ISAC and statistics about the
background error aren't available for this grid. The background error in RAMS-3DVar is computed
by the NMC method (Parrish and Derber, 1992), which requires a number of simulations (at least
two-weeks) verifying at the same time but starting with a lag of 12 h. These simulations are not
performed in this paper and background error statistics for the domain D3 are not available.

Being lightning assimilated by nudging, they could be assimilated over the domain D3.
Nevertheless, to preserve the rationale of the paper, i.e. comparing simulations with or without data
assimilation for specific domains, we didn't assimilate lightning over the domain D3.

Because lightning and radar cloud scale observations, their assimilation at higher horizontal
resolution is foreseeable in future works."

RC Third, the radius of influence/decorrelation length scale chosen for radar reflectivity factor (50
km) is far too large for convective scale applications and would incur unrealistically large amount of Qv mass added into the domain – which will undoubtedly yield to spin-up issues and the
generation of convective-scale gravity waves that will degrade longer term (>= 3h) solutions
(please provide plot of perturbation pressure in your response). In that regard, the authors should
indicate and contrast the total amount of Qv mass added by RAD and LIGHT.

AR: The 50 km length is not a distance to spread the innovation introduced by the radar reflectivity
factor data assimilation. It represents a search radius to compute the pseudo-profile of relative
humidity used in 3D-Var. A discussion about this point was introduced in the new section on radar
reflectivity factor data assimilation (Section 3.3).
In particular, we wrote:

"It is important to point out that the 50 km length-scale of the above step doesn't
represent the horizontal correlation length-scale of the background error, which
determines the horizontal spread of the innovations in the 3D-Var data assimilation (the
latter length-scale is between 14 and 25 km depending on the level). The 50 km length-
scale is used to set a square for computing the pseudo-profile of relative humidity (Eqn.
(2)). This profile is given by a weighted average whose weights are determined by the
agreement between the simulated and observed reflectivity factor. The larger the
agreement the larger the weight. This distance seems appropriate because the spatial
error of meteorological models in simulating meteorological features, for example fronts,
can be of this order. The control simulation for the two events considered in this paper
confirms this choice."

**RC (3).** In the context of forecast improvements, the Qv-based method they borrowed/adapted was
scaled for total lightning data (> 50% detection efficiency of intra-cloud [IC] flashes). I was
surprised to find that absolutely no information on the detection efficiency and geolocation
accuracy of the lightning network used (LINET) is provided in the text [no figures either]. Given
the large area covered by this study, it is thus very likely that the geolocation accuracy of this
network remains very poor for low amplitude flashes and for all flashes over oceanic regions. Given
the low sferics amplitudes of IC flashes, the VLF portion of the sensor will miss nearly all these
flashes, while the VHF portion only is able to detect some of the IC flashes within a few tens of
kilometers away from the station [e.g., Rison, MacGorman works]. Thus, it is relevant to state and
underscore that LINET only detects a very small portions of the total IC flashes in the study domain
(likely < 5%). Motivation for scaling the F12 method for IC flashes (in lieu of cloud-to-ground
[CG] flashes), lies in the well-documented finding that, in contrast to CGs, ICs are well correlated
with thunderstorm kinematic and microphysical evolution (updraft strength, updraft volume,
graupel mass etc, see Wiens et al. 2005, Schultz et al. 2011 among many others). CGs, on the other
hand, were found to be correlated with the descent of reflectivity cores and the onset of the demise
of the storm's updraft core [MacGorman and Nielsen 1991, MacGorman et al. 1989, Rutledge and
Lang's seminal works etc]. Not surprisingly, ICs were found to lag CG by an average of 15 min
[see one of the recent MacGorman study]. Moreover, Boccippio et al. 2001 and Medici et al. 2017
found that in deep continental convection, IC flashes always outnumber CGs by a ratio sometime
exceeding 10:1. Based on these facts, it becomes clear why the Fierro method emphasized the use
of IC flashes [or total lightning] for their application. Further motivation arises from the recent
successful launches of the GLM instrument aboard GEOS-16/17, which will provide continuous
day/night coverage of total lightning at ~90% detection efficiency (DE) over a large domain
covering the Americas (Gurka et al. 2006; Goodman et al. 2012, 2013, Rudlosky et al. 2018). Note
that GLM will provide flash extent information of lightning, while the metric derived from the
(limited) point flash data in this study can only provide a very rough surrogate for CG flash location
density at best. Similar space-borne technology to detect lightning have been developed by China
(Feng-Yun-4, yang et al. 2016) with these data being assimilated in recent works by Wang et al.

(2017, 2018) – which were never referenced either. Apart from their propensity to detect total
lightning at a high DE, the chief advantage of this technology lies in its ability to retrieve lightning
over remote oceanic regions.
AR. LINET has been started and used operationally since 2004. Since then, more than 100
publications provided evidence about both DE and location accuracy (LA). In particular, since the
beginning in 2004 LINET exhibited a statistical average location accuracy of some 100 m. Because a
minimum of 5 sensor reports are exploited for each stroke solution, the LA does not deteriorate
within several 100 km from a sensor. Thus, the LA is excellent all over the present study region.

LINET Europe comprises more than 200 sensors and provides more extensive stroke data than any
other VLF/LF system in the region.

LINET detects and records stroke signals down to currents of a few kA (CG normalization). This is
the reason why LINET ranges are large enough to exploit >=5 sensors for geolocation without
reducing the typical baselines of 250-300 km. The resulting DE is good enough to detect any CG.
Over the Mediterranean the stroke DE diminishes due to larger baselines. However, the flash
detection is less sensitive because of the stronger strokes that characterize a flash.

Like any other VLF/LF system signals are recorded whether CG or IC. Thus, the detected IC portion
is certainly not lower than in any other VLF/LF system. As a consequence, total lightning is
reported at least as efficient than in any other VLF/LF system, and will be beneficial for the
purposes in the present paper.

IC discrimination of LINET is based on TOA analysis. The advantage is a unique discrimination when
the detection geometry is within certain ranges; the disadvantage is decreasing discrimination
power when the distance to the closest sensor become too large, because of too small TOA
differences between CG and IC at the same 2D location. Thus, over water far from land the
identified IC fraction decreases, though total lightning counts remain relevant.

It is true that leader steps signify discharge processes (see, e.g., well-known LMA results).
However, it is well-proven that VLF/LF detects pulses from IC activity that are very similar to CG
strokes; this is why CG-IC discrimination is very challenging for VLF/LF systems. We think, though,
that any VHF issue is not relevant here, because there is no large-scale VHF system that covers
Italy and the surrounding sea with baselines of a few 10 km.

Observations from global networks or satellites may be a point of future concern, but do not
represent any focus in the present paper; also, IC discrimination is either not yet possible or poor.
It may be mentioned that GLM lightning data is not yet an issue; interestingly, Eumetsat/NASA on
behalf of NOAA have selected LINET to carry out the first evaluation of the new lightning data
source. This has been communicated in Science Team Meetings and conferences (see GLM Cal/Val
2017 Ground Validation Field Campaign 2017).

The discussion about the LINET network has been extended in the revised version of the paper.
RC (4) The following key information pertaining to the respective DA methods are missing/never
discussed:
(a) What are the background/observation errors for reflectivity/lightning? (b) What statistics are
used for model error?

AR. Lightnings are assimilated by nudging and no error is associated with them. The error matrix
for model error has been clarified in the section on the radar reflectivity factor data assimilation
(Section 3.3).

RC (c) How is the adjoint for the lightning data assimilation operator derived?
AR. The derivation of the adjoint of lightning data assimilation was performed using two case
studies of the HyMeX-SOP1 (unpublished work) as commented in Federico et al., (2017a). We
commented about this point in the new section on lightning data assimilation. Also, the
supplemental material (Section S2) shows the sensitivity of the rainfall VSF scores (POD and ETS)
to the nudging formulation.

RC (d) What assumptions are made for grid points with zero lightning or zero reflectivity
observations ? Does the radar DA or LDA treat those as missing observations or equate those to the
background values to reduce spread ?
AR. Lighting are assimilated by nudging and this comment doesn't apply. In the case of radar, grid
points with zero reflectivity factor and zero simulated reflectivity factor are assumed missing
observation, and the innovations can spread according to the background error matrix. This has
been clarified in the revised version of the paper (Section 3.3).

RC (e) What Gaussian decorrelation length scales are assumed for each observation? Please
specify/justify/explain. How would the selection of a given length scale value, influence the results
?
AR. The observation error matrix for radar reflectivity is diagonal (this was already stated in the
first submission of the paper). We acknowledge that the sensitivity tests proposed by the reviewer
are interesting, nevertheless they are left for future studies. The importance of this point is discussed
shortly in the revised version of the paper. We wrote:

"The observation error matrix R in Eqn. (4) is diagonal and observations' errors are uncorrelated.
This choice is partially justified by under sampling the radar reflectivity factor observation by
choosing one point every five grid points in both horizontal directions of the radar observations
Cartesian grid (Rohn et al., 2001). However, correlation observations errors have significant impact
on the final analysis, as shown for example in Stewart et al. (2013), and different choices of the
matrix R will be considered in future studies.
The value of the elements on the diagonal of R depends on the vertical level and are 1/4 of the
diagonal element of the $B_z$ matrix at the corresponding height. By this choice, we give more credit
to the observations than to the background and analyses strongly adjust the background towards
observations."

RC (f) Is spurious convection addresses by either DA method ? Please elaborate.
AR. Yes, in the radar reflectivity factor data assimilation, but not in lightning data assimilation. The
point is already present in the discussion paper, but it has been clarified in the revised version of the
paper in the section dedicated to the radar data assimilation (Section 3.3).

RC (g) Does the variational minimization set use a multi-scale approach ? If yes, what influence
radii are chosen and how many cycles ?
AR. We don't use the multi-scale approach. This has been clarified in the paper in the section
dedicated to the radar data assimilation (Section 3.3).

RC (5) Why did the authors not include the fractions skill score FSS as the main evaluation metric
for their forecast? Several works have posited that, in contrast to ETS, FSS does not penalize displacement errors as much and, arguably, FSS offers a more accurate measure of skill on convection-allowing grids (Mittermaier et al. 2013).

Additionally, more recent studies evaluating forecast performance have been making usage of the so-called performance diagrams, which conveniently merge several key contingency table elements into one single diagram (Roebber 2009). The authors should show such diagrams to provide a more complete and succinct view of the overall forecast performance of the case they selected.

AR. Considering POD and ETS gives the possibility to show the many facets of a forecast, and this is important. These scores are also widely used in the bibliography and this make the results of this paper comparable with other papers. We, of course, acknowledge that there are other interesting measurements of the model performance, as FSS, that could be considered. Taking into consideration this comment and the comment of the reviewer 2 about the score we propose the following solution: we consider three neighborhood radii to take into account for displacement errors; nearest neighborhood (as in the first submission), 25 km and 50 km. We put the scores in three tables (Tables 4-6 of the revised paper) following a remark of the second reviewer.

RC (6) The case studies selected are cherry-picked given the confession that CTRL generally failed to provide reasonable forecast estimates of precipitation for both cases herein. For good measure, fairness and to better underscore the performance of the DA method, the authors should show the results for one case in which CTRL did not perform well and contrast it to one case where CTRL did preform reasonably well.

AR. The events were selected because they were missed by several forecasts and, for this reason, they are challenging. Moreover, they had important consequences because nine people died and damage to properties was extensive. We stressed this point in the introduction by writing:

"The forecast of severe events at the local scale still remains a challenge because of the multitude of physical processes involved over a wide range of scales (Stensrud et al., 2009). The Serano case, being localized in space, poses challenges in forecasting the exact position and timing of convection initiation; the Livorno event involves the interaction between a high impact storm with the complex orography of Italy, which is difficult to simulate at the local scale. For the above reasons the forecast of both events was challenging, as confirmed by the poor forecast of RAMS@ISAC. The difficulty to forecast timely and accurately the precipitation fields of the two events is the reason for choosing them as test cases."

RC (7) The authors omit to mention that the degradation of the forecast at >= 3h is mainly due to saturation of the model solution by errors and biases within the initial / boundary conditions derived from large scale models or re-analysis datasets. This needs to be shown for both cases, especially given the unrealistically large (50 km) decorrelation length scale used for radar reflectivity factor.

AR. Agreed, we considered this point in the revised version of the paper. However, the model errors play also a role in the degradation of the forecast in addition to IC/BC. Again 50 km is not the decorrelation length scale for radar reflectivity factor.

Section 5 (Discussion and Conclusions) was updated by adding the following statements:

"Another important point to study is how long the innovations introduced by data assimilation lasts in the model forecast. While in this study we consider the VSF at 3h, future studies must explore longer time ranges. This kind of study was performed for lightning data assimilation (Fierro et al., 2015; Federico et al., 2017b; Lynn et al. 2015 among others) and for radar data assimilation (Hu et al. 2006; Jones et al. 2014, among others), using a rationale similar to that used in this paper.

In general, the performance of the forecast and the impact of lightning and radar data assimilation
decrease with forecasting time because of the propagation of boundary conditions inside the
domain and because of model errors. Improving the data assimilation system also contributes to a
longer resilience of model performance. The studies cited above showed that lightning and radar
data assimilation can have an impact up to 24h depending on several factors (model, data
assimilation, quality of the data, meteorological conditions, initial and boundary conditions).
A study considering both radar reflectivity factor and lightning should be performed to understand
the resilience of the innovations introduced by data assimilation. "

RC (8) Title: Revise to include that: (i) primarily CG flashes are assimilated and (2) the model
vehicle is RAMS.

AR. We included in the title that RAMS@ISAC is the model vehicle. As stated above, the IC
flashes for the case studies considered in this paper is about 30%, which is not a small fraction of
the total lightning. The discussion on the method assimilating lightning has been widened to
consider this and other points. Considering the comments of the reviewer 3 we decided to include
the word "preliminary" to the paper title. The new title is:

**"Preliminary results of the impact of lightning and radar data assimilation on the very short
term rainfall forecasts of RAMS@ISAC: application to two case studies in Italy"**

RC Because these issues are collectively substantial and would require thorough rewriting of the
manuscript in many places, I opted not to dwell on editorial comments for the time being.
Additionally, the level of English remains, in my view, unacceptable for publication.

AR. We revised the English of the paper, also according to the comments of the reviewer 2 in the
PDF file. The copy-editing service of the journal will also improve the quality of the English.

RC Figures:
Figures 5, 6, 8, 12a, 13a, 14a, 15a, 16a, 17a, 18a: The use of colored dots makes it very difficult to
effectively compare the observations with those of the simulations: For consistency, either both sets
of plots should show colored dots or shaded contours. For lightning, the authors should effectively
show the gridded lightning data that were used to create the Qv or RH pseudo-observations.

AR. Agreed. It is always difficult to choose the right representation of the precipitation field when
comparing model output with raingauges. We acknowledge that the solution suggested by the
reviewer is a good one, however we also like our representation because: a) rainfall at the
raingauges is not interpolated, avoiding errors introduced by the interpolation process; b) the
rainfall predicted by the model shown as a field gives the possibility to see the behavior of the
model also in parts uncovered by raingauges. We propose the following solution: a) redrawing the
RAMS@ISAC rainfall field changing the colorbar to match exactly the raingauges colorbar; b)
adding the representation suggested by the reviewer as supplemental material to the paper
(Figures S3-S5 of the supplemental material); c) recalling the supplemental material when
discussing the second VSF of Livorno to highlight the wet frequency bias when assimilating radar
reflectivity factor (see Figure S5 of the supplemental material).
Ok for the Figures about lightning. They have been redrawn according to the reviewer remark.

RC Additional comments:

General comment: What is the main rationale for using a model that is marginally known by the
community (RAMS) versus a more commonly used, battle tested, publicly available model such as
WRF-ARW ? The authors not only seem to re-invent the wheel here but render any potential future
work dedicated to reproducibility of the results - to the least - very challenging.
AR. RAMS@ISAC is used/maintained/developed at ISAC-CNR since several years (and it is also
operational over Italy since 2000, in different versions/adaptations etc), and it is important for us to
test our tool for challenging cases, as those considered in this work.
Also, we are WRF users too (see, for example, Avolio and Federico, 2018) and for the cases of this
paper no substantial differences were found for the performance of WRF and RAMS@ISAC
models (using the same initial and dynamic boundary conditions). The performance of the WRF
model for the Livorno case is shown, for example, in Ricciardelli et al. (2018). The reviewer can
verify that the comments given in this paper about the performance of RAMS@ISAC for the most
intense phase of the Livorno case can also be applied to the WRF model (see specifically their
Figures 11 and 12 for the most intense phases in Livorno). It has to be noted  that Ricciardelli et al.
(2018) used ECMWF IC/BC, which are different from that used in this paper. So, the results of this
paper could be even more valuable because they are "more general" and not linked to the specific
modelling tool.
To clarify this point we added the following sentence in Section 4.2.3:
"Considering the evolution of CTRL rainfall forecast for the two VSF of Livorno, we concluded that
CTRL was able to predict abundant rain over Livorno, but the rainfall forecast was delayed
compared to the real occurrence. A similar behaviour was found in Ricciardelli et al. (2018) using
the WRF model, showing that the results of this paper for Livorno are likely not tied to the specific
model used."
RC (1) Bottom, page 2: what are "conventional data" ? Why are radial velocity data not used ? Line
70: the main advantage of using 3DVAR vs 4DVAR, EnKF or hybrid methods lies in their already
low computational burden. Thus, I do not agree with this justification. Also, variables are not
"perturbed"; but adjusted by VAR methods.
AR. See reply to the RC on the same topic, i.e., assimilation of Doppler velocity. We changed the
paper according to the reviewer suggestion for the other parts of the comment.
RC (2) Pages 3 and 4: Please refer to Major Comments 1 and 3. Lines 105: Given that "Federico et
al. (2017a) implemented the methodology of Fierro et al. (2012) ...", how come on line 112 "We
use the method of Federico et al. (2017a) to assimilate lightning..." ? Please revise accordingly.
AR. Ok for this comment. We added the reference to Fierro et al. (2012). The comment of line 112
come from the fact that we intended to cite the adaptation of the methodology, that is discussed in
Federico et al. (2017a). We clarified that conventional data are SYNOP and RAOB.
RC (3) Line 124: c.f. end of Major Comment 1.
AR. We wrote "total lightning".
RC (4) Line 240: RAMS used diagnostic relationships (vs explicit) to forecast lightning as it does
not explicitly solves for the 3D electric field. Line 243: "Fourth"
AR. For the first comment we wrote: "Second, it predicts the occurrence of lightning following the
diagnostic methodology of Dahl et al. (2012),…."
RC (5) Line 290: Delete equation set as these are considered basic/common knowledge.

AR. In some papers, where we omitted the equations, we had the opposite comment. However, for
this paper, to reduce length and to give more space to the important points raised by the reviewers
the equations were deleted.

RC (6) Section 3.2, lines 300-312: Explicitly state and indicate that equation (2) is from Fierro et al.
(2012, 2015) and not from Federico et al. Line 305: Please explain the rationales behind the choices
of these constants: In particular, how are the forecast metrics affected for a 20% change in A, which
has been shown to exhibit the most notable influence on the forced convection?
AR. Ok for the reference. The functional form is of the same as in Fierro et al. (2012, 2015), but the
coefficients were adapted to RAMS@ISAC as shown in Federico et al (2017a). In Federico et al.
(2017a) it is clearly stated that the method is that of Fierro et al. (2012), the only difference being
the adaptation to RAMS@ISAC model. A sensitivity test to the nudging formulation is shown in
the supplemental material (Section S2).

RC (7) Line 316-317: c.f. Major Comment 2.
AR. We clarified better the ability of LINET to detect both IC an CG.

RC (8) End of page 11: c.f. Major Comment 2
AR. Not applicable to the revised version.

RC (9) Line 356: do the authors refer to the LFC or the LCL, (which may I add is an idea borrowed
from Marchand and Fuelberg 2014 and Fierro et al. 2016). What is the top of the adjustment layer
for lightning ? Please elaborate.
AR. It is the LCL. The idea is of Caumont et al. (2010), we didn't add the reference to this point of
the paper because the whole methodology is taken from Caumont et al. (2010), already cited
several times. The top adjustment for lightning is -25°C. However, this is already stated in the
paper (Lines 314-315 "The check and eventual substitution of the water vapor is performed every
five minutes and it is made only in the charging zone (0 °C, -25°C).").

RC (10) Line 410 and elsewhere. This is similar to the results of Fierro et al. 2016. C.f. Major
Comment 1. Please establish comparisons with previous works throughout the manuscript.
AR. We integrated better the results of this paper with those of previous works.

RC (11) Line 669: This statement is incorrect. The DE of ground based sensors levels off very
rapidly with distance from land. This is where space-borne lightning detection systems such as the
GLM or Feng Yun-4 can fill the gap.
AR. Ok, however the good coverage of the LINET network for some important areas, as between
Corsica and Italian mainland (both Liguria and Tuscany) makes this point "less problematic" for the
Livorno case, while the Serano event occurred over the Italian mainland.

RC (12) Lines 716-725: c.f. Major Comment 1.
AR We added the references. (Fierro et al., 2015; Federico et al., 2017b; Lynn et al. 2015, Hu et al.,
2006; Jones et al., 2014).

REFERENCES (it misses some references given by the reviewer)

Alexander, G. D., Weinman, J. A., Karyampoudi, V. M., Olson, W. S., and Lee, A. C. L.: The effect of
assimilating rain rates derived from satellites and lightning on forecasts of the 1993 superstorm,
Mon. Weather Rev., 127, 1433–1457, 1999.

Avolio,E., Federico,S. : WRF simulations for a heavy rainfall event in southern Italy: Verification and
sensitivity tests, Atmospheric Research, Volume 209, 2018, Pages 14-35, ISSN 0169-8095,
https://doi.org/10.1016/j.atmosres.2018.03.009.

Barker, D.M., Huang, W., Guo, Y.-R., and Xiao, Q.N.: A Three-Dimensional Variational Data
Assimilation System For MM5: Implementation And Initial Results, Monthly Weather Review, 132,
897-914, 2004.

Benjamin, S. G., and Coauthors, 2004: An hourly assimilation– forecast cycle: The RUC. Mon. Wea.
Rev., 132, 495–518, doi:10.1175/1520-0493(2004)132,0495:AHACTR.2.0.CO;2.

Boccippio, D. J., K. L. Cummins, H. J. Christian, and S. J. Goodman, 2001: Combined satellite- and
surface-based estimation of the intracloud–cloud-to-ground lightning ratio over the continental
United    States.    Mon.    Wea.    Rev.,    129,    108–122,    doi:10.1175/1520-
0493(2001)129,0108:CSASBE.2.0.CO;2

[revised manuscript text omitted]

Pessi, A. T., and S. Businger, 2009: The impact of lightning data assimilation on a winter storm simulation over the North Pacific Ocean. Mon. Wea. Rev., 137, 3177–3195, doi:10.1175/ 2009MWR2765.1

Ricciardelli, E.; Di Paola, F.; Gentile, S.; Cersosimo, A.; Cimini, D.; Gallucci, D.; Geraldi, E.; Larosa, S.; Nilo, S.T.; Ripepi, E.; Romano, F.; Viggiano, M. Analysis of Livorno Heavy Rainfall Event: Examples of Satellite-Based Observation Techniques in Support of Numerical Weather Prediction. Remote Sens. 2018, 10, 1549.

Rohn, M., Kelly, G., Saunders, R. W.: Impact of a New Cloud Motion Wind Product from Meteosat on NWP Analyses and Forecasts, Monthly Weather Review, 129, 2392-2403, 2001.

Stensrud, D.J., M. Xue, L.J. Wicker, K.E. Kelleher, M.P. Foster, J.T. Schaefer, R.S. Schneider, S.G. Benjamin, S.S. Weygandt, J.T. Ferree, and J.P. Tuell, 2009: Convective-Scale Warn-on-Forecast System. Bull. Amer. Meteor. Soc., 90, 1487–1500, https://doi.org/10.1175/2009BAMS2795.1

Stewart, L. M., Dance, S. L., Nichols, N. K.: Data assimilation with correlated observation errors: experiments with a 1-D shallow water model, Tellus A: Dynamic Meteorology and Oceanography, 65:1, 2013, DOI: 10.3402/tellusa.v65i0.19546

Wiens, K. C., S. A. Rutledge, and S. A. Tessendorf, 2005: The 29 June 2000 supercell observed during STEPS. Part II: Lightning and charge structure. J. Atmos. Sci., 62, 4151–4177, doi:10.1175/JAS3615.1

**Answer to the reviewer #2 comments on NHESS-2018-319.**

We acknowledge the reviewer for the useful comments on the paper, both in the general comments section and in the pdf file. Our answers are in red. The Author Replies (AR) to the Reviewer Comments (RC) are in blue.

AR: Before discussing in details the RCs, it's import to outline that the subject paper contained two typos. The first refers to the length scale of the background error matrix in the x and y directions that varies between 14 and 25 km and not , between 20 and 30 km, as erroneously reported in the manuscript. The second refers to the correct lightning number for each day are 82 331 for the 9 September, 291 164 for the 10 September (170 000 is written into the manuscript) and 105 467 for the 16 September (60 000 written in the manuscript). Despite the typos, the results shown in the paper were obtained using the correct number of flashes and the correct length scales in the background error matrix.

RC: Extracted from the general comments:

This manuscript addresses an interesting and challenging topic, moreover represents a substantial contribution to the understanding of natural hazards and their consequences matching the scope of NHESS. However, the scientific and presentation quality are poor, above all because the results are presented in a "repetitive and heavy" manner and the English language needs a deep revision.

Specific comments: please see the notes on the pdf for each section and also for each figure's caption, moreover please deeply motivate the reason why: a) radial velocity is not assimilated: the operator is not implemented or the data are not available?; b) data assimilation is performed in the domain D02 only; c) you used a background error matrix of fall 2012 for case studies of late summer 2017. Technical corrections: please see the notes on the pdf for each section and also for each figure's caption.

AR: Comment "...in a repetitive and heavy manner". In the first submission of the paper we stressed the important improvement given by the data assimilation at the local scale on the precipitation VSF (Very Short term Forecast, 0-3h). To highlight this point, we showed the many ways in which the forecast is improved by the assimilation of lightning, radar or both data. For example, the two stages of the Serano case show that the radar (first phase 03-06 UTC on 16 September) or lightning (second phase of the event, 18-21 UTC on 16 September) were the key observation to assimilate in order to improve the precipitation VSF. Also, other stages had specific aspects that we discussed.

Our attempt, however, was not successful, considering the comments of reviewer #1 and reviewer #2 and the results section (Section 4) underwent a substantial rewriting. In particular, in the revised version of the paper, we deleted Section 4.1.2 (second phase of the Serano case) and Section 4.2.1 (first case of the Livorno case). The results of Section 4.2.1 are shortly commented in Section 5 (Discussion and conclusions) to highlight that there is space for improvement. Following the comments of the reviewer #2, the scores of the phases discussed into the paper have been put in three tables (Tables: 4-6) for specific thresholds (1, 6, 10, 20, 30, 40 mm/3h and, for Livorno, also 50 mm/3h) and, following the remarks of the reviewer #1, different neighborhood radii have been used to compute ETS and POD scores.

The space gained by deleting the two sub-sections stated above and by other corrections was
used to extend the discussion about the methods of assimilating lightning and radar in the 3D-
Var model. These sections should be clear in the revised version of the paper.

Finally, to avoid the repetitive discussion, we added supplemental material to this paper to
show how lightning and radar data assimilation works together, presenting the evolution of
the total water averaged for all VFS of the two cases and including in this discussion the
assimilation stage (Section S1).

The supplemental material shows also sensitivity tests of the precipitation scores (POD and
ETS) to the nudging formulation (Section S2). Section S2 requested new simulations with
different model settings (see Table S1 in the supplemental material).

Also, the supplemental material contains some plots requested by the reviewer #1 and the
formula used to compute the reflectivity factor (dBz) of RAMS@ISAC requested by reviewer
#3.

We didn't include the supplemental material in the paper, as stated in the discussion phase of
this paper, to avoid excessive length. However, the paper has few references to the
supplemental material to help readers to decide if they are interested to this material.

"…the English language needs a deep revision". We revised the English of the paper, also
according to the remarks of the reviewer #2 in the PDF file. Also, the copy-editing service of
the journal will improve the quality of the English before the eventual publication of the
paper.

AR: Specific question a) We are working on the assimilation of the radial velocity but the
operator is not yet implemented in the 3D-Var. Also, while the reflectivity factor measured by
the radar network is operationally available, the product of radial velocity is under
development. At the moment, it needs further research to solve some issues (complex
orography, operations of the radars not optimal for the Doppler retrieval and others). For
these reasons, the attention was on the assimilation of radar reflectivity factor. These
motivations are discussed in the revised version of the paper in Section 3.3 by writing:

"Radial velocity is not assimilated within the RAMS@ISAC model because it is not operationally
processed, the scan strategy being optimized for QPE purposes. Furthermore, the implementation of
a radial velocity data assimilation scheme is under development in RAMS-3DVAR and it is not
currently available for testing. For these reasons, we didn't consider the assimilation of this
parameter. "

Specific question b) Data assimilation is not performed on domain D3 (R1) because the use of
this domain is exceptional and we don't have background error statistics for this grid.
Background error statistics for the domain D2 are computed by the NMC method, which, for
this paper, is based on HyMeX-SOP1 simulations. The Appendix A and B of Federico (2013)

shows the details of the application of the method, which requires a number of simulations
(see Barker et al., 2004 for the general discussion). These simulations are not available for the
innermost grid of the Livorno case, which was introduced to better resolve the precipitation
at the local scale and to show how precise can be the impact of lightning and radar data
assimilation on the VSF. These motivations are clarified in the revised version of the paper.

Of course, this limitation is only for radar reflectivity factor because flashes are assimilated by
nudging. Nevertheless, we couldn't compare simulations with or without data assimilation for
a specific domain assimilating lightning in the innermost domain and, for this reason, we
assimilated flashes over the domain D2 only.

In the revised version of the paper we specified better the role of the domain D3 and the
reason for not assimilating lightning and radar reflectivity factor over the domain D3.

We wrote in section 3.1 (RAMS@ISAC and simulations set-up)

"The third domain covers the Tuscany Region, has 4/3 km horizontal resolution (R1), and it is
used for Livorno to represent with higher spatial detail the precipitation field over Tuscany
and to show better the precision of the rainfall VSF using data assimilation at the local scale.
The fine structures of the precipitation field are smeared out over Tuscany using only
domains D1 and D2. The operational implementation of the RAMS@ISAC model uses the
domains D1 and D2 and no refinement for specific areas of Italy are used because Italy is a
complex orography country and grid refinements for a specific event can be done only a-
posteriori, i.e. after the occurrence of the event."

And few lines below:

"It is noted that data assimilation is performed over the domain D2 (R4) only, and the
innovations are transferred to domain D3 (R1), for the Livorno case, by the two way-nesting.
The domain D3 is used for the Livorno case to refine the resolution of the precipitation field
over Tuscany and to show the spatial and temporal precision of the precipitation forecast
over Tuscany using data assimilation. However, its usage is exceptional because, as stated
above, Italy is a complex orography country and grid refinements for specific areas are used
only after the occurrence of an event. For these reasons the domain D3 is usually not used in
RAMS@ISAC and statistics about the background error aren't available for this grid. The
background error in RAMS-3DVar is computed by the NMC method (Parrish and Derber,
1992), which requires a number of simulations (at least two-weeks) verifying at the same
time but starting with a lag of 12 h. These simulations are not performed in this paper and
background error statistics for the domain D3 are not available.

Being lightning assimilated by nudging, they could be assimilated over the domain D3.
Nevertheless, to preserve the rationale of the paper, i.e. comparing simulation with or without
data assimilation for specific domains, we didn't assimilate lightning over the domain D3.

Being lightning and radar cloud scale observations, their assimilation at higher horizontal
resolution is foreseeable in future works. "

Specific questions c) We chose the background error matrix computed for HyMeX-SOP1
because the period was characterized by several convective events over Italy, as documented
in Ferretti et al., (2014), while the period preceding the convective events of this paper was
characterized by fair weather, typical of the summer Mediterranean season. For this reason,
we believe that the matrix for the HyMeX-SOP1 is more representative of convective events
compared to the matrix computed for the period of the storms occurrence. We wrote to
comment on this point:

"The background error matrix is computed using the NMC method (Parrish and Derber, 1992;
Barker et al. 2004) applied to the HyMeX-SOP1 (Hydrological cycle in the Mediterranean
Experiment – First Special Observing Period occurred in the period 6 September-6 November
2012; Ducroq et al., 2014). This choice is motivated by the fact that HyMeX-SOP1 contains
several heavy precipitation events over Italy and the background error matrix is
representative of the convective environment of the cases considered in this paper. In
particular, 10 out of 20 declared IOP (Intense Observing Period) of HyMeX-SOP1 occurred in
Italy (Ferretti et al., 2014). On the contrary, the period of September 2017, especially before
the events selected in this study was characterized by fair and stable weather conditions over
Italy and the background error matrix for September 2017 is less representative of the
convective environments that characterise the events of this paper."

RC: PDF file with technical corrections:

AR: Considering the pdf file, all corrections were accepted. There are, however, few points that
need a short discussion. They are listed below.

Elements of novelty of the paper: we highlighted better the elements of novelty of this paper in the
introduction. We wrote:

"This paper presents for the first time the assimilation of the radar reflectivity factor in the
RAMS@ISAC model and shows how the assimilation of the radar reflectivity factor works together
with lighting data assimilation. Also, this paper shows how accurate in space and time can be the
forecast of the precipitation field using cloud scale observations over complex terrain, contributing
in this way to a number of works on the same subject."

Comment on Line 276: The frequency bias was not shown to keep the discussion concise. However,
was important to point out that the model has a wet bias, especially when assimilating radar reflectivity factor. For this purpose, we introduced the score that was used to highlight that the
model has a wet bias, nevertheless we didn't show any figure to keep the discussion more concise.
In the revised version of the paper the wet bias of the model is highlighted better and a discussion
on how to reduce the wet bias is added in the "Discussion and conclusions" section (Section 5).

In particular in the "Discussion and Conclusions" section we wrote:

"The wet bias of RAD and RADLI forecast is the main drawback of the results of this paper. To
reduce the moisture added by radar and lightning data assimilation further research is needed and
different approaches are possible (Fierro et al., 2016). In particular: a) assimilating for a shorter
time period (0-6h in this paper); b) reducing the length-scales of the 3D-Var in the horizontal
directions to limit the spreading of the innovations or assuming an innovation equal to zero for
grid points without lightning and with zero reflectivity factor observed and simulated; c) reducing
the amount of water vapour added to the model (for example reducing the values of A and B
constants for lightning data assimilation or relaxing the request of saturation when radar
reflectivity is observed in areas where the model has zero reflectivity); d) adding moisture to a
shallower vertical level.
It is also noted that a combination of heating and moistening could provide the same buoyancy
with less water vapour addition (Marchand and Fulberg, 2014)."

While in the supplemental material (Section S2) we wrote:

"It is finally noted that RAD and RADLI have high POD values for all thresholds, nevertheless their
ETS is below that of LIGHT and SAT up to 32 mm/3h (RADLI) and 42 mm/3h (RAD). This behavior is
caused by the larger number of false alarms given by assimilating radar reflectivity factor
compared to simulations assimilating lightning. This result shows again that RAD and RADLI
configurations have a wet frequency bias. In particular, the frequency bias of RAD and RADLI
configuration is about 3 between 20 and 40 mm/3h."

Line 306: There was a typing error into equation 2 (equation 1 in the revised version of the paper).
The correct equation is:

$$q_v = Aq_s + Bq_s \tanh(CX)(1 - \tanh(Dq_g^x))$$

and $q_g$ is in the last term. We also note that the number of lightning assimilated by the model is
larger than that reported in the first submission of the paper for both cases for a mistake we did in
checking the numbers. The correct number of assimilated flashes is reported at the start of this
answer. The results, however, were obtained using the correct number of flashes.

Line 314: "The check and eventual substitution of the water vapor is performed every five minutes
and it is made only in the charging zone (0 °C, -25°C)." To better explain this choice, we
reformulated the sentence adding references to the charging zone:

"The check and eventual substitution of the water vapor is performed every five minutes and it is
made within the mixed phase layer zone (0 °C, -25°C), wherein electrification processes are the
most active (Takahashi 1978, Emersic and Sounders, 2010; Fierro et al., 2015)."

Line 314: the comment is: could you add some more details about this quality control?

We added two references. We wrote:

[revised manuscript text omitted]

**ANSWER to the Reviewer #3**

Review of the manuscript
„The impact of lightning and radar data assimilation on the performance of very short term rainfall forecasts for two case studies in Italy"
by
Stefano Federicio, Rosa Claudia Torcasio, Elenio Aviolo, Olivier Caumont, Mario Montopoli, Luca Baldini, Gianfranco Vulpiani and Stefano Dietrich
The study discusses the impact of the assimilation of lightning and radar reflectivity data on the performance of very short-range rainfall forecasts for two convective case studies in Italy. They showed that especially the combined assimilation of both observation types has a clear and positive impact on the forecast performance.
The manuscript is interesting and tackles a very important subject, since the forecast of severe precipitation is still a major weakness of current forecast systems. However, you have to provide more information on how the two observation types are assimilated. The equations and the text you provide are not enough to convince the reader that the methodology with all its coefficients is scientifically justified. Furthermore, I am concerned about the coarse vertical resolution you applied for the simulations.
Therefore, major revisions of the methodology section are necessary before I can suggest the publication of the manuscript.
In the following, I spilt my judgement into major and minor comments.

We acknowledge the reviewer for the useful comments on the paper, both in the general comments section and in the pdf file. Our answers are in red. The Author Replies (AR) to the Reviewer Comments (RC) are in blue.

AR: Before discussing in details the RCs, it's import to outline that the subject paper contained two typos. The first refers to the length scale of the background error matrix in the x and y directions that varies between 14 and 25 km and not , between 20 and 30 km, as erroneously reported in the manuscript. The second refers to the correct lightning number for each day are 82 331 for the 9 September, 291 164 for the 10 September (170 000 is written into the manuscript) and 105 467 for the 16 September (60 000 written in the manuscript). Despite the typos, the results shown in the paper were obtained using the correct number of flashes and the correct length scales in the background error matrix.

We acknowledge the reviewer for the interesting remarks, which helped to improve the paper. In general, we note that both lightning and radar reflectivity factor data assimilation sections have been expanded, as requested by the reviewer, and the "Data and Methods" section should be much clearer in the revised version of the paper. In the following there are important points raised by the reviewer (the choice of the background error matrix, or superobs for example), which we can only comment without performing again the simulations. Because they are important for the data assimilation system and we can only comment on them, we prefer to add the word "preliminary" to the title of the paper. In this way we highlight that important points of the physical options of the software are still to be fully tested.

The new title proposed is: "Preliminary results of the impact of lightning and radar reflectivity factor data assimilation on the very short term rainfall forecasts of RAMS@ISAC: application to two case studies in Italy"

The revised paper has supplemental material where we investigate the following two points: a) the relative contribution of lightning and radar data assimilation to the behavior of total water mass in the simulation; b) the sensitivity of the rainfall VSF to lightning data formulation.

**RC: Major comments**
- You show the horizontal dimensions and resolutions of your domains. However, for the vertical dimension you only write that it covers the troposphere and the lower stratosphere. How many levels do you use up to which height? How many of these levels are in the boundary layer? These are important information strongly influencing your results. This needs to be mentioned in the text. I found the total number of vertical levels and the model top height in the table you provide. However, to my opinion a horizontal resolution of 1 km combined with only 36 levels up to more than 22 km height is way too coarse to adequately describe the developing processes.

AR: The set-up of the RAMS@ISAC in this paper is the same as the operational setting, with the exception of the usage of domain D3 for the Livorno flood. In the operational setting a compromise must be chosen between the grid resolution and the computational time. While a future release of the operational setting will use more vertical levels (42), for the current year we are still using 36 levels. Among them 10 are below 1 km, 15 below 2 km and 18 below 3 km. The level 21 is at 5200 m in the terrain following coordinates used by RAMS. Above 6 km the model levels are more than 1000 m apart, with a maximum of 1200 m for the vertical layer at the model top.
Of course more vertical levels are useful to resolve important processes in the vertical as, as in cases of fronts, Planetary Boundary Layer processes, clouds etc., nevertheless a compromise between vertical resolution and computing time is necessary. Note, also, that this vertical grid was used with success in several heavy precipitation events over Italy.  We wrote (Section 3.1):

"The resolutions and the extensions of the grids in the vertical direction is the same for the three domains. The vertical  grid covers the troposphere and the lower stratosphere. Vertical levels have different spacings and are more packed close to the ground. Among the 36 levels used in this paper 10 are below 1 km, 15 below 2 km and 18 below 3 km. The first vertical level is at 24 m above the surface in the terrain following coordinates used by RAMS@ISAC, the level 21 is at 5200 m. Above 6 km the model levels are about 1000 m apart, with a maximum of 1200 m for the vertical layer at the model top."

RC: You write that the R10 simulation is applied as lateral boundary condition for the inner domain simulations and that you use assimilation in the inner domains. O.k. so far – but do you adjust the lateral boundaries provided by R10 to the new situation after the assimilation? If not they may negatively influence your forecast. If you think that this is not necessary for your short-term forecasts, this has to be mentioned in the text.

AR: This point is related with the operational setting of RAMS@ISAC. Updating the R10 domain to the new situation after data assimilation is beneficial for the simulation.

Nevertheless, this would require additional simulations of the R10 model increasing the
computing time for the whole chain. We added the following comment (in Section 3.1):
"The R10 run is not updated after the acquisition of new data by the analysis system and
this is a limitation of the results shown in this paper."
RC: The description of the lightning data assimilation is too short. The reader has no
chance to understand equation (2) with all its coefficients without further explanation. In
the text, you mention graupel mixing ratio qg, but the equation only contains qs. What is
qs or should it be qg? The way you present it sounds like "Voodoo".
AR: Lightning section was mostly rewritten in the revised version of the paper and
extended to consider this and other comments of the reviewers. In the revised version of
the paper it should be more readable. The correct form of Equation (2) (Eqn. (1) in the
revised version of the paper is):

$$q_v = Aq_s + Bq_s \tanh(CX)(1 - \tanh(Dq_g^\alpha))$$

and now $q_g$ is present.
RC: How do the assimilation system deals with sharp gradients along the vertical profile
when you only adjust the profile in a certain height region?
AR: Lightning data assimilation as well as radar data assimilation can produce sharp
gradients in the vertical directions. For lightning data assimilation a nudging is performed
to avoid a direct insertion of the data in the model, while in the case of radar the data are
directly introduced in RAMS@ISAC. Nevertheless, our experience with RAMS@ISAC
shows that, at least for the setting of this paper, the sharp gradients introduced in the
model do not produce incorrect results or blowing up of the model. We highlighted this
point into the paper (Section 3.3):
"Lightning and radar data assimilation may produce sharp gradients in vertical direction
caused by the addition of water vapour to specific layers. In the case of lightning, the
water vapour is added by nudging to reduce the sharp gradients. However, radar data
assimilation, which accounts for the largest mass of water added to RAMS@ISAC (see
Section S.1 of the supplemental material), directly inserts the water vapour into the
model. Our experience with RAMS@ISAC, however, shows that results are reliable and
the sudden addition of water vapour doesn't cause shocks to the model simulation,
despite the notable gradients of specific humidity."
RC: You only adjust qv? How do you make sure that this results in more precipitation? Do
you tune this with the coefficients? If yes – you have to include the information in the text.
AR: The increase/decrease of the water vapour depends on the data assimilation.
Lightning can only increase the water vapour while radar can increase or decrease the
water vapour. However, if the added/removed water vapour determines more/less
precipitation depends on the physical and dynamical processes occurring in
RAMS@ISAC and no specific tuning is done. The only exception is that if, after the data
assimilation, the model is oversaturated, the water vapour is reduced to the saturated
value (at the RAMS@ISAC temperature).
We added the following sentence at the end of Section 3.3 "Data and Methods": "It is
finally noted that the data assimilation increase/decrease the water vapour into the model
depending on the cases. The eventual increase/decrease of the forecasted rainfall depends on the physical and dynamical processes occurring into the meteorological
model, without any specific tuning."
RC: I understand why you reduce the resolution of the radar data. But is a pure sampling
the best method? Usually one uses a kind of "super obbing" to avoid the implementation
of errors (e.g. by insects) or extremes. Please comment on this and add explanation why
you choose sampling.
AR: This is an aspect that need to be improved in future version of the software. Using
superobs does a better job compared to the sampling of the data for the reasons stated
by the reviewer and superobs improve the performance of the data assimilation. The only
point that favours the sampling method compared to superobs is that the latter could
increase the correlation among the observations' errors. We highlighted the point in
Section 3.3 by writing:
"It is important to note the pure sampling of the data could result in implementation of
errors (for example reflectivity given by insects or birds) or extremes. Creating
superobservations would reduce this problem, the main drawback being the missing of
very localised phenomena. While the aim of this paper is to present the update of the
data assimilation system of RAMS@ISAC and its application to two challenging cases,
the problem of using superobservations will be considered in future studies because it
impacts the results."
RC: The error value of 1 to 3 dBz seems to be too small, making the system very sensitive
to the radar data. Especially when combining this with a pure sampling of the radar data
sounds dangerous to me. Please explain why you use this error value.
AR: The choice of a small error for the reflectivity factor is motivated by two reasons: a)
the data are carefully checked by the Civil protection Department; b) the performance of
the control simulation, not assimilating any data, is rather poor for the case studies.
Nevertheless, the choice could not be optimal for other cases. We highlighted this point
in Section 3.3 by writing:
"The error of radar data is assumed small (1dBz) for two reasons: a) reflectivity data are
carefully checked by the Civil Protection Department; b) the performance of the control
simulation, not assimilating any data, is rather poor for the case studies. This setting,
however, could not be optimal for cases when the control forecast performs better and
different choices should be done for those cases."
RC: You mention that cross correlations between the variables are neglected in this
study. Do you neglect them in the observation operator only or in the assimilation
system? Every assimilation system needs cross correlations between the variables e.g. to
spread the information of the observations horizontally and vertically.
AR: The expression we used is not correct. We neglect cross correlations among different
variables. In the current form, the data assimilation system uses the variables: (u=zonal
velocity, v=meridional velocity, T=temperature, q=water vapor mixing ratio). For the
results of this paper we neglected cross correlations in the **B** matrix among different
variables (specifically (u,q), (v,q) (T,q)), while we maintained the error correlations among
different levels and in the horizontal plane of q to spread the innovations in the vertical
and horizontal directions (the latter being shown in Figure 14b of the revised paper). This is a point that need to be improved in future versions of the software, by considering also cross correlations among different variables. We will precise better this point adding the following sentence (Section 3.3):

"Cross correlations among different variables of the data assimilation system are neglected in this study and the application of the RAMS-3DVar affects the water vapour mixing ratio only. Cross correlations among different variables can improve the performance of data assimilation system, and an example of their impact in the RAMS-3DVar is shown in Federico (2013). Nevertheless, the impact the cross correlations among different variables of the 3D-Var in the precipitation VSF will be explored in future works.".

RC: - Mention not only that the forward operator for reflectivity is from the WSM6 scheme of WRF – also mention the equation of the forward operator.

AR: We added the expression of the forward radar operator in the supplemental material (Section S4) to avoid excessive paper length. A reference was put into the paper to the reference material (Section 3.3).

**RC: Minor comments**
- Although the text is well readable, the language can be improved. Although, I am not a native speaker, I stumbled over several things. Here some examples:
- Page 4, line 113: ... case studies occurred in September 2017 ...
- Page 5, line 135: ... developed to the lee ...
- Page 5, line 151: Also notable is the feeding ...
- Page 5, line 158: ... is also clearly seen in the radar ...
- Page 5, line 163: ... can be noted over central-northern Italy.
- Page 6, line 165: ... cloud system was active for several ...
- Page 6, line 168: ... were recorder during the day; ...
- Page 6, line 170: ... from 00 UTC ...
- Page 6, line 186: ... occurred within a few hours.
- Page 6, line 192: illustrated better than represented? We chose illustrated.
- Page 6, line 195: ... interaction between the air-masses and the Western Alps generated a depression ...
- Page 7, line 197: ... it is noted that divergent ...
- Page 7, line 198: ... it is apparent that the equivalent ... We wrote "it is evident the equivalent...'
- Page 7, line 201: ... low pressure system ...Not applicable in the revised version of the paper.
- Page 7, line 204: From a synoptic point of view, ...
- Page 7, line 206: ... more intense than the Serano case ...
- Page 7, line 211: ... recorded over Italy, following ...
- Page 7, line 213: ... for the Serano case.
- Page 7, line 215: ... it is well evident that the cloud system ...

AR: All the above points were corrected, with the exceptions indicated in blu.

RC: Abstract: Lines 29 to 31. Do you need this sentence? To my opinion, the sentence above is enough.

AR: Deleted.
RC: Abstract: Merge lines 32 to 34 with the paragraph above.
AR: Done.
RC: Mention once in the text why you use the term "reflectivity factor" instead of
reflectivity.
AR: Ok. We added the following footnote to express the point, the first time the term
"reflectivity factor" is introduced in the paper (excluding the abstract). "Throughout the
paper we use the expression radar reflectivity factor, which is the quantity provided by
the radar (and expressed in $mm^6m^{-3}$ or dBz) after conversion from the received power.
The radar reflectivity factor is different from reflectivity and is obtained in the special case
of Rayleigh approximation. Reflectivity is not the quantity that radars usually provide and
display on their screens although most of people refer to it."
RC: For me "Probability of dectetion" and "Hit rate" is the same. What you defined with
Hit      rate      is      the      so-called      "Hit      score"      e.g.      following
https://iri.columbia.edu/~jhansen/mason11july.pdf
AR: Thank you for noting this point. We used the definition of the Wilks book (Chapter 7).
However, in the revised version of the paper only the POD is considered. Following the
remarks of the reviewer#1 we deleted the equations for the scores. This helped to have a
shorter paper. Also the scores were put in tables (not graphs), following a comment of
reviewer #2. Graphs of the scores are presented in the supplemental material (Section
S2).
RC: Translate the acronym "GPROF"
AR: It stands for Goddard Profiling Algorithm (added into the paper).
RC: Before you start to discuss the result, mention once how you name your different
experiments. It gets clear during the reading, but if you mention it once, you do not need
to repeat it later during the manuscript.
AR: Thanks for noting this point (also requested by reviewer#2). We did it at the start of
Section 4 (results). We also added a table (Table 3) to better clarify the point. Also the
supplemental material of the paper has a table (Table S1) specifying the types of
simulations considered.
RC: Page 19, line 570: you mean LIGHT instead of FLASH?
AR: Yes. It was an error. Corrected.
RC: Page 22, lines 673 to 677: You mention that reflectivity data assimilation helps to
better represent light precipitation events and lightning data helps to represent strong events. One abstract later (lines 684 to 686) you argue the other way round. So, the influence of the different observation types also depend on the situation.

AR: Thanks for noting the point. We added the following sentence: "These results show also that the influence of different observations depends on the meteorological situation."

RC: Page 23, line 702: Start a new paragraph and sentence after the promising results and the drawbacks.

AR: Done.

RC: What do you think is the reason for the increased false alarm rate in the RADLI forecasts? How do you think you can improve the situation in future versions of the system?

AR: The reason for having more false alarms in RADLI forecasts compared to other configurations is the larger amount of water vapour added to this kind of simulation, a direct consequence of the addition of water vapour given by both radar and lightning. In the supplemental material of this paper the evolution of the water vapour mass is presented, including the assimilation stage. Results, as expected, show the largest amount of water (not only in the vapour form) added to RADLI by data assimilation. Possible ways to decrease false alarms in future versions of the software are shortly introduced in Section 5 (Conclusion and discussion). We wrote: "To reduce the moisture added by radar and lightning data assimilation further research is needed and different approaches are possible (Fierro et al., 2016). In particular: a) assimilating for a shorter time (0-6h in this paper); b) reducing the length-scales of the 3D-Var in the horizontal directions to limit the spreading of the innovations or assuming an innovation equal to zero for grid points without lightning and with zero reflectivity factor; c) reducing the amount of water vapour added to the model (for example reducing the values of A and B constants for lightning data assimilation or relaxing the request of saturation when radar reflectivity is observed in areas where the model has zero reflectivity); d) adding moisture to a shallower vertical layer.

It is also noted that a combination of heating and moistening could provide the same buoyancy with less water vapour addition (Marchand and Fulberg, 2014) and this approach could be used in future studies."

RC: Figures: Increase the sizes of the figures. You have space on the page to do this. If not put only two instead of three Figures on one page as done for Figures 10 to 12 on page 37.

AR: We enlarged the figures in the revised version of the paper. We will consider this point when revising the proofs of the paper, if accepted for publication.

LIST OF THE MAJOR CHANGES MADE IN THE PAPER

-The Title was changed.

-The introduction was partially rewritten, especially in the part considering lightning data assimilation.

-Section 3.1 had major revisions to specify the vertical grid of RAMS@ISAC model and to clarify the role of domain D3 in the simulations of the Livorno case study.

-Section 3.2 underwent a substantial rewriting to clarify better how the radar reflectivity factor data assimilation is performed. An example of analysis is introduced and all the details of the 3D-Var setting have been hopefully clarified.

-Section 3.3 underwent a substantial rewriting to clarify better how lightning data assimilation works. A sensitivity experiment to the formulation of lightning data assimilation was added in the supplemental material.

-Section 4 underwent major revision. Three VSF are considered (revised version) instead of five (first submission) to reduce the length of the paper and to avoid repetitive comments. Scores were computed for three neighborhood radii and were put in tables (Table 4-6) according to reviewers' comments.

[revised manuscript text omitted]


Table 6 ETS and POD scores for three different neighbourhood radii. Scores are computed over the
domain D2.

| Threshold (mm/3h) | ETS nearest neighboorhood (CTRL, RAD, LIGHT, RADLI) | POD nearest neighboorhood (CTRL, RAD, LIGHT, RADLI) | ETS 25 km (CTRL, RAD, LIGHT, RADLI) | POD 25 km (CTRL, RAD, LIGHT, RADLI) | ETS 50 km (CTRL, RAD, LIGHT, RADLI) | POD 50 km (CTRL, RAD, LIGHT, RADLI) |
|---|---|---|---|---|---|---|
| 1 | (0.41,0.63,0.61,0.65) | (0.66,0.89,0.89,0.93) | (0.79,0.83,0.82,0.83) | (0.89,0.95,0.95,0.96) | (0.88,0.92,0.93,0.94) | (0.93,0.97,0.98,0.98) |
| 6 | (0.2,0.4,0.39,0.47) | (0.43,0.82,0.77,0.88) | (0.45,0.63,0.71,0.76) | (0.63,0.90,0.95,0.96) | (0.72,0.86,0.88,0.92) | (0.82,0.96,0.97,0.96) |
| 10 | (0.,0.24,0.18,0.28) | (0.14,0.78,0.55,0.80) | (0.14,0.47,0.58,0.62) | (0.24,0.86,0.82,0.93) | (0.32,0.91,0.96,0.95) | (0.35,0.95,0.97,0.97) |
| 20 | (-0.03,0.18,0.13,0.22) | (0.01,0.81,0.30,0.80) | (0.09,0.46,0.57,0.61) | (0.11,0.86,0.59,0.90) | (0.15,0.84,0.91,0.96) | (0.15,0.90,0.92,0.97) |
| 30 | (-0.02,0.22,0.13,0.28) | (0.,0.90,0.23,0.88) | (0.01,0.79,0.46,0.80) | (0.01,0.93,0.47,0.94) | (0.02,0.95,0.93,0.99) | (0.02,0.95,0.93,0.99) |
| 40 | (-0.1,0.24,0.08,0.36) | (0.,0.83,0.12,0.89) | (0.01,0.83,0.37,0.83) | (0.02,0.97,0.38,0.97) | (0.1,0.97,0.95,0.98) | (0.02,0.98,0.95,0.98) |
| 50 | (-0.01,0.27,0.,0.43) | (0.,0.67,0.,0.92) | (0.,0.90,0.,0.90) | (0.,0.94,0.,0.96) | (0.,0.96,0.,0.96) | (0.,0.96,0.,0.96) |

**FIGURES**

[Figure]

Figure 1: Daily precipitation (P) [mm] over Italy on 16 September 2017. Only raingauges observing at least 0.2 mm/day are shown. The first number in the figure title within brackets represents the available raingauges, while the second number represents raingauges observing at least 0.2 mm/3h. The lowest precipitation class is represented by smaller dots, the largest by a red square. The locations of Città di Castello and Mount Serano are indicated.

a)

HGT[m] - WSP[m/s] - 20170916000000 - z=  500 hPa

[Figure]

CONTOUR FROM 245 TO 269 BY 3

5530 5558 5587 5615 5644 5673 5701 5730 5759 5787 5816 5844 5873

0.313E+02
MAXIMUM VECTOR

b)

Thetae[K] − slp [hPa] − 20170916000000 − z=    24 m

CONTOUR FROM 997 TO 1017 BY 2

297.2  300.4  303.6  306.8  310.1  313.3  316.5  319.8  323.0  326.2  329.4  332.7  335.9  339.1

0.125E+02

¶
¶
¶

[Figure]

Spostato in giù [1]:

Spostato (inserimento) [1]

[revised manuscript text omitted]

Tipo di carattere: 12 pt

| Pagina 49: [2] Eliminato | stefano federico | 14/01/19 12:04:00 |
|---|---|---|

| Pagina 57: [3] Eliminato | stefano federico | 21/03/19 07:27:00 |
|---|---|---|

| Pagina 65: [4] Formattato | stefano federico | 02/02/19 19:23:00 |
|---|---|---|

Tipo di carattere: 12 pt

| Pagina 65: [4] Formattato | stefano federico | 02/02/19 19:23:00 |
|---|---|---|

Tipo di carattere: 12 pt

| Pagina 65: [5] Formattato | stefano federico | 02/02/19 19:23:00 |
|---|---|---|

Bordo: : (Nessun bordo)

| Pagina 65: [6] Formattato | stefano federico | 02/02/19 19:23:00 |
|---|---|---|

Bordo: : (Nessun bordo)

| Pagina 65: [7] Formattato | stefano federico | 02/02/19 19:23:00 |
|---|---|---|

Bordo: : (Nessun bordo)

| Pagina 65: [8] Formattato | stefano federico | 02/02/19 19:23:00 |
|---|---|---|

Bordo: : (Nessun bordo)

| Pagina 65: [9] Formattato | stefano federico | 02/02/19 19:23:00 |
|---|---|---|

Bordo: : (Nessun bordo)

| Pagina 65: [10] Formattato | stefano federico | 02/02/19 19:23:00 |
|---|---|---|

Bordo: : (Nessun bordo)

| Pagina 65: [11] Formattato | stefano federico | 02/02/19 19:23:00 |
|---|---|---|

Bordo: : (Nessun bordo)

| Pagina 65: [12] Formattato | stefano federico | 02/02/19 19:23:00 |
|---|---|---|

Bordo: : (Nessun bordo)

| Pagina 65: [13] Formattato | stefano federico | 02/02/19 19:23:00 |
|---|---|---|

Bordo: : (Nessun bordo)

| Pagina 65: [14] Formattato | stefano federico | 02/02/19 19:23:00 |
|---|---|---|

Bordo: : (Nessun bordo)

| Pagina 65: [15] Formattato | stefano federico | 02/02/19 19:23:00 |
|---|---|---|

Bordo: : (Nessun bordo)

| Pagina 65: [16] Formattato | stefano federico | 02/02/19 19:23:00 |
|---|---|---|

Bordo: : (Nessun bordo)

| Pagina 65: [17] Formattato | stefano federico | 02/02/19 19:23:00 |
|---|---|---|

Bordo: : (Nessun bordo)

| Pagina 65: [18] Formattato | stefano federico | 02/02/19 19:23:00 |
|---|---|---|

Bordo: : (Nessun bordo)

| Pagina 65: [19] Formattato | stefano federico | 02/02/19 19:23:00 |
|---|---|---|

Bordo: : (Nessun bordo)

| Pagina 65: [20] Formattato | stefano federico | 02/02/19 19:23:00 |
|---|---|---|

Bordo: : (Nessun bordo)

| Pagina 65: [21] Formattato | stefano federico | 02/02/19 19:23:00 |
|---|---|---|

Bordo: : (Nessun bordo)

| Pagina 65: [22] Formattato | stefano federico | 02/02/19 19:23:00 |
|---|---|---|

Bordo: : (Nessun bordo)

| Pagina 65: [23] Formattato | stefano federico | 02/02/19 19:23:00 |
|---|---|---|

Bordo: : (Nessun bordo)

| Pagina 65: [24] Formattato | stefano federico | 02/02/19 19:23:00 |
|---|---|---|

Bordo: : (Nessun bordo)

| Pagina 65: [25] Formattato | stefano federico | 02/02/19 19:23:00 |
|---|---|---|

Bordo: : (Nessun bordo)

| Pagina 65: [26] Formattato | stefano federico | 02/02/19 19:23:00 |
|---|---|---|

Bordo: : (Nessun bordo)

| Pagina 65: [27] Formattato | stefano federico | 02/02/19 19:23:00 |
|---|---|---|

Bordo: : (Nessun bordo)

| Pagina 65: [28] Formattato | stefano federico | 02/02/19 19:23:00 |
|---|---|---|

Bordo: : (Nessun bordo)

| Pagina 65: [29] Formattato | stefano federico | 02/02/19 19:23:00 |
|---|---|---|

Bordo: : (Nessun bordo)

| Pagina 65: [30] Formattato | stefano federico | 02/02/19 19:23:00 |
|---|---|---|

Bordo: : (Nessun bordo)

| Pagina 65: [31] Formattato | stefano federico | 02/02/19 19:23:00 |
|---|---|---|

Bordo: : (Nessun bordo)

| Pagina 65: [32] Formattato | stefano federico | 02/02/19 19:23:00 |
|---|---|---|

Bordo: : (Nessun bordo)

| Pagina 65: [33] Formattato | stefano federico | 02/02/19 19:23:00 |
|---|---|---|

Bordo: : (Nessun bordo)

| Pagina 65: [34] Formattato | stefano federico | 02/02/19 19:23:00 |
|---|---|---|

Bordo: : (Nessun bordo)

| Pagina 65: [35] Formattato | stefano federico | 02/02/19 19:23:00 |
|---|---|---|

Bordo: : (Nessun bordo)

| Pagina 65: [36] Formattato | stefano federico | 02/02/19 19:23:00 |
|---|---|---|

Bordo: : (Nessun bordo)

| Pagina 65: [37] Formattato | stefano federico | 02/02/19 19:23:00 |
|---|---|---|

Bordo: : (Nessun bordo)

| Pagina 65: [38] Formattato | stefano federico | 02/02/19 19:23:00 |
|---|---|---|

Bordo: : (Nessun bordo)

| Pagina 65: [39] Formattato | stefano federico | 02/02/19 19:23:00 |
|---|---|---|

Bordo: : (Nessun bordo)

| Pagina 65: [40] Formattato | stefano federico | 02/02/19 19:23:00 |
|---|---|---|

Bordo: : (Nessun bordo)

| Pagina 65: [41] Formattato | stefano federico | 02/02/19 19:23:00 |
|---|---|---|

Bordo: : (Nessun bordo)

| Pagina 65: [42] Formattato | stefano federico | 02/02/19 19:23:00 |
|---|---|---|

Bordo: : (Nessun bordo)

| Pagina 65: [43] Formattato | stefano federico | 02/02/19 19:23:00 |
|---|---|---|

Bordo: : (Nessun bordo)

| Pagina 65: [44] Formattato | stefano federico | 02/02/19 19:23:00 |
|---|---|---|

Bordo: : (Nessun bordo)

| Pagina 65: [45] Formattato | stefano federico | 02/02/19 19:23:00 |
|---|---|---|

Bordo: : (Nessun bordo)

| Pagina 65: [46] Formattato | stefano federico | 02/02/19 19:23:00 |
|---|---|---|

Bordo: : (Nessun bordo)

| Pagina 65: [47] Formattato | stefano federico | 02/02/19 19:23:00 |
|---|---|---|

Bordo: : (Nessun bordo)

| Pagina 65: [48] Formattato | stefano federico | 02/02/19 19:23:00 |
|---|---|---|

Bordo: : (Nessun bordo)

| Pagina 65: [49] Formattato | stefano federico | 02/02/19 19:23:00 |
|---|---|---|

Bordo: : (Nessun bordo)

| Pagina 65: [50] Formattato | stefano federico | 02/02/19 19:23:00 |
|---|---|---|

Bordo: : (Nessun bordo)

| Pagina 65: [51] Formattato | stefano federico | 02/02/19 19:23:00 |
|---|---|---|

Bordo: : (Nessun bordo)

| Pagina 65: [52] Formattato | stefano federico | 02/02/19 19:23:00 |
|---|---|---|

Bordo: : (Nessun bordo)

| Pagina 65: [53] Formattato | stefano federico | 02/02/19 19:23:00 |
|---|---|---|

Bordo: : (Nessun bordo)

| Pagina 65: [54] Formattato | stefano federico | 02/02/19 19:23:00 |
|---|---|---|

Bordo: : (Nessun bordo)

| Pagina 65: [55] Formattato | stefano federico | 02/02/19 19:23:00 |
|---|---|---|

Bordo: : (Nessun bordo)

| Pagina 65: [56] Formattato | stefano federico | 02/02/19 19:23:00 |
|---|---|---|

Bordo: : (Nessun bordo)

| Pagina 65: [57] Formattato | stefano federico | 02/02/19 19:23:00 |
|---|---|---|

Bordo: : (Nessun bordo)

| Pagina 65: [58] Formattato | stefano federico | 02/02/19 19:23:00 |
|---|---|---|

Bordo: : (Nessun bordo)

| Pagina 65: [59] Formattato | stefano federico | 02/02/19 19:23:00 |
|---|---|---|

Bordo: : (Nessun bordo)

| Pagina 65: [60] Formattato | stefano federico | 02/02/19 19:23:00 |
|---|---|---|

Bordo: : (Nessun bordo)

| Pagina 65: [61] Formattato | stefano federico | 02/02/19 19:23:00 |
|---|---|---|

Bordo: : (Nessun bordo)

| Pagina 65: [62] Formattato | stefano federico | 02/02/19 19:23:00 |
|---|---|---|

Bordo: : (Nessun bordo)

| Pagina 65: [63] Formattato | stefano federico | 02/02/19 19:23:00 |
|---|---|---|

Bordo: : (Nessun bordo)

| Pagina 65: [64] Formattato | stefano federico | 02/02/19 19:23:00 |
|---|---|---|

Bordo: : (Nessun bordo)

| Pagina 65: [65] Formattato | stefano federico | 02/02/19 19:23:00 |
|---|---|---|

Bordo: : (Nessun bordo)

| Pagina 65: [66] Formattato | stefano federico | 02/02/19 19:23:00 |
|---|---|---|

Bordo: : (Nessun bordo)

| Pagina 65: [67] Formattato | stefano federico | 02/02/19 19:23:00 |
|---|---|---|

Bordo: : (Nessun bordo)

| Pagina 65: [68] Formattato | stefano federico | 02/02/19 19:23:00 |
|---|---|---|

Bordo: : (Nessun bordo)

| Pagina 65: [69] Formattato | stefano federico | 02/02/19 19:23:00 |
|---|---|---|

Bordo: : (Nessun bordo)

| Pagina 65: [70] Formattato | stefano federico | 02/02/19 19:23:00 |
|---|---|---|

Bordo: : (Nessun bordo)

| Pagina 65: [71] Formattato | stefano federico | 02/02/19 19:23:00 |
|---|---|---|

Bordo: : (Nessun bordo)

| Pagina 65: [72] Formattato | stefano federico | 02/02/19 19:23:00 |
|---|---|---|

Bordo: : (Nessun bordo)

| Pagina 65: [73] Formattato | stefano federico | 02/02/19 19:23:00 |
|---|---|---|

Bordo: : (Nessun bordo)

| Pagina 65: [74] Formattato | stefano federico | 02/02/19 19:23:00 |
|---|---|---|

Bordo: : (Nessun bordo)

| Pagina 65: [75] Formattato | stefano federico | 02/02/19 19:23:00 |
|---|---|---|

Bordo: : (Nessun bordo)

| Pagina 65: [76] Formattato | stefano federico | 02/02/19 19:23:00 |
|---|---|---|

Bordo: : (Nessun bordo)

| Pagina 65: [77] Formattato | stefano federico | 02/02/19 19:23:00 |
|---|---|---|

Bordo: : (Nessun bordo)

| Pagina 65: [78] Formattato | stefano federico | 02/02/19 19:23:00 |
|---|---|---|

Bordo: : (Nessun bordo)

| Pagina 65: [79] Formattato | stefano federico | 02/02/19 19:23:00 |
|---|---|---|

Bordo: : (Nessun bordo)

| Pagina 65: [80] Formattato | stefano federico | 02/02/19 19:23:00 |
|---|---|---|

Bordo: : (Nessun bordo)

| Pagina 65: [81] Formattato | stefano federico | 02/02/19 19:23:00 |
|---|---|---|

Bordo: : (Nessun bordo)

| Pagina 65: [82] Formattato | stefano federico | 02/02/19 19:23:00 |
|---|---|---|

Bordo: : (Nessun bordo)

| Pagina 65: [83] Formattato | stefano federico | 02/02/19 19:23:00 |
|---|---|---|

Bordo: : (Nessun bordo)

| Pagina 65: [84] Formattato | stefano federico | 02/02/19 19:23:00 |
|---|---|---|

Bordo: : (Nessun bordo)

| Pagina 65: [85] Formattato | stefano federico | 02/02/19 19:23:00 |
|---|---|---|

Bordo: : (Nessun bordo)

| Pagina 65: [86] Formattato | stefano federico | 02/02/19 19:23:00 |
|---|---|---|

Bordo: : (Nessun bordo)

| Pagina 65: [87] Formattato | stefano federico | 02/02/19 19:23:00 |
|---|---|---|

Bordo: : (Nessun bordo)

| Pagina 65: [88] Formattato | stefano federico | 02/02/19 19:23:00 |
|---|---|---|

Bordo: : (Nessun bordo)

| Pagina 65: [89] Formattato | stefano federico | 02/02/19 19:23:00 |
|---|---|---|

Bordo: : (Nessun bordo)

| Pagina 65: [90] Formattato | stefano federico | 02/02/19 19:23:00 |
|---|---|---|

Bordo: : (Nessun bordo)

| Pagina 65: [91] Formattato | stefano federico | 02/02/19 19:23:00 |
|---|---|---|

Bordo: : (Nessun bordo)

| Pagina 65: [92] Formattato | stefano federico | 02/02/19 19:23:00 |
|---|---|---|

Bordo: : (Nessun bordo)

| Pagina 65: [93] Formattato | stefano federico | 02/02/19 19:23:00 |
|---|---|---|

Bordo: : (Nessun bordo)

| Pagina 65: [94] Formattato | stefano federico | 02/02/19 19:23:00 |
|---|---|---|

Bordo: : (Nessun bordo)

| Pagina 65: [95] Formattato | stefano federico | 02/02/19 19:23:00 |
|---|---|---|

Bordo: : (Nessun bordo)

| Pagina 65: [96] Formattato | stefano federico | 02/02/19 19:23:00 |
|---|---|---|

Bordo: : (Nessun bordo)

| Pagina 65: [97] Formattato | stefano federico | 02/02/19 19:23:00 |
|---|---|---|

Bordo: : (Nessun bordo)

| Pagina 65: [98] Formattato | stefano federico | 02/02/19 19:23:00 |
|---|---|---|

Bordo: : (Nessun bordo)

| Pagina 66: [99] Formattato | stefano federico | 02/02/19 19:23:00 |
|---|---|---|

Bordo: : (Nessun bordo)

| Pagina 66: [100] Formattato | stefano federico | 02/02/19 19:23:00 |
|---|---|---|

Bordo: : (Nessun bordo)

| Pagina 66: [101] Formattato | stefano federico | 02/02/19 19:23:00 |
|---|---|---|

Bordo: : (Nessun bordo)

| Pagina 66: [102] Formattato | stefano federico | 02/02/19 19:23:00 |
|---|---|---|

Bordo: : (Nessun bordo)

| Pagina 66: [103] Formattato | stefano federico | 02/02/19 19:23:00 |
|---|---|---|

Bordo: : (Nessun bordo)

| Pagina 66: [104] Formattato | stefano federico | 02/02/19 19:23:00 |
|---|---|---|

Bordo: : (Nessun bordo)

| Pagina 66: [105] Formattato | stefano federico | 02/02/19 19:23:00 |
|---|---|---|

Bordo: : (Nessun bordo)

| Pagina 66: [106] Formattato | stefano federico | 02/02/19 19:23:00 |
|---|---|---|

Bordo: : (Nessun bordo)

| Pagina 66: [107] Formattato | stefano federico | 02/02/19 19:23:00 |
|---|---|---|

Bordo: : (Nessun bordo)

| Pagina 66: [108] Formattato | stefano federico | 02/02/19 19:23:00 |
|---|---|---|

Bordo: : (Nessun bordo)

| Pagina 66: [109] Formattato | stefano federico | 02/02/19 19:23:00 |
|---|---|---|

Bordo: : (Nessun bordo)

| Pagina 66: [110] Formattato | stefano federico | 02/02/19 19:23:00 |
|---|---|---|

Bordo: : (Nessun bordo)

| Pagina 66: [111] Formattato | stefano federico | 02/02/19 19:23:00 |
|---|---|---|

Bordo: : (Nessun bordo)

| Pagina 66: [112] Formattato | stefano federico | 02/02/19 19:23:00 |
|---|---|---|

Bordo: : (Nessun bordo)

| Pagina 66: [113] Formattato | stefano federico | 02/02/19 19:23:00 |
|---|---|---|

Bordo: : (Nessun bordo)

| Pagina 66: [114] Formattato | stefano federico | 02/02/19 19:23:00 |
|---|---|---|

Bordo: : (Nessun bordo)

| Pagina 66: [115] Formattato | stefano federico | 02/02/19 19:23:00 |
|---|---|---|

Bordo: : (Nessun bordo)

| Pagina 66: [116] Formattato | stefano federico | 02/02/19 19:23:00 |
|---|---|---|

Bordo: : (Nessun bordo)

| Pagina 66: [117] Formattato | stefano federico | 02/02/19 19:23:00 |
|---|---|---|

Bordo: : (Nessun bordo)

| Pagina 66: [118] Formattato | stefano federico | 02/02/19 19:23:00 |
|---|---|---|

Bordo: : (Nessun bordo)

| Pagina 66: [119] Formattato | stefano federico | 02/02/19 19:23:00 |
|---|---|---|

Bordo: : (Nessun bordo)

| Pagina 66: [120] Formattato | stefano federico | 02/02/19 19:23:00 |
|---|---|---|

Bordo: : (Nessun bordo)

| Pagina 66: [121] Formattato | stefano federico | 02/02/19 19:23:00 |
|---|---|---|

Bordo: : (Nessun bordo)

| Pagina 66: [122] Formattato | stefano federico | 02/02/19 19:23:00 |
|---|---|---|

Bordo: : (Nessun bordo)

| Pagina 66: [123] Formattato | stefano federico | 02/02/19 19:23:00 |
|---|---|---|

Bordo: : (Nessun bordo)

| Pagina 66: [124] Formattato | stefano federico | 02/02/19 19:23:00 |
|---|---|---|

Bordo: : (Nessun bordo)

| Pagina 66: [125] Formattato | stefano federico | 02/02/19 19:23:00 |
|---|---|---|

Bordo: : (Nessun bordo)

| Pagina 66: [126] Formattato | stefano federico | 02/02/19 19:23:00 |
|---|---|---|

Bordo: : (Nessun bordo)

| Pagina 66: [127] Formattato | stefano federico | 02/02/19 19:23:00 |
|---|---|---|

Bordo: : (Nessun bordo)

| Pagina 66: [128] Formattato | stefano federico | 02/02/19 19:23:00 |
|---|---|---|

Bordo: : (Nessun bordo)

| Pagina 66: [129] Formattato | stefano federico | 02/02/19 19:23:00 |
|---|---|---|

Bordo: : (Nessun bordo)

| Pagina 66: [130] Formattato | stefano federico | 02/02/19 19:23:00 |
|---|---|---|

Bordo: : (Nessun bordo)

| Pagina 66: [131] Formattato | stefano federico | 02/02/19 19:23:00 |
|---|---|---|

Bordo: : (Nessun bordo)

| Pagina 66: [132] Formattato | stefano federico | 02/02/19 19:23:00 |
|---|---|---|

Bordo: : (Nessun bordo)

| Pagina 66: [133] Formattato | stefano federico | 02/02/19 19:23:00 |
|---|---|---|

Bordo: : (Nessun bordo)

| Pagina 66: [134] Formattato | stefano federico | 02/02/19 19:23:00 |
|---|---|---|

Bordo: : (Nessun bordo)

| Pagina 66: [135] Formattato | stefano federico | 02/02/19 19:23:00 |
|---|---|---|

Bordo: : (Nessun bordo)

| Pagina 66: [136] Formattato | stefano federico | 02/02/19 19:23:00 |
|---|---|---|

Bordo: : (Nessun bordo)

| Pagina 66: [137] Formattato | stefano federico | 02/02/19 19:23:00 |
|---|---|---|

Bordo: : (Nessun bordo)

| Pagina 66: [138] Formattato | stefano federico | 02/02/19 19:23:00 |
|---|---|---|

Bordo: : (Nessun bordo)

| Pagina 66: [139] Formattato | stefano federico | 02/02/19 19:23:00 |
|---|---|---|

Bordo: : (Nessun bordo)

| Pagina 66: [140] Formattato | stefano federico | 02/02/19 19:23:00 |
|---|---|---|

Bordo: : (Nessun bordo)

| Pagina 66: [141] Formattato | stefano federico | 02/02/19 19:23:00 |
|---|---|---|

Bordo: : (Nessun bordo)

| Pagina 66: [142] Formattato | stefano federico | 02/02/19 19:23:00 |
|---|---|---|

Bordo: : (Nessun bordo)

| Pagina 66: [143] Formattato | stefano federico | 02/02/19 19:23:00 |
|---|---|---|

Bordo: : (Nessun bordo)

| Pagina 66: [144] Formattato | stefano federico | 02/02/19 19:23:00 |
|---|---|---|

Bordo: : (Nessun bordo)

| Pagina 66: [145] Formattato | stefano federico | 02/02/19 19:23:00 |
|---|---|---|

Bordo: : (Nessun bordo)

| Pagina 66: [146] Formattato | stefano federico | 02/02/19 19:23:00 |
|---|---|---|

Bordo: : (Nessun bordo)

| Pagina 66: [147] Formattato | stefano federico | 02/02/19 19:23:00 |
|---|---|---|

Bordo: : (Nessun bordo)

| Pagina 66: [148] Formattato | stefano federico | 02/02/19 19:23:00 |
|---|---|---|

Bordo: : (Nessun bordo)

| Pagina 66: [149] Formattato | stefano federico | 02/02/19 19:23:00 |
|---|---|---|

Bordo: : (Nessun bordo)

| Pagina 66: [150] Formattato | stefano federico | 02/02/19 19:23:00 |
|---|---|---|

Bordo: : (Nessun bordo)

| Pagina 66: [151] Formattato | stefano federico | 02/02/19 19:23:00 |
|---|---|---|

Bordo: : (Nessun bordo)

| Pagina 66: [152] Formattato | stefano federico | 02/02/19 19:23:00 |
|---|---|---|

Bordo: : (Nessun bordo)

| Pagina 81: [153] Eliminato | stefano federico | 14/01/19 14:35:00 |
|---|---|---|

| Pagina 82: [154] Eliminato | stefano federico | 14/01/19 14:07:00 |
|---|---|---|
| Pagina 84: [155] Eliminato | stefano federico | 25/01/19 06:06:00 |

| Pagina 84: [155] Eliminato | stefano federico | 25/01/19 06:06:00 |
|---|---|---|

| Pagina 84: [155] Eliminato | stefano federico | 25/01/19 06:06:00 |
|---|---|---|

| Pagina 84: [155] Eliminato | stefano federico | 25/01/19 06:06:00 |
|---|---|---|

| Pagina 84: [155] Eliminato | stefano federico | 25/01/19 06:06:00 |
|---|---|---|

| Pagina 84: [155] Eliminato | stefano federico | 25/01/19 06:06:00 |
|---|---|---|

| Pagina 84: [155] Eliminato | stefano federico | 25/01/19 06:06:00 |
|---|---|---|

| | | |
|---|---|---|
| **Pagina 84: [155] Eliminato** | **stefano federico** | **25/01/19 06:06:00** |
| **Pagina 84: [155] Eliminato** | **stefano federico** | **25/01/19 06:06:00** |
| **Pagina 84: [155] Eliminato** | **stefano federico** | **25/01/19 06:06:00** |
| **Pagina 84: [155] Eliminato** | **stefano federico** | **25/01/19 06:06:00** |
| **Pagina 87: [156] Eliminato** | **stefano federico** | **14/01/19 14:31:00** |

---

## Referee Report (RR1)

**Review of nhess-2018-319 – version 2:**

**Title:** "Preliminary results of the impact of lightning and radar reflectivity factor data assimilation on the very short term rainfall forecasts of RAMS@ISAC: application to two case studies in Italy." by Federico et al.

**Recommendation:** Major revisions.

**Main Comments:**

(1) While the authors have gone at length to address some of my earlier concerns, I still found most of the analysis presented (especially on pages 19-22) relatively rudimentary. The text is essentially reduced to unnecessarily detailed and repetitive descriptions of rainfall plots. More targeted, concise descriptions clearly highlighting the pros and cons of each DA methods should be elaborated instead. As indicated in my original review, the authors should - in that regard - show Roebber performance diagrams in their analysis to provide clearer, concise estimates of the performance of their forecasts. Additionally and as also indicated in my original review, soundings complemented with horizontal cross-sections of RH/Qv should be provided to highlight, quantitively, how the RH/Qv field are adjusted by each respective DA approach at the analysis time (radar vs lightning). Such analysis is highly desired and, arguably, critical for readers to gain a better appreciation of the first order impact of the DA, especially given that both observations used essentially adjust the same field (cf main comment 2 below).

(2) Most importantly, as indicated in my original review, the current data assimilation set up suffers from one major drawback in that both the VAR assimilation of radar reflectivity factor and the nudging of lightning data essentially adjust the same variable (Qv ~ RH via Qv/QVsat) resulting in redundant overlap. A proper combined DA set up should ensure that adjustments from different observation sources are applied to different prognostic variables: E.g., lightning adjusts the Qv field while the reflectivity factor acts on specific hydrometeor mixing ratios (even through simple linear functional relationships). With this overlap, it is also unclear how the forecast results are impacted depending on whether the nudging of Qv is performed before or after the 3DVAR of reflectivity factor data. Until this critical point is not properly addressed, I will be inclined to maintain the current editorial decision of "Major Revisions".

(3) To complement (2), ideally, the Doppler (radial) winds should adjust the three Cartesian components of the wind field. Stating that such data "aren't yet ready" is, in my view, a succinct pretext to evade the (necessary) work.

(4) To provide a more equitable estimate of the performance of the DA method, the authors should also select at least one high impact weather event wherein the CTRL forecast performed reasonably well (e.g., a strongly forced case along a cold front). This point also was indicated in my original review but hasn't been satisfactorily addressed by the authors.

(5) For radar reflectivity factor, the DA should make use of 50-km disks instead of squares. This is easy to code in Fortran by using e.g., a 2-D mask array to find the grid points fitting within the disks.

(6) Last, I still found the level of English relatively poor and generally not suited/inappropriate for the level of a peer-reviewed journal. Because these issues are collectively substantial, I opted, for now, not to dwell on such editorial comments (including grammar).

Additional comments:

(1) Line 42: Radar data are far from being "unconventional".
(2) Line 44: Model "deficiencies" are by no means limited to oceanic regions.
(3) Line 47: "high spatio-temporal resolution"
(4) Line 75: "Lightning is another source …". Line 95: "convection-resolving"
(5) Line 99: Not "extended" but "adopted".
(6) Line 17-18, 39-40, 47,134, 249, 258-260, 266-267, 389, 711-712, 761 (among many other instances): Consider revising (grammar).
(7) Line 57-60: delete or include well known (seminal) studies such as those from Tong and Xue, Gao's, Aksoy's, Zhang's to name a few.
(8) Line 340: "subjectively as a compromise between increasing …"
(9) Line 389: downscaled ? Please elaborate/explain.
(10) Line 414: This is reminiscent of the Gaspari and Cohn function for EnKF localization.
(11) Line 480: "With these settings, larger weights are given to". Line 489: "In contrast,". Line 497 "highlights the difficulty". Line 507: "reduce the relative". Line 703: "room for improvement"
(12) Lines 527-530: redundant; delete.

---

## Referee Report (RR2)

**Review of the manuscript**

**„The impact of lightning and radar data assimilation on the performance of very short term rainfall forecasts for two case studies in Italy"**

by

Stefano Federicio, Rosa Claudia Torcasio, Elenio Aviolo, Olivier Caumont, Mario Montopoli, Luca Baldini, Gianfranco Vulpiani and Stefano Dietrich

The study discusses the impact of the assimilation of lightning and radar reflectivity data on the performance of very short-range rainfall forecasts for two convective case studies in Italy. They showed that especially the combined assimilation of both observation types has a clear and positive impact on the forecast performance.

The manuscript is interesting and tackles a very important subject, since the forecast of severe precipitation is still a major weakness of current forecast systems. With the supplemental material, the methodology is now described with enough detail. In addition, most of my other comments are discussed in the new manuscript. I am nevertheless still concerned about your coarse vertical resolution. Since I guess that you anyway plan to increase the number of vertical levels in the future, the paper would greatly benefit when you include one first test with a higher horizontal resolution. If the results are similar or the same – fine.

Therefore, I stick to it. A **major revision** is needed before I can suggest the publication of the manuscript.

**Comments**

- To generalize your results, a simulation with a larger number of vertical levels is needed for at least one of the cases.  If it shows the same or similar results, you can be sure that your methodology works as expected.
- The error value of 1 to 3 dBz seems to be too small, making the system very sensitive to the radar data. Especially when combining this with a pure sampling of the radar data sounds dangerous to me. Please explain why you use this error value.
- You mention in the new manuscript that it is a limitation of the current manuscript that the R10 run is not updated after the acquisition of new data. True, but this needs to be quantified in a way. Depending on the situation, it is well possible that your very short forecasts are not influenced by this weakness. But it is necessary to show it.
- The readability of the original proposal was better. Therefore, the English needs considerable improvement before the manuscript can be accepted.

---

## Referee Report (RR3)

**Review of the manuscript**

**„The impact of lightning and radar data assimilation on the performance of very short term rainfall forecasts for two case studies in Italy"**

by

Stefano Federicio, Rosa Claudia Torcasio, Elenio Aviolo, Olivier Caumont, Mario Montopoli, Luca Baldini, Gianfranco Vulpiani and Stefano Dietrich

The study discusses the impact of the assimilation of lightning and radar reflectivity data on the performance of very short-range rainfall forecasts for two convective case studies in Italy. They showed that especially the combined assimilation of both observation types has a clear and positive impact on the forecast performance.

The manuscript is interesting and tackles a very important subject, since the forecast of severe precipitation is still a major weakness of current forecast systems. With the supplemental material and the new material provided in the text, all my former comments are reasonably discussed.

One thing that is still open - the language needs a revision. English is not my mother tongue, but I stumbled over several things (some examples below in the minor comments – definitely not complete).

After a language revision, **I suggest the publication of the manuscript**.

**Minor Comments**

- Page 1 lines 16-17:  …two severe weather events that occurred in Italy … over central Italy that occurred o 16 September 2017.
- Page 1, line 29: … purpose because it changes a missed forecast …
- Page 1, line 32:  … both data set are assimilated.
- Page 2, line 60 to 62: using the AROME model … using the HARMONIE model … using the JNoVa model
- Page 3, line 70: polar-orbiting satellite
- Page 3, line 79: I would replace "convective scheme" by "convection scheme"
- Page 3, line 83: convective events that occurred over Greece.
- Page 5, line 155-157: Infrared satellite images …, show that the cold front moved slowly from NW to SE. … it is apparent that the well-defined cloud system … caused most of the daily …
- Page 6, line 181: occurred within a few hours.
- Page 6, line 186/187: … a trough extended from …
- Page 6, line 188: "low pressure system", instead of "pressure low"
- Page 6, line 189: It is noted that … favoured …
- Page 7, line 196: … the Livorno and Serrano cases were similar and represented …
- Page 7, line 197: … the Livorno case was more intense …
- Page 7, line 205:  It is well evident that the cloud system was associated with a cold front …

- Page 7, line 212: …reflectivity factors up to … Other clouds caused …
- Page 7, line 213: The CAPPI shown in Figure 10a is the last one assimilated ….
- Page 7, lines 214 and 218: "described in detail in section …" instead of "shown in section …"
- Page 7, line 216: … reflectivity factors up to …
- Page 8, lines 241 to 243: Here it sounds as if it is not possible to implement the third domain only because of the complex orography. Clarify the sentence. I guess you mean:
  - Technically an operational implementation would be possible if enough computer performance is available
  - It is definitely not possible to do this for every region in Italy unless you have enough computer power to cover whole Italy with a 1 km nest
  - And the implementation needs careful testing – this is what you do in your manuscript
- Page 9: When you describe your simulations, I would consistently use past tense because the simulations are finished.
- Page 11, line 332: Use "with sensitivity studies for …" instead of "by trial and errors considering ".
- Page 11, line 332:  HYMEX-SOP1 occurs here for the first time. Describe the abbreviation here and not on the bottom of page 15.
- Page 12, line 354: What is DE?
- Page 13, line 407: It assumes a Marshall-Palmer hydrometeor size distribution …
- Page 14, line 413: … the control simulation …
- Page 15, line 453/454: … between 6 September and 6 November 2012; …
- Page 15, line 463: … choice is partially justified due to the sampling of …
- Page 15, line 464: … every fifth grid point …
- Page 15, lines 471/472: Here it is not necessary to describe HYMEX-SOP1
- Page 16, line 505: Since lightning data assimilation also adjusts only the water vapour …
- Page 18, line 557: Use "radar forward operator " instead of "forward radar operator"
- Page 19, line 575: The RADLI forecast …
- Page 19, line 580: A performance diagram …
- Page 20, line 605: The location of the maximum is well represented, but the forecasted value … underestimated …
- Page 20, lines 627/628: … heavy rainfall that occurred over the region.
- Page 22, line 675/676: … caused by deeper convection …
- Page 22, line 682: Comparing RAD and LIGHT, it is evident that …
- Page 22, line 688: … over the western part …
- Page 22, line 689: … reflectivity factor data assimilation.
- Page 24, line 751: "to some extent" instead of "in some measure"
- Page 25, line 780: "forecast range" instead of "forecasting time"

---

## Author Response (AR2)

Dear Editor,

First of all, we acknowledge both reviewers for their useful comments. We answered to all points raised by the reviewers but one (the third point of Reviewer 1 cannot be answered at the current development state of observations and data assimilation software and it is definitively out of the scope of this paper). We are also a bit surprised to see again the comment 2 of the Reviewer 1. We already answered to this point in the first review. Anyway, we added material to better answer to this point (Figure 18, Figure 19 and comments).

In order to keep the paper length short, most of the answers are provided in the supplemental material, and are discussed in the paper when necessary. This choice was made because most of the answers requested additional simulations with discussion and figures. Putting this material in the paper would result in a very long paper. We adjusted the paper according to the results of the supplemental material. In particular, we added the following sensitivity tests: a) increasing the observation error of radar reflectivity factor; b) changing the shape of the searching area to compute the relative humidity pseudo-profile; c) updating IC/BC as new observations are available; d) increasing the vertical resolution of RAMS@ISAC by using 42 vertical levels. All these tests generalized the results of the paper.

Both reviewers consider the English level poor. We did our best. We corrected some errors following the comments of Reviewer 1. The journal provides a copy-editing service that will solve most language problems. If requested, we could use an external copy-editing service if the paper reaches the minor revision status.

**Reviewer #1:**

**Title: "**Preliminary results of the impact of lightning and radar reflectivity factor data assimilation on the very short term rainfall forecasts of RAMS@ISAC: application to two case studies in Italy." by Federico et al.

**Recommendation:** Major revisions.

**Main Comments:**

(1) While the authors have gone at length to address some of my earlier concerns, I still found most of the analysis presented (especially on pages 19-22) relatively rudimentary. The text is essentially reduced to unnecessarily detailed and repetitive descriptions of rainfall plots. More targeted, concise descriptions clearly highlighting the pros and cons of each DA methods should be elaborated instead. As indicated in my original review, the authors should - in that regard - show Roebber performance diagrams in their analysis to provide clearer, concise estimates of the performance of their forecasts. Additionally and as also indicated in my original review, soundings complemented with horizontal cross- sections of RH/Qv should be provided to highlight, quantitively, how the RH/Qv field are adjusted by each respective DA approach at the analysis time (radar vs lightning). Such analysis is highly desired and, arguably, critical for readers to gain a better appreciation of the first order impact of the DA, especially given that both observations used essentially adjust the same field (cf main comment 2 below).

We added the Roebber performance diagram in the revised version of the paper. The discussion on the precipitation fields for different VSF was shortened, but we believe that information provided at this stage is essential and should not be further reduced. We included, for the second stage of the Livorno case, a map showing the averaged water vapor mixing ratio between 3 and 10 km for the different model settings (Figure 18) and vertical cross-sections of relative humidity (Figure 19). The specific VSF was chosen because more information about this phase of the storm can be found in the paper when discussing the cases studies (Section 2) and radar reflectivity factor data assimilation (Section 3.1). So, this VSF is better supported by the paper discussion. Results show clear differences between radar and lighting data assimilation despite they assimilate the same both the water vapour mixing ratio.

(2) Most importantly, as indicated in my original review, the current data assimilation set up suffers from one major drawback in that both the VAR assimilation of radar reflectivity factor and the nudging of lightning data essentially adjust the same variable (Qv ~ RH via Qv/QVsat) resulting in redundant overlap. A proper combined DA set up should ensure that adjustments from different observation sources are applied to different prognostic variables: E.g., lightning adjusts the Qv field while the reflectivity factor acts on specific hydrometeor mixing ratios (even through simple linear functional relationships). With this overlap, it is also unclear how the forecast results are impacted depending on whether the nudging of Qv is performed before or after the 3DVAR of reflectivity factor data. Until this critical point is not properly addressed, I will be inclined to maintain the current editorial decision of "Major Revisions".

The fact that different observations adjust the same field is appropriate. Of course, it could create redundancy (observations are of different type, however, and this redundancy often doesn't occur) but it avoids using simplified assumptions of the relationship between reflectivity and hydrometeors. Also, the method leave to the model the task of evolving water vapour added/subtracted, which is a good feature. The analyses of the two observations are different for two reasons: a) observations are different, as clearly stated in Section 3.3; b) lightning and radar data assimilation have different impacts on the relative humidity (or water vapor mixing ratio) via the data assimilation system.

The method used to assimilate radar reflectivity is well known in the bibliography and was shown to have a huge impact on the analysis also when compared to methods using reflectivity – hydrometeor's mixing ratios relationships (Fabry and Sun, 2010). The method used to assimilate lightning is well known in bibliography (Fierro et al., 2012) and widely applied. We are not discussing the methods, we are applying them in a way that is simple, straightforward and effective for rainfall VSF of two challenging cases.

It was clearly shown that assimilating lightning or radar reflectivity factor had a different impact on the precipitation field (3 VSF), on the evolution of water vapor (Section S1 of the supplemental material) and, in this version of the paper, on the maps of water vapor mixing ratio averaged between 3 and 10 km and on cross sections of relative humidity for one VSF. Differences are also apparent and interesting for other VSFs, but, as stated above, the VSF chosen is better supported by the paper.

The problem of assimilating first lightning and then radar and vice-versa is not very well understood because lightning is assimilated before and after radar data and vice-versa in the current setting (there is not a specific order). We could skip the first radar file to start with lighting data assimilation (we note that we did this experiment several times in the past, also for Serano and Livorno, and the results of this paper remain valid), but this is a different problem because part of the data are missing.

In summary, to better answer to this point compared to the already revised version of the paper, we added: a) a discussion in Section 3.3 to better support the results given in Section 4; b) maps of averaged mixing ratio between 3 and 10 km at the end of the assimilation period for one of the VSF; c) cross sections of relative humidity at the end of the assimilation period for one VSF.

 (3) To complement (2), ideally, the Doppler (radial) winds should adjust the three Cartesian components of the wind field. Stating that such data "aren't yet ready" is, in my view, a succinct pretext to evade the (necessary) work.

This comment seems inappropriate considering the amount of work we did in the first and in this review. Again, radial velocity is not available at the moment for the motivation already provided in the paper and its assimilation is outside the scope of this paper.

(4) To provide a more equitable estimate of the performance of the DA method, the authors should also select at least one high impact weather event wherein the CTRL forecast performed reasonably well (e.g., a strongly forced case along a cold front). This point also was indicated in my original review but hasn't been satisfactorily addressed by the authors.

We considered this point in the new Section S.5 of the supplemental material. In particular, we considered a case study of a well predicted event in Rome (5 November 2017). We showed the limited impact of lightning data assimilation for this case, despite the large number of lightning observed for the event.

(5) For radar reflectivity factor, the DA should make use of 50-km disks instead of squares. This is easy to code in Fortran by using e.g., a 2-D mask array to find the grid points fitting within the disks.

We applied a well-known and widely accepted assimilation method. In the original formulation, a square is used. We agree that a circular shape could be also a good choice. However, the impact of this shape is expected to be small. We showed this point with a sensitivity experiment using a 50 km diameter disk rather than a square with 50 km edge in the assimilation of radar reflectivity factor. The experiment is discussed in Section S4 of the supplemental material: "Sensitivity to radar formulation".

 (6) Last, I still found the level of English relatively poor and generally not suited/inappropriate for the level of a peer-reviewed journal. Because these issues are collectively substantial, I opted, for now, not to dwell on such editorial comments (including grammar).

We did our best to improve the English. Consider also that the journal provides a copy-editing service before the publication of the paper. If requested by the reviewer/editor we can use an external copy editing service if the paper goes to minor revision status.

Additional comments:

    (1) Line 42: Radar data are far from being "unconventional".

The sentence was changed to avoid "unconventional".
(2) Line 44: Model "deficiencies" are by no means limited to oceanic regions.
Ok. We deleted "and model deficiencies".
(3) Line 47: "high spatio-temporal resolution"
Corrected.Thanks.
(4) Line 75: "Lightning is another source ...". Line 95: "convection-resolving"
Ok.
(5) Line 99: Not "extended" but "adopted".
Ok.
(6) Line 17-18, 39-40, 47,134, 249, 258-260, 266-267, 389, 711-712, 761 (among many other instances): Consider revising (grammar).
Ok. We changed/corrected these sentences. Also in other parts of the paper.
(7) Line 57-60: delete or include well known (seminal) studies such as those from Tong and Xue, Gao's, Aksoy's, Zhang's to name a few.
Deleted.
(8) Line 340: "subjectively as a compromise between increasing ..."
Ok.Thanks.
(9) Line 389: downscaled ? Please elaborate/explain.
Ok. We used downscaled. The explanation is already given.
(10) Line 414: This is reminiscent of the Gaspari and Cohn function for EnKF localization.
Ok. No actions were taken.
(11) Line 480: "With these settings, larger weights are given to". Line 489: "In contrast,". Line 497 "highlights the difficulty". Line 507: "reduce the relative". Line 703: "room for improvement"
Ok.Thanks.
(12) Lines 527-530: redundant; delete.
These lines were added for a specific request of Reviewer 2. We wouldn't delete them.

**Reviewer #2:**

**„The impact of lightning and radar data assimilation on the performance of very short term rainfall forecasts for two case studies in Italy"**

by

Stefano Federicio, Rosa Claudia Torcasio, Elenio Aviolo, Olivier Caumont, Mario Montopoli, Luca Baldini, Gianfranco Vulpiani and Stefano Dietrich

The study discusses the impact of the assimilation of lightning and radar reflectivity data on the performance of very short-range rainfall forecasts for two convective case studies in Italy. They showed that especially the combined assimilation of both observation types has a clear and positive impact on the forecast performance.

The manuscript is interesting and tackles a very important subject, since the forecast of severe precipitation is still a major weakness of current forecast systems. With the supplemental material, the methodology is now described with enough detail. In addition, most of my other comments are discussed in the new manuscript. I am nevertheless still concerned about your coarse vertical resolution. Since I guess that you anyway plan to increase the number of vertical levels in the future, the paper would greatly benefit when you include one first test with a higher horizontal resolution. If the results are similar or the same – fine.

Therefore, I stick to it. A **major revision** is needed before I can suggest the publication of the manuscript.

In the first review of this paper we answered to the points of Reviewer 2 by discussion, i.e. without the support of sensitivity tests. However, they are all good points and deserves further investigation, as requested by the reviewer. In this review, we answer to these points using the results of sensitivity tests.

**Comments**

- To generalize your results, a simulation with a larger number of vertical levels is needed for at least one of the cases. If it shows the same or similar results, you can be sure that your methodology works as expected.

This point is discussed in Section S4: Sensitivity to model simulation. We considered a simulation with 42 vertical levels (the future operational setting of the model). We considered only the Livorno case. The results are similar to those of 36 levels showing that the findings of this paper are not sensitive to the number of vertical levels, at least for the numbers of levels considered. However, the results show that increasing the vertical levels from 36 to 42 should positively impact the rainfall VSF of the CTRL setting. This (small) improvement, however, is not transferred to the VSF assimilating both lightning and radar reflectivity factor, likely because the background error matrix in the 3D-Var is not optimally set for RAMS@ISAC with 42 levels.

- The error value of 1 to 3 dBz seems to be too small, making the system very sensitive to the radar data. Especially when combining this with a pure sampling of the radar data sounds dangerous to me. Please explain why you use this error value.

This point is discussed in Section S4: Sensitivity to radar formulation. The choice of the error value when computing pseudo-profiles is not very important for the cases considered in this paper. Motivations are explained in section S4. In general, however, this error could be too small, as suggested by the reviewer. We highlighted this comment in the paper.

- You mention in the new manuscript that it is a limitation of the current manuscript that the R10 run is not updated after the acquisition of new data. True, but this needs to be quantified in a way. Depending on the situation, it is well possible that your very short forecasts are not influenced by this weakness. But it is necessary to show it.

This point is discussed in Section S4: Sensitivity to model formulation. This sensitivity test did not show an important impact on the results of this paper. The motivations are discussed in Section S4.

- The readability of the original proposal was better. Therefore, the English needs considerable improvement before the manuscript can be accepted.

We did our best to improve the English, and corrections have been also made by Reviewer 1. The journal provides a copy-editing service before the publication of the paper. If requested by the reviewer/editor we can contact an external copy editing service if the paper goes to minor revision status.

**LIST OF RELEVANT CHANGES**

We introduced the performance diagram in Figures 15-17 (panels f), as requested by Reviewer 1.

A discussion was put in Section 3.3 to clarify why we expect different results for very short term forecasts assimilating radar reflectivity factor or lightning.

Figure 18 and 19 are new and are used to show that, despite radar reflectivity factor and lightning data assimilation both adjust the water vapour mixing ratio, results of the experiment assimilating either radar reflectivity factor or lightning can be quite different. These figures, the discussion put in Section 3.3, and the results already presented in the paper should definitively answer to the major point 2 of Reviewer 1.

The sensitivity tests requested by Reviewer 2 were put in the supplemental material of the paper. The results of these sensitivity tests are recalled in the paper whenever necessary. The sensitivity tests are the following: a) increasing the observation error of radar reflectivity factor; b) updating IC/BC as new observations are available; c) increasing the vertical resolution of RAMS@ISAC by using 42 vertical levels.

Section S5 of the supplemental material discusses a case study well predicted by the model and the low impact of lightning data assimilation for this case. This case was added to provide a more equitable evaluation of DA.

A sensitivity test is discussed in the supplemental material to answer to the point 5 of Reviewer 1.

We corrected some English errors following the comments of Reviewer 1. More suggestions are welcome. The journal provides a copy-editing service that will solve most language problems.

[revised manuscript text omitted]

**Pagina 37: [4] Eliminato**      **stefano federico**      **05/06/19 13:10:00**

b)                                                          c)

[Figure]

*00UTC 16 sep*      *12UTC 16 sep*      *00UTC 17 sep*

**Pagina 38: [5] Eliminato**      **stefano federico**      **12/06/19 07:32:00**

b)                    c)

[Figure]

*12UTC 09 sep*                    *00UTC 10 sep*                    *12UTC 10 sep*

*Supplemental material of the paper: nhess-2018-319*

**The impact of lightning and radar reflectivity factor data assimilation on the very short term rainfall forecasts of RAMS@ISAC: application to two case studies in Italy**

Stefano Federico[1], Rosa Claudia Torcasio[1], Elenio Avolio[2], Olivier Caumont[3], Mario Montopoli[1], Luca Baldini[1], Gianfranco Vulpiani[4], Stefano Dietrich[1]

1. ISAC-CNR, via del Fosso del Cavaliere 100, Rome, Italy
2. ISAC-CNR, zona Industriale comparto 15, 88046 Lamezia Terme, Italy
3. CNRM UMR 3589, University of Toulouse, Météo-France, CNRS, 42 avenue G. Coriolis, 31057 Toulouse, France
4. Dipartimento Protezione Civile Nazionale Ufficio III - Attività Tecnico Scientifiche per la Previsione e Prevenzione dei Rischi, 00189 Rome

**S1 Introduction**

In this supplemental material, we discuss several sensitivity tests of lightning and radar reflectivity factor data assimilation. In particular: a) the contribution of data assimilation to the evolution of total water for each source of data is considered in Section S2; b) the sensitivity of rainfall VSF to the formulation of lightning data assimilation is discussed in Section S3; c) the sensitivity of rainfall VSF to two specific aspects of radar reflectivity factor data assimilation is considered in Section S4; d) the sensitivity of rainfall VSF to RAMS@ISAC setting is discussed in Section S5. Section S6 shows the impact of lightning data assimilation for a case study well predicted by the control forecast, which doesn't assimilate neither lightning nor radar reflectivity factor. A different representation of the Figures 15-17 of the paper is provided in Section S7. The form of the forward radar operator is provided in Section S8. Conclusions are given in section S9. Table 1 shows the list of the simulations discussed in this supplemental material.

**S2 Evolution of total water**

Because both lightning data assimilation and radar reflectivity factor data assimilation adjust the water vapour mixing ratio ($q_v$), it is interesting to evaluate the contribution of each data source to the $q_v$ adjustment including in this evaluation the assimilation phase (0-6 h).

Fierro et al. (2015) used the total water substance mass (accumulated precipitation + total hydrometeors and water vapour mass) to quantify the impact of lightning data assimilation by nudging. Here we use a similar approach. More specifically, we consider the forecasted accumulated precipitation and the total hydrometeors and water vapour mass averaged over the grid columns. Moreover, we averaged all VSFs for Serano and Livorno. Figure S1a shows the evolution of accumulated precipitation forecast, while Figure S1b shows the evolution of hydrometeors plus water vapour mass forecast.

Figures S1a and S1b show that flashes add less water vapour compared to radar reflectivity factor data assimilation and, of course, RADLI has the largest impact. In particular, the total water mass added to the background at the end of VSF is 2.5%, 5.7% and 7.4% of the background value for LIGHT, RAD and RADLI, respectively.

Interestingly, the total water mass added by RADLI to the background is less than the sum of the total water masses added by RAD and LIGHT. This happens because RAMS-3DVar adds water to the background limiting the impact of nudging during the simulation and vice-versa. Accumulated precipitation accounts for the largest part of the water mass added to the simulation, similarly to Fierro et al. (2015). At the end of the assimilation phase (6h), the evolution of the hydrometeors plus water vapour mass converges towards the background as boundary conditions propagate into the domain.

*S3 Sensitivity to nudging formulation*

As stated in Section 3.2 of the paper, the application of the Fierro et al. (2012) method to RAMS@ISAC is not straightforward. Furthermore, the optimal setting of the coefficients of Eqn. (1) (see the paper for the expression of the equation) depends on the case study. For these reasons, it is important to evaluate the sensitivity of the results to the nudging formulation. For this purpose, we show the variability of ETS and POD scores with A and B coefficients of Eqn. (1). The scores are computed considering all VSF of the two case studies for different configurations: A_76 has the coefficients A=0.76 and B=0.25; LIGHT has A=0.86 and B=0.15 (default setting), SAT has A=1.01 and B=0; RADLI has A=0.86 and B=0.15 (default setting).

Scores are computed for RAMS@ISAC second domain considering the nearest neighbourhood rainfall for all VSF of Serano and Livorno. ETS score (Figure S2a) shows that all configurations assimilating either lightning or radar reflectivity factor or both observations improve the forecast

for all thresholds. RADLI has the best ETS for rainfall intensity larger than 32 mm/3h in agreement with the results of the three VSF discussed in the paper.

The simulations assimilating lightning perform better than simulations assimilating radar reflectivity factor for thresholds below 32 mm/3h because they have less false alarms (not shown). A_76 has the worst score among all simulations assimilating lightning. The comparison between LIGHT and SAT shows mixed results: SAT performs better up to 32 mm/3h, while LIGHT is better for higher thresholds. This behaviour is confirmed by the POD (Figure S2b). A visual inspection of the model output reveals that, for high rainfall intensities, SAT generates spurious convection in some areas while misses convection in other areas that are correctly forecast by LIGHT.

Lynn et al. (2015) implemented a method suggested by Fierro et al. (2012) to suppress spurious convection in WRF (Weather Research and Forecasting Model). This method compares the lightning forecast during the assimilation period with observations to filter out spurious convection. The application of the methodology on 10 July 2013 improved the forecast of the squall line from Texas to Iowa, which was the focus of the forecast on that day; however, the application of the method to 19 and 21 March 2012 over the CONUS gave mixed results, improving the forecast in the first 6h and worsening it in the following hours. The implementation of this method could be used in RAMS@ISAC in future applications of the nudging scheme, to suppress spurious convection.

It is finally noted that RAD and RADLI have high POD values for all thresholds, nevertheless their ETS is below that of LIGHT and SAT for rainfall intensities up to 32 mm/3h for RADLI and up to 42 mm/3h for RAD. This behaviour is caused by the larger number of false alarms in simulations assimilating radar reflectivity factor compared to those assimilating lightning. This result shows again that RAD and RADLI configurations have a wet bias. In particular, the frequency bias of RAD and RADLI configuration is about 3 for thresholds between 20 and 40 mm/3h.

*S4 Sensitivity to radar formulation*

In this section sensitivity tests involving two different settings of radar reflectivity factor data assimilation are performed: a) observation error (1 to 3 dBz for the default setting); b) the shape of the area used for computing the relative humidity pseudo-profiles.

We limit the discussion to the Livorno case, which is the most intense between the two events considered in the paper.

For the sensitivity to the radar reflectivity factor observation error, it is important to note that this
error is used when computing the relative humidity pseudo profiles and not in RAMS-3DVar, where
the NMC method (Parrish and Derber, 1992) is used. Because the model missed the event, the
assimilation of radar reflectivity factor caused a model wetting. This humidity, however, is mainly
added for the following reason: RAMS@ISAC doesn't simulate any reflectivity factor while the radars
show positive values of reflectivity factor (for example most of the relative humidity added over
central Italy and over Sardinia is produced by this occurrence). When this happens, the model is
saturated above the LCL where the observed reflectivity factor is greater than zero and the error of
radar observations is not used (the error of radar reflectivity factor is used for computing pseudo-
profiles, which are used when the background provides already a good forecast of reflectivity
factor). Although in general the error of radar reflectivity factor observations is important and a too
small value could make the method too sensitive to radar observation, especially when combined
with a pure sampling of the radar data as in our setting, this problem is less important for the case
studies considered in this paper because they are missed by RAMS@ISAC.
The shape of the area used for computing relative humidity pseudo-profiles for the radar data
assimilation is a square in this paper, according to Caumont et al. (2010). However, a circle is also a
good choice for this shape because it considers grid points equidistant from the centre along the
circumference. The impact of this geometry, however, is expected to be negligible because pseudo
profiles are less important in the data assimilation of the cases considered in this paper, as explained
above.
Figure S3 shows the precipitation forecast between 06 and 09 UTC on 10 September 2017 by the
VSF assimilating radar with the default setting (RAD), by the VSF assimilating radar reflectivity factor
with and error increased by 5 compared to the RAD simulation (in this case the radar reflectivity
factor error varies between 5dBz and 15 dBz), and by the VSF using a circle with 50 km diameter for
computing relative humidity pseudo-profiles (CIRC). There are small differences at the local scale
but the precipitation VSF are very similar for different set-up. The POD and ETS scores computed for
the ten VSF of the Livorno case (Figure S4) further confirm this result. Differences among RAD, RAD5
and CIRC are very small and increasing the radar reflectivity factor error or changing the shape of
the area used for computing relative humidity pseudo-profiles has a minor impact on the rainfall
VSF for the Livorno case study.
*S5 Sensitivity to model formulation*

In this section, we study the sensitivity of the rainfall VSF for the Livorno case to two aspects of the model formulation: a) updating initial (IC) and boundary conditions (BC) (RLAA simulation); b)

increasing the number of vertical levels from 36 to 42 (simulations CTRL42 and ANL42).

The RLAA simulation uses updated IC/BC that assimilates new data as they become available. IC and

BC for the R4 domain are interpolated from the output of R10 domain, and, in order to update IC

and BC, analyses are done for the R10 domain.

These analyses assimilate radar reflectivity factor every one-hour by RAMS-3DVar and lightning by nudging, similarly to R4 domain. The background error matrix for the RAMS-3DVar for the R10

domain is obtained applying the NMC method to the HyMeX-SOP1 period.

Ten VSF are run with R10. Each VSF lasts 9h and data assimilation is performed for the first six-hours.

Those VSF are used to create IC/BC for the RLAA simulations.

The impact of updating IC and BC for the R4 VSF is expected to be small for the setting of this paper.

The impact of BC is presumed low because both radar and lightning observations are inside the R4

domain.

The impact of updating IC is also expected to be low because even if IC are substantially changed by the radar reflectivity factor data assimilation over the R10 domain, when the VSF starts on R4 an analysis is made assimilating radar reflectivity factor on R4 domain. So, if the IC for this VSF forecast on R4 are interpolated from the R10 background (setting of the paper) the innovations given by the analysis over the R4 at initial time are large; if IC are interpolated from an R10 analysis (RLAA

setting), the innovations of the first analysis over the R4 domain are small, because IC already take into account for the radar reflectivity factor data assimilation. However, the final result is similar in both cases.

The above considerations are confirmed by the results for the Livorno case. In particular, POD and

ETS for the RLAA simulation are similar to those of RADLI forecast (Figure S5). POD for RLAA has slightly better performance (2-3%) compared to RADLI for specific thresholds, showing a positive impact of updating IC/BC as new data become available, nevertheless the impact is small and a detailed study, considering more cases, is needed to draw conclusions about this improvement.

It is important to note, however, that if the observations are close to the edge of the domain or cross the domain, the impact of BC is expected to be more important than that found in this paper.

To show the sensitivity of the results to the number of vertical levels we consider the simulation of the Livorno case using RAMS@ISAC with 42 levels (hereafter R_42) instead of 36 levels (R_36). This choice is motivated by the fact that RAMS@ISAC with 42 levels will be operational starting from

September 2019. R_42 has a higher vertical resolution than R_36. The complete list of levels used in R_36 and R_42 is reported in Table S2.

We simulated the Livorno case using R_42 and considering the assimilation of lightning and radar reflectivity factor data assimilation (ANL42). This experiment needed a control run using R_42 (CTRL42).

It is important to note that the background error matrix for RAMS@ISAC with 42 levels was interpolated/extrapolated from that of RAMS@ISAC with 36 levels (the application of the NMC method would require the simulation of the entire HyMeX-SOP1 period using R_42). While we believe that this choice is reasonable for this experiment, it could result in non-optimal adjustments given by RAMS-3DVar.

Figure S6a and S6b show, respectively, the rainfall VSF for CTRL and CTRL42 between 06 and 09 UTC on 10 September 2017, when the storm was active mainly over Lazio (Section 4.2.2 of the paper). The increasing of the number of levels did not result in an improvement of the precipitation forecast over Lazio. There are, however, differences at the local scale especially over Tuscany and NE of Italy. It is also notable the higher rainfall between Corsica and Italian peninsula for CTRL42. This feature is systematic for all VSF of the Livorno case and it is likely caused by a better representation of the interaction between the air-masses and the complex orography of Corsica in R_42. Figure S6c and S6d show the rainfall VSF between 06 and 09 UTC given by RADLI and ANL42. Differences between the two forecasts are small and at the local scale.

POD and ETS scores for R_42 considering the ten VSF of the Livorno case over the R4 domain are shown in Figure S5 for both CTRL42 and ANL42. The POD of CTRL42 is higher than that of CTRL but the improvement is small (2-3%). The POD of ANL42 is slightly worse than that of RADLI. Difference between RADLI and ANL42 could be the result of the specific case considered or a consequence of the non-optimal setting of RAMS-3DVar for ANL42.

The results for ETS score, which penalizes false alarms, show less differences between R_36 and R_42 settings.

Thus, the results of the experiment using 42 vertical levels in RAMS@ISAC are similar to those using 36 levels and show again the crucial role of lightning and radar reflectivity factor data assimilation for the successful forecast of the Livorno case.

*S6 A well predicted case study*

In this section, we show the impact of data assimilation for a case well predicted by the CTRL
simulation, without lightning or radar reflectivity factor data assimilation. To keep the discussion
concise, we limit the analysis to only lightning data assimilation.
The case study occurred on 5 November 2017 and was chosen because it is not very different from
those of Serano and Livorno from a synoptic perspective. In particular, the storm was caused by a
trough extending from northern Europe towards the Mediterranean. The interaction between the
trough and the Alpine orography caused a low pressure over the Gulf of Genova (not shown). The
storm propagated towards SE and, in these conditions, humid and unstable air masses were
advected from the Tyrrhenian Sea towards the Italian mainland.
The convection developed over the Tyrrhenian Sea and over the Italian peninsula (especially on its
western side), as shown by the lightning density observation on this day (Figure S7): more than
100.000 flashes were detected for this intense event. Moderate to heavy rainfall occurred in several
parts of Italy. In particular, between 12 and 15 UTC intense precipitation fell around Rome (Figure
S8a) with values greater than 50 mm/3h reported by several raingauges. Some areas of the city were
flooded, and problems occurred in local transportation system in outdoor activities.
The intense precipitation over Rome is well predicted by the VSF of the CTRL forecast (Figure S8b),
even if there is a shift to the north of the precipitation pattern (15-20 km). The intense precipitation
over NE of Italy and the rainfall over Liguria and Tuscany are also well forecast.
Figure S8c shows the rainfall VSF for LIGHT simulation. The VSF follows a 6 h assimilation phase (6-
12 UTC for this specific VSF), when more than 34000 flashes are assimilated in RAMS@ISAC
following the method of Fierro et al. (2012). LIGHT rainfall VSF is similar to CTRL and lightning data
assimilation has a lower impact on the rainfall VSF compared to Livorno or Serano case studies. Of
course, considering the high number of assimilated lightning, there are differences between CTRL
and LIGHT rainfall VSF, but they do not change substantially the forecast given by CTRL. Rainfall
simulated by LIGHT is shifted to the south (15-20 km) compared to CTRL, in better agreement with
observations. However, LIGHT VSF overestimates the area of intense precipitation (>30-40 mm/3h).
To discuss more in detail the lower impact of lightning data assimilation for the 5 November case
study compared to Serano and Livorno, we consider the vertical cross section of relative humidity
at 42°N (Figure S9a) and at the end of the assimilation phase (12 UTC). The vertical section shows
very humid layers (relative humidity >92.5%). One of these layers is over the Tyrrhenian Sea (11 °E
-12.5 °E). Considering that 0 °C and -25 °C isotherms heights are about 2500 m and 7000 m, it is
expected a low impact of lightning data assimilation for this layer. This is confirmed by Figure S9b, which shows the same cross section of Figure S9a for LIGHT simulation. The humid layer over the
Tyrrhenian Sea is slightly wider for LIGHT, but differences are overall small. The analyses of other
fields, as the averaged specific humidity between 3 and 10 km, also show the low impact of lightning
data assimilation for this VSF.
In conclusion, the analysis of the 5 November 2017 event, shows that the impact of lightning data
assimilation is much lower when the CTRL VSF has a good performance. Interestingly, lightning data
assimilation improves the rainfall forecast at the local scale even for well predicted events, while
overestimates the precipitation. This is the main drawback of lightning data assimilation in
RAMS@ISAC.

*S7 New plots*
Figures S10-S12 show a different representation of the Figures 15-17 of the paper. In particular, we
show the rainfall predicted by RAMS@ISAC for the three VSF considered in the paper interpolated
at the stations' positions. From Figure S12, in particular, is evident the overestimation of the
precipitation field given by both RAD and RADLI (see also Section 4.2.2 in the paper).

*S8 Forward radar operator*
In the method of Caumont et al. (2010) there is the need to simulate reflectivity factor (in dBz) from
the model output. To compute the reflectivity factor we use the forward operator of Stoelinga used
in the RIP (Read/Interpolate/Plot) software of WRF (https://dtcenter.org/wrf-
nmm/users/OnLineTutorial/NMM/RIP/index.php, last access 03 March 2019).
The software assumes Rayleigh scattering regime (at C-band this assumption can be considered as
valid for light to moderate rain) and includes the contribution of rain, snow and graupel. Particles
are assumed spherical with constant density ($\rho_r = \rho_l$=1000 kg/m$^3$; $\rho_s$=100 kg/m$^3$; $\rho_g$ =400 kg/m$^3$; $r$
stands for rain, $l$ for liquid, $s$ for snow and $g$ for graupel).
The size distribution of the hydrometeors follows an exponential distribution given by:

$$N(D) = N_0 e^{-\lambda D}$$
(S1)

Where $N_0$ is constant for each hydrometeor ($N_{0r}$=8x10$^6$,$N_{0s}$=2x10$^7$,$N_{0g}$=4x10$^6$ m$^{-4}$).
Using these assumptions, the reflectivity factor for rain $Z_{er}$, which is the sixth moment of the size
distribution, is given by:

$$Z_{er} = \Gamma(7) N_{0r} \lambda^{-7}$$
(S2)

where $\Gamma$ is the gamma function. The shape factor $\lambda$ depends on the simulated mixing ratio ($q_r$) and it is given by:

$$\lambda_r = \left( \frac{\pi N_{0r} \rho_r}{\rho_a q_r} \right)^{1/4}$$

(S3)

where $\rho_a$ is the density of dry air.

In the case of snow, the reflectivity factor $Z_{es}$ is given by:

$$Z_{es} = \Gamma(7) N_{0s} \lambda^{-7} \left( \frac{\rho_s}{\rho_i} \right)^2 \alpha$$

(S4)

where $\alpha$=0.224. The reflectivity factor for graupel is the same as (S4) with $N_{0g}$ replacing $N_{0s}$, and $\rho_g$

replacing $\rho_s$ . Since the reflectivity factor, when expressed in mm$^6$/m$^3$, is an additive quantity, the contributions of rain, snow, and graupel can be added to obtain the reflectivity factor:

$Z_{etot}=Z_{er}+Z_{eg}+Z_{es}$

and in dBz is given by:

$Z_e(dBz)=10\ log(Z_{etot}$ (in mm$^6$m$^{-3}$))

*S.9 Conclusions*

The analysis of the evolution of the total water mass shows that flashes add less water vapour to the VSF than radar reflectivity factor data assimilation. This, however, even if in agreement with other studies (Fierro et al., 2016) could be a result of the specific case studies.

The sensitivity of the rainfall VSF to the nudging formulation for lightning data assimilation shows that reducing the amount of water vapour added to RAMS@ISAC compared to the default set-up has a worse impact on ETS and POD. Nevertheless, assuming saturation (SAT) for grid points where lightning is observed gave mixed results. Spurious convection was generated in the SAT

configuration, which decreased the performance of the model for thresholds larger than 34 mm/3h.

A method proposed by Fierro et al. (2012) and used in Lynn et al. (2015) could be used in future implementations of the nudging scheme to suppress spurious convection.

Increasing the radar reflectivity factor error (RAD5) or changing the shape of the area used to compute pseudo-profiles (CIRC) had a minor impact on the rainfall VSF. Furthermore, updating

IC/BC as new data are available (RLAA) and increasing the number of vertical levels in RAMS@ISAC

(CTRL42, ANL42) gave minor changes to the rainfall VSF. Therefore, the sensitivity tests generalize the findings of the paper.

Finally, the results for a case study well predicted by the background show a limited impact of
lightning data assimilation.

Table S1: Simulations considered in this supplement material.

| Experiment | Description | Data assimilated | Model variable impacted | Note |
|---|---|---|---|---|
| CTRL | Control run | None | None | / |
| RAD | RADAR data assimilation | Reflectivity factor CAPPI (RAMS-3DVar) | Water vapour mixing ratio | / |

| | | | | |
|---|---|---|---|---|
| LIGHT | Lightning data assimilation (A=0.86; B=0.15 in Eqn (1)) | Lightning density (nudging) | Water vapour mixing ratio | / |
| RADLI | RADAR + lightning data assimilation (A=0.86; B=0.15 in Eqn (1)) | Reflectivity factor CAPPI (RAMS-3DVar) + Lightning density (nudging) | Water vapour mixing ratio | / |
| A_76 | Lightning data assimilation (A=0.76; B=0.25 in Eqn (1)) | Lightning density (nudging) | Water vapour mixing ratio | / |
| SAT | Lightning data assimilation (A=1.01; B=0. in Eqn (1)) | Lightning density (nudging) | Water vapour mixing ratio | / |
| RAD5 | RADAR data assimilation. | Reflectivity factor CAPPI (RAMS-3DVar) | Water vapour mixing ratio | As RAD simulation but with the error of radar reflectivity factor increased by 5. |
| CIRC | RADAR data assimilation | Reflectivity factor CAPPI (RAMS-3DVar) | Water vapour mixing ratio | As RAD but with a circular shape to compute relative humidity pseudo-profiles |

| | | | | |
|---|---|---|---|---|
| RLAA | RADAR + lightning data assimilation (A=0.86; B=0.15 in Eqn (1)). | Reflectivity factor CAPPI (RAMS-3DVar) + Lightning density (nudging) | Water vapour mixing ratio | As RADLI but with updated IC/BC as new data are available |
| CTRL42 | Control run | None | None | As CTRL simulation but using 42 vertical levels |
| ANL42 | RADAR + lightning data assimilation (A=0.86; B=0.15 in Eqn (1)) | Reflectivity factor CAPPI (RAMS-3DVar) + Lightning density (nudging) | Water vapour mixing ratio | As RADLI simulation but using 42 vertical levels |

Table S2: Vertical levels of RAMS@ISAC with 36 levels (default setting, R_36) and RAMS@ISAC with
42 levels (R_42).

| RAMS@ISAC CONFIGURATION | LEVEL (m) |
|---|---|
| R_36 | 0, 50, 108, 174, 250, 337, 438, 553, 686, 839, 1015, 1217, 1450, 1718, 2025, 2379, 2786, 3254, 3792, 4411, 5122, 5941, 6882, 7964, 9164, 10364, 11563, 12764, 13964, 15164, 16364, 17564, 18764, 19964, 21164, 22364 |
| R_42 | 0, 50, 106, 167, 235, 311, 396, 489, 593, 708, 836, 978, 1136, 1311, 1505, 1720, 1959, 2225, 2520, 2847, 3210, 3613, 4061, 4557, 5109, 5721, 6400, 7154, 7991, 8920, 9951, 1096, 12296, 13496, 14696, 15896, 17096, 18296, 19496, 20696, 21896, 23096 |

a)

[Figure]

b)

Figure S1: a) Evolution of accumulated precipitation for different model configurations and for all
forecast hours; b) as in a) for the hydrometeors plus water vapour mass per unit area. All quantities
are expressed in [mm] and are averaged over the number of grid columns.

a)

[Figure]

b)

[Figure]

Figure S2: a) ETS score for all VSF considered in this paper; b) as in a) for the POD score. Scores are
computed for the R4 domain considering all VSF for Livorno and Serano cases and using the nearest
neighbourhood value. Scores are computed for the nearest neighbourhood and for the thresholds:
1mm/3h, 2mm/3h and then every 2 mm/3h up to 60 mm/3h.

a)

[Figure]

b)

[Figure]

c)

[Figure]

Figure S3: a) rainfall VSF between 06 and 09 UTC on 10 September for RAD; b) as in a) for RAD5; c)
as in a) for CIRC.

a)

[Figure]

b)

Figure S4: a) POD score for Livorno; b) as in a) for ETS score. CTRL is the control simulation, RAD is the simulation assimilating radar reflectivity factor, RAD5 is the simulation with a reflectivity factor error five times that of RAD; CIRC is the simulation using a circle for computing relative humidity pseudo-profiles. Scores are computed for the R4 domain considering the ten VSF of the Livorno case. Scores are computed for the nearest neighbourhood and for the threshold of: 1mm/3h, 2mm/3h and then every 2 mm/3h up
to 60 mm/3h, considering the R4 domain and the ten VSF of the Livorno case.

a)

[Figure]

b)

Figure S5: a) POD score for Livorno; b) as in a) for the ETS score. CTRL is the control simulation, RLAA
is the simulations with updated IC/BC, CTRL42 is the control simulation using 42 model vertical level,

ANL42 is the simulation assimilating radar reflectivity factor and lightning and using 42 model vertical
levels. Scores are computed for R4 domain considering all the ten VSF of the Livorno case. Scores are
computed for the nearest neighbourhood and for the thresholds: 1mm/3h, 2mm/3h and then every 2
mm/3h up to 60 mm/3h.

a)

b)

[Figure]

c)

[Figure]

d)

[Figure]

Figure S6: rainfall VSF between 06 and 09 UTC on 10 September for CTRL; b) as in a) for CTRL42; c) as in
a) for RADLI; d) as in a) for ANL42.

[Figure]

Figure S7: a) Lightning density (lightning number per 16 km$^2$ for the whole day) recorded on 05 November
2017. The total number of flashes is shown in the title.

a)

[Figure]

b)

[Figure]

c)

[Figure]

Figure S8: a) rainfall reported by raingauges between 12 and 15 UTC on 5 November 2017. Only stations
reporting at least 0.2 mm/3h are shown. The first number in the title within brackets represents the number
of raingauges available over the domain, while the second number shows those observing at least 0.2
mm/3h; b) rainfall VSF of CTRL for the same time interval as in a); c) as in b) for LIGHT forecast.

a)

[Figure]

b)

Figure S9: a) Relative humidity longitude-height cross-section at 42°N and at the end of the assimilation
period (12 UTC on 5 November 2017) for the CTRL simulation; b) as in a) for LIGHT simulation. Only longitudes
between 5 E and 17 E and altitudes between 0 km and 10 km are shown for clarity.

[Figure]

b)

c)

[Figure]

d)

e)

[Figure]

Figure S10: a) rainfall reported by raingauges between 03 and 06 UTC on 16 September 2017. Only raingauges
observing at least 0.2 mm/day are shown. The first number in the title within brackets represents the available
raingauges, while the second number represents those observing at least 0.2 mm/3h; b) as in a) for CTRL
forecast; c) as in a) for RAD forecast; d) as in a) for LIGHT forecast; e) as in a) for RADLI forecast.

a)                                          b)

[Figure]

c)                                    d)

[Figure]

e)

[Figure]

Figure S11: a) rainfall reported by raingauges between 00 and 03 UTC on 10 September 2017. Only stations
reporting at least 0.2 mm/3h are shown. The first number in the title within brackets represents the number of
raingauges available over the domain, while the second number shows those observing at least 0.2 mm/3h; b)
as in a) for CTRL forecast; c) as in a) for RAD forecast; d) as in a) for LIGHT forecast; e) as in a) for RADLI
forecast.

a)

[Figure]

b)

c)

[Figure]

d)

e)

[Figure]

Figure S12: a) rainfall reported by raingauges between 06 and 09 UTC on 10 September 2017. For this time
period 2695 raingauges reported valid observations in the domain, however only stations reporting at least 0.2
mm/3h are shown. The first number in the title within brackets represents the number of raingauges available
over the domain, while the second number shows those observing at least 0.2 mm/3h; b) as in a) for CTRL
forecast; c) as in a) for RAD forecast; d) as in a) for LIGHT forecast; g) as in a) for RADLI forecast.